# The GATAD2B-NuRD complex drives DNA:RNA hybrid-dependent chromatin boundary formation upon DNA damage

Zhichao Liu[1], Kamal Ajit[1], Yupei Wu[2], Wei-Guo Zhu[2] & Monika Gullerova [1]✉

## Abstract

**Double-strand breaks (DSBs) are the most lethal form of DNA damage. Transcriptional activity at DSBs, as well as transcriptional repression around DSBs, are both required for efficient DNA repair. The chromatin landscape defines and coordinates these two opposing events. However, how the open and condensed chromatin architecture is regulated remains unclear. Here, we show that the GATAD2B–NuRD complex associates with DSBs in a transcription- and DNA:RNA hybrid-dependent manner, to promote histone deacetylation and chromatin condensation. This activity establishes a spatio-temporal boundary between open and closed chromatin, which is necessary for the correct termination of DNA end resection. The lack of the GATAD2B–NuRD complex leads to chromatin hyperrelaxation and extended DNA end resection, resulting in homologous recombination (HR) repair failure. Our results suggest that the GATAD2B–NuRD complex is a key coordinator of the dynamic interplay between transcription and the chromatin landscape, underscoring its biological significance in the RNA-dependent DNA damage response.**

**Keywords** NuRD Complex; GATAD2B; DNA:RNA Hybrids; DNA Damage; Chromatin Boundary
**Subject Categories** Chromatin, Transcription & Genomics; DNA Replication, Recombination & Repair; RNA Biology

## Introduction

Maintenance of genome stability is crucial for normal biological and cellular activities (Jackson and Bartek, 2009). The human genome is continuously exposed to various exogenous and endogenous factors that can lead to different types of DNA damage, of which DNA double-strand breaks (DSBs) are the most lethal. If DNA damage is left unrepaired or repaired incorrectly, it can lead to fatal consequences, such as loss of genetic information, cell death, mutagenesis, and premature aging (McKinnon, 2009; White and Vijg, 2016). To recognize, signal and correctly repair DNA breaks, cells have developed a complex pathway called the DNA damage response (DDR) (Ciccia and Elledge, 2010). The majority of DSBs are repaired by two major pathways, homologous recombination (HR) or non-homologous end-joining (NHEJ). The canonical and precise HR pathway utilizes sister chromatids as the DNA template. HR employs DNA end resection to create a single-stranded DNA (ssDNA) overhang, which in is turn used to search for and anneal with the intact DNA template of the sister chromatid (Heyer et al, 2010). On the other hand, the error-prone NHEJ may rapidly ligate two broken DNA ends (Lieber, 2010) at any stage of the cell cycle.

Recent studies have revealed that transcription at DSBs plays an important role in DDR (Aymard et al, 2014; Clouaire et al, 2018a; D'Alessandro et al, 2018; Francia et al, 2016; Ketley et al, 2022; Ketley and Gullerova, 2020; Long et al, 2021; Lu et al, 2018; Vitor et al, 2019). Specifically, phosphorylated RNA polymerase II (RNAPII) transcribes long noncoding RNA, such as damage-induced lncRNA (dilncRNA) (Michelini et al, 2017; Pessina et al, 2019) and damage-responsive transcripts (DARTs) (Burger et al, 2019). These DSB-derived long noncoding RNAs have been shown to be further processed into smaller noncoding RNAs, termed DNA damage-derived RNAs (DDRNAs) (Bonath et al, 2018; Francia et al, 2012) and are suggested to facilitate the recruitment of DDR factors, including 53BP1, to DSBs. DilncRNA can also anneal with the ssDNA overhang template after resection to form DNA:RNA hybrids(D'Alessandro et al, 2018; Francia et al, 2016). DNA:RNA hybrids are widely considered to be a transcription by-product and potential threats to genomic stability (Skourti-Stathaki and Proudfoot, 2014). However, they are also found to form at DSB sites predominantly in transcriptionally active loci (Bader and Bushell, 2020; Cohen et al, 2018; Marnef and Legube, 2021; Yasuhara et al, 2018). Furthermore, R-loops, which are DNA:RNA hybrids along with the ssDNA strand, formed behind pausing RNAPII, have been implicated in the regulation of repair pathway choices and the recruitment of DSB factors, such as CSB, RPA, Rad52, BRCA1, BRCA2, and 53BP1 (Burger et al, 2019; D'Alessandro et al, 2018; Hatchi et al, 2015; Mazina et al, 2020; Tan et al, 2020; Teng et al, 2018; Yasuhara et al, 2018). R-loops are involved in the regulation

[1]Sir William Dunn School of Pathology, South Parks Road, Oxford OX1 3RE, United Kingdom. [2]Guangdong Key Laboratory of Genome Instability and Human Disease Prevention, Shenzhen University International Cancer Center, Marshall Laboratory of Biomedical Engineering, Department of Biochemistry and Molecular Biology, Shenzhen University School of Medicine, 518055 Shenzhen, China. ✉E-mail: monika.gullerova@path.ox.ac.uk

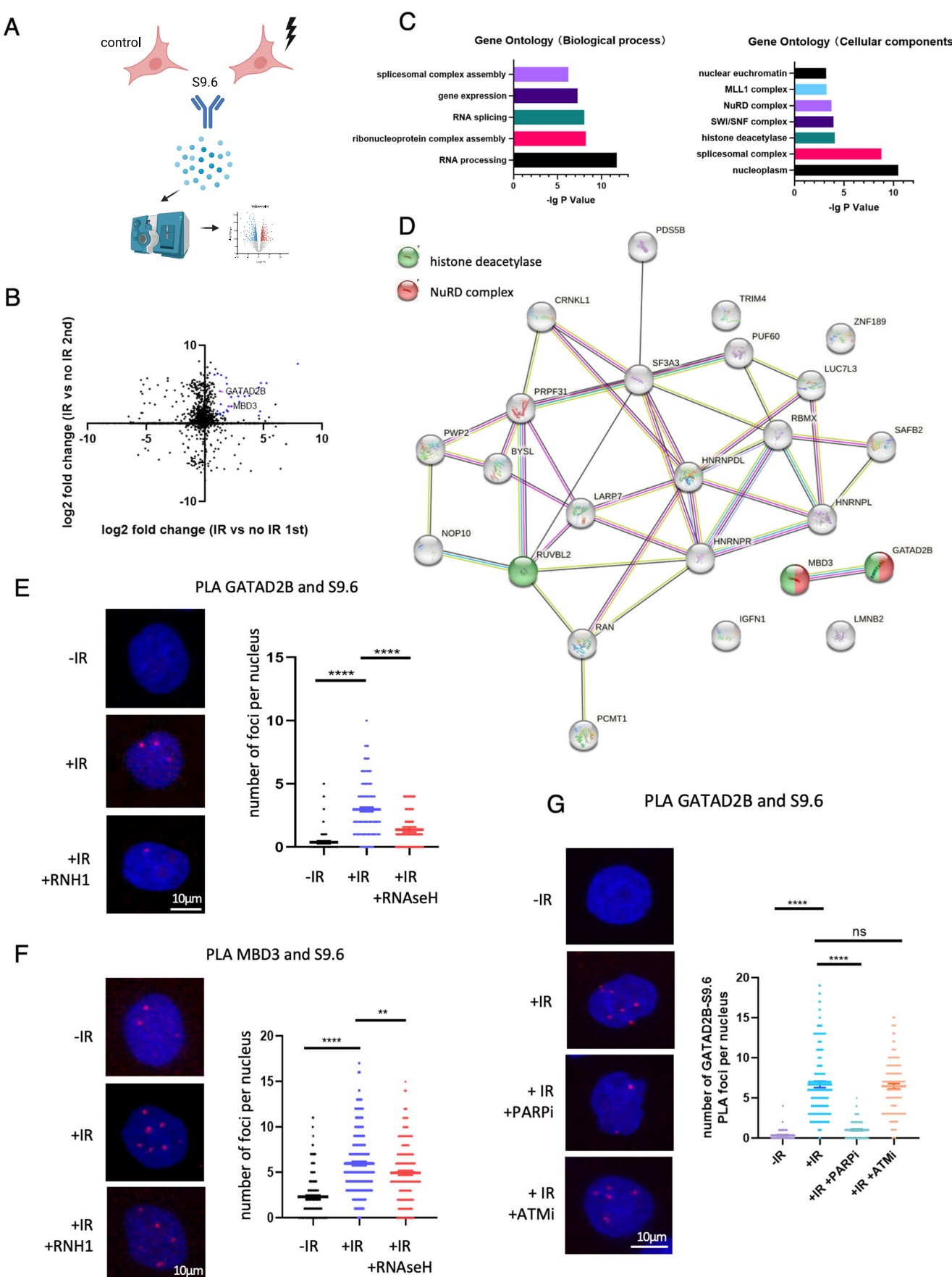

**Figure 1. Components of the NuRD complex bind to R-loops upon DNA damage.**

(A) Diagram of S9.6 mass spectroscopy-based proteomics approach. Cells were subjected to either no IR treatment or be treated with 10 Gy IR followed by 20 min recovery before being collected. Image was created with BioRender.com (B) Volcano plot showing significantly enriched S9.6 binders at 20 min after 10 Gy IR treatment in two biological replicates. GATAD2B and MBD3 are highlighted and named. (C) GO analysis of proteins significantly enriched in S9.6 IP after IR treatment. Significance was calculated by Fisher's Exact test. (D) STRING analysis of proteins significantly enriched in S9.6 IP after IR treatment. (E) PLA of GATAD2B and S9.6 in cells with or without 5 Gy IR treatment followed by 20 min recovery and overexpression of RNAseH1. IR = 5 Gy. Left: representative confocal microscopy images from three independent replicates; right: quantification of left, error bar = mean ± SEM, significance was determined using nonparametric Mann–Whitney test. ****$P \leq 0.0001$. Scale bar = 10 μm, $n > 50$ cells from three biological repeats. (F) PLA of MBD3 and S9.6 in cells with or without IR treatment followed by 20 min recovery and overexpression of RNAseH1. IR = 5 Gy. Left: representative confocal microscopy images from three independent replicates; right: quantification of left, error bar = mean ± SEM, significance was determined using nonparametric Mann–Whitney test. ****$P \leq 0.0001$, **$P \leq 0.01$. Scale bar = 10 μm, $n > 50$ cells from three biological repeats. (G) Left: Representative confocal images from three independent replicates showing PLA of GATAD2B and S9.6 in cells with or without 5 Gy IR followed by 20 min recovery, and treatment with PARP1 or ATM/ATR inhibitors. Right: quantification of left, error bar = mean ± SEM, significance was determined using nonparametric Mann–Whitney test. ****$P \leq 0.0001$. Scale bar = 10 μm, $n > 50$ cells from three biological repeats. Source data are available online for this figure.

of several biological processes under normal conditions (Garcia-Muse and Aguilera, 2019; Petermann et al, 2022). Specifically, they are associated with active transcription at promoter regions (Ginno et al, 2012; Santos-Pereira and Aguilera, 2015), facilitate transcription termination (Skourti-Stathaki et al, 2014) and regulate chromatin modifications and remodeling (Castellano-Pozo et al, 2013; Herrera-Moyano et al, 2014; Prendergast et al, 2020). The role of R-loops and DNA:RNA hybrids at DSBs, however, remains enigmatic.

Chromatin architecture is regulated by chromatin-remodeling complexes and histone-modifying enzymes. Open chromatin generally correlates with active transcription, while condensed heterochromatin is indicative of transcriptionally silent regions. The remodeling of the chromatin landscape is an important step in DDR (Price and D'Andrea, 2013). Several studies have demonstrated that upon DNA damage, chromatin opens to promote transcription, the loading of DDR factors and efficient completion of DNA repair (Hauer et al, 2017; Luijsterburg et al, 2016; Murr et al, 2006). Conversely, proteins involved in chromatin compaction and heterochromatin formation were detected around DSBs (Ayrapetov et al, 2014; Kalousi et al, 2015; Yang et al, 2017). Furthermore, recent bioimaging data have revealed opened chromatin at DSBs surrounded by compacted chromatin (Lou et al, 2019), suggesting the existence of a border between these two chromatin landscapes. Nevertheless, the mechanism and dynamics of chromatin structure and function upon DNA damage remain unclear.

In this study, we employed mass spectroscopy-based proteomics to screen for proteins that bind to DNA:RNA hybrids following DNA damage and identified GATAD2B and MBD3, two subunits of the Nucleosome Remodeling Deacetylase (NuRD) complex, as significant interactors. Both, GATAD2B and MBD3 associate with DSBs in a transcription and DNA:RNA hybrid-dependent manner. At the sites of damage, the GATAD2B–NuRD complex promotes histone deacetylation, which is followed by chromatin condensation. A lack of GATAD2B leads to excessive chromatin relaxation, resulting in hyper-resection of DNA ends and failed HR repair. Our results demonstrate that transcription and DNA:RNA hybrids recruit the GATAD2B–NuRD complex to form spatio-temporal boundaries between open and condensed chromatin around DSBs. These chromatin boundaries are crucial for terminating DNA end resection and to facilitate efficient HR repair.

# Results

## Components of the NuRD complex bind to DNA:RNA hybrids following DNA damage

To identify novel DNA:RNA hybrids interactors relevant to DDR, we employed the S9.6 antibody, which recognizes DNA:RNA hybrids and R-loops (Bou-Nader et al, 2022; Cristini et al, 2018) in conjunction with mass spectrometry-based proteomics (Fig. 1A). HEK293T cells were subjected to a 10 Gy irradiation (IR) treatment followed by a 20-min recovery period. First, we pulled down proteins bound to the S9.6 antibody to test whether we could detect known DNA:RNA hybrid interactors in our immunoprecipitated (IP) fractions. We indeed detected previously identified DNA:RNA hybrid interactors, such as DHX9 and XRN2, in our IP samples (Appendix Fig. S1A–E) (Cristini et al, 2018). Next, we performed S9.6 IP from control (−IR) and irradiated (+IR) HEK293 cells, followed by mass spectrometry-based proteomics. In our control (−IR) samples we identified many well-known DNA:RNA hybrid binders, such as DDX5, XRN2 and PARP1 (Laspata et al, 2023) (Appendix Fig. S1F). We also identified proteins that were significantly enriched in the S9.6 IP samples, specifically upon DNA damage (Fig. 1B; Appendix Fig. S1G). Gene Ontology (GO) and STRING analyses revealed an enrichment of proteins involved in RNA processing, gene expression regulation and histone deacetylation (Fig. 1C,D). Interestingly, we found MBD3 and GATAD2B, two components of the NuRD complex, among the DNA:RNA hybrid interactors specific to damage conditions (Torchy et al, 2015). The NuRD complex, which has histone deacetylation and ATP-dependent chromatin-remodeling abilities, has been found to regulate several biological processes, including the repression of transcription (Torchy et al, 2015).

To further validate whether components of the NuRD complex could interact with DNA:RNA hybrid upon DNA damage, we employed optimized proximity ligation assay (PLA) that allows for the detection of protein/R-loop complexes in vivo (Alagia et al, 2022). Using antibodies against endogenous GATAD2B or MBD3 and S9.6, we detected a significant increase in number of PLA foci following IR treatment. The number of PLA foci decreased in the presence of RNAseH1, an enzyme that resolves DNA:RNA hybrids, confirming the specificity of this experimental approach (Fig. 1E,F). Single antibodies served as negative controls for the PLA (Fig. EV1A). To test whether the binding of GATAD2B to

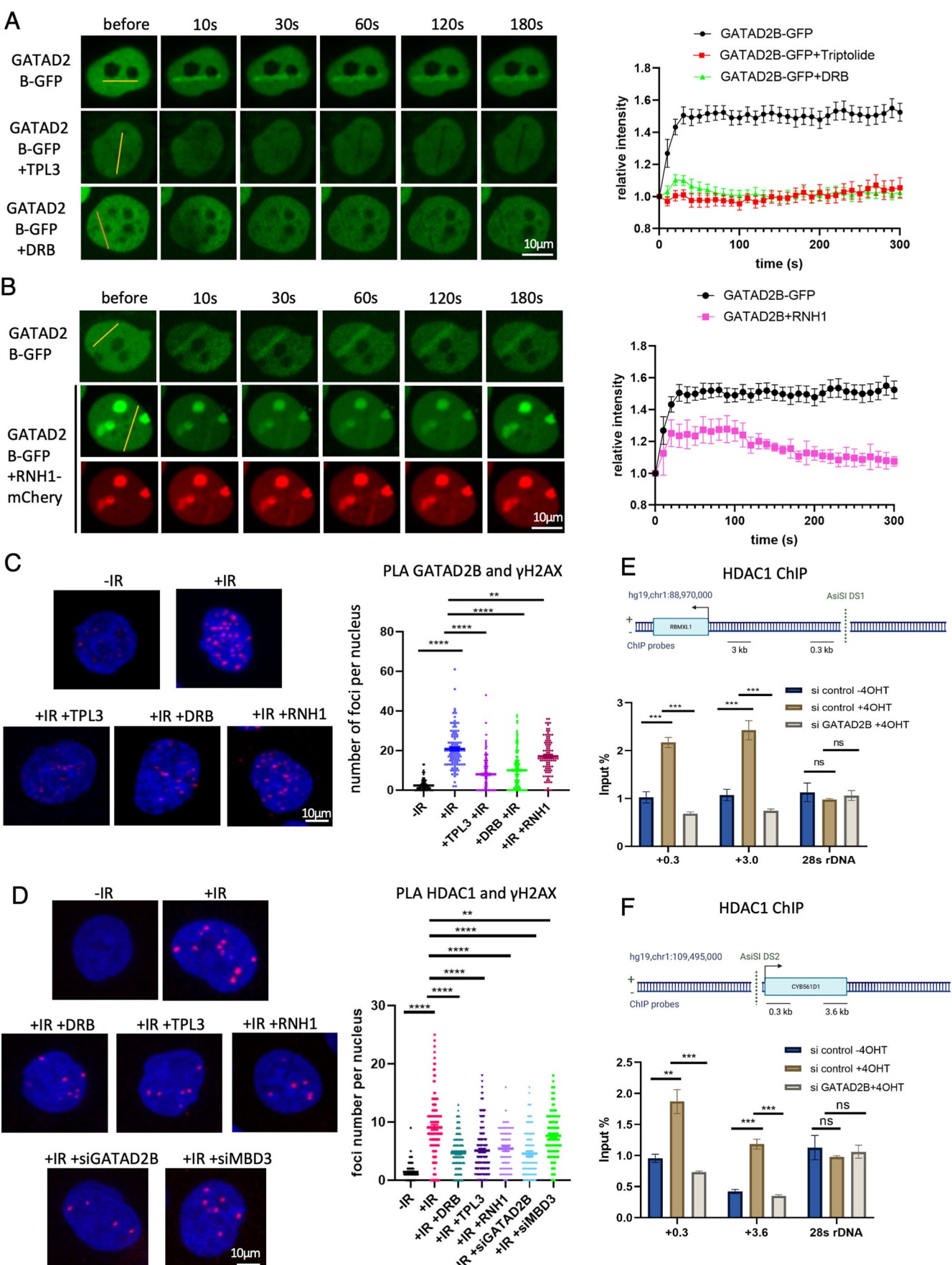

**Figure 2.   GATAD2B is recruited to DSBs in transcription and DNA:RNA hybrids dependent manner and required for HDAC1 localization to DSBs.**

(A) Laser stripping of GATAD2B-GFP cells with or without treatment with transcription inhibitors triptolite (TPL3) and DRB. Representative spinning disk confocal microscopy images from three independent replicates and quantification ($n \geq 10$) showing GFP signals before and after laser striping at indicated time points; error bar = mean ± SEM. Scale bar = 10 μm, $n > 3$. (B) Laser stripping of GATAD2B-GFP cells with or without transiently expression of RNAseH1-mCherry plasmid. Representative spinning disk confocal microscopy images from three independent replicates and quantification ($n \geq 10$) showing GFP and RFP signals before and after laser striping at indicated time points; error bar = mean ± SEM. Scale bar = 10 μm, $n > 3$. (C) PLA of GATAD2B and γH2AX in cells with or without IR treatment followed by 20 min recovery and transcription inhibition (TLP3 or DRB) or overexpression of RNAseH1. IR = 5 Gy. Left: representative confocal microscopy images from three independent replicates; right: quantification of left, error bar = mean ± SEM, significance was determined using nonparametric Mann–Whitney test. ****$P \leq 0.0001$, **$P \leq 0.01$. Scale bar = 10 μm, $n > 3$. (D) PLA of HDAC1 and γH2AX in cells with or without IR treatment followed by 20 min recovery and transcription inhibition (TLP3 or DRB), overexpression of RNAseH1 or depletion of GATAD2B and MBD3. IR = 5 Gy. Left: representative confocal microscopy images from three independent replicates; right: quantification of left, error bar = mean ± SEM, significance was determined using nonparametric Mann–Whitney test. ****$P \leq 0.0001$, **$P \leq 0.01$. Scale bar = 10 μm, $n > 3$. (E) Top: drawing showing DS1 genomic region with AsiSI cut site and position of ChIP probes. Bottom: Bar chart showing HDAC1 ChIP levels at indicated sites next to AsiSI cut in cells with or without 4 h 4OHT treatment and depletion of GATAD2B, error bar = mean ± SEM, significance was determined using nonparametric Mann–Whitney test. ***$P \leq 0.001$, n.s. not significant. (F) As in (E), for DS2. ***$P \leq 0.001$, **$P \leq 0.01$, n.s. not significant. Image was created with BioRender.com. Source data are available online for this figure.

R-loops depends on any DDR signaling pathways, such as ATM or PARP1, we performed GATAD2B and S9.6 PLA in cells treated with ATM inhibitor or PARP1 inhibitors and detected that the inhibition of PARP1 significantly decreased the binding of GATAD2B to R-loops, whereas ATM inhibition did not (Fig. 1G). The histone deacetylation activity of the NuRD complex is facilitated by the histone deacetylase 1 (HDAC1) subunit. To determine whether HDAC1 could also bind to DNA:RNA hybrids, we performed PLA using S9.6 and HDAC1 antibodies and detected a significantly increased number of RNAseH1-sensitive PLA foci following IR treatment (Fig. EV1B).

These results demonstrate that GATAD2B, MBD3, and HDAC1 preferentially bind to DNA:RNA hybrids upon DNA damage.

## The GATAD2B–NuRD complex is recruited to DSBs in transcription and DNA:RNA hybrid-dependent manner

Our data identified components of the NuRD complex as novel DNA damage-specific DNA:RNA hybrid interactors. Since DNA:RNA hybrids are associated with transcription, and transcription at DSBs is required for efficient DNA repair (Yasuhara et al, 2018), we aimed to determine whether the GATAD2B–NuRD complex is recruited to DSBs in a DNA:RNA hybrid-dependent manner. Initially, we employed micro-irradiation and verified that our laser settings indeed led to DNA damage induction, as evidenced by a positive γH2AX immunofluorescence signal in micro-irradiated cells (Fig. EV1C). Subsequently, HeLa cells transfected with GFP-tagged GATAD2B and MBD3 were subjected to laser micro-irradiation, followed by time-lapse microscopy. Both proteins were found to be rapidly recruited to the sites of laser-induced DNA damage within 30 s. Interestingly, when transcription was inhibited by triptolide or DRB, the recruitment of both GATAD2B and MBD3 to DSBs was significantly reduced (Figs. 2A and EV1D). Further, in cells transfected with RNaseH1-mCherry and subjected to laser micro-irradiation, the presence of GATAD2B and MBD3 at DSBs was diminished (Figs. 2B and EV1E). In addition, the kinetics of RNaseH1 and GATAD2B recruitment to DSBs, showed a similar profile, suggesting a correlation in their binding to DSBs (Fig. EV1F).

We then investigated whether the interaction between DNA:RNA hybrids and GATAD2B is specifically regulated as a part of DDR, or if DNA damage causes the retention of NuRD at actively transcribed regions to which it normally binds. We selected three genes known to be bound by the NuRD complex and performed chromatin immunoprecipitation in cells either over-expressing RNaseH1 or treated with transcription inhibitor triptolide. The results indicated that neither transcription inhibition nor RNaseH1 overexpression affected the level of NuRD at these loci under non-damage conditions, suggesting that the NuRD complex is recruited de novo to DSBs in a DNA:RNA hybrid and transcription-dependent manner (Fig. EV1G).

To further support these results, we performed PLA using GATAD2B or MBD3 and γH2AX antibodies and observed an increased number of PLA foci upon IR treatment compared to control cells. The foci resulting from both, GATAD2B/γH2AX and MBD3/γH2AX were sensitive to transcription inhibition (treatments with TPL3 and DRB) and the resolution of DNA:RNA hybrids (overexpression of RNAseH1) (Figs. 2C and EV1H, single antibodies served as PLA-negative controls).

We also aimed to determine whether GATAD2B plays a role in the localization of HDAC1 to DSBs. A PLA using HDAC1 and γH2AX antibodies revealed PLA foci upon IR treatment, which were sensitive to transcription inhibition (with DRB and TPL3), as well as RNAseH1 overexpression. Interestingly, the depletion of GATAD2B also resulted in a reduced number of HDAC1/ γH2AX PLA foci (Figs. 2D and EV2A,B). Supporting this, we conducted HDAC1 ChIP in the DIvA cell line, where an ER-fused AsiSI restriction enzyme can be translocated into the nucleus upon the addition of tamoxifen (4OHT) to cleave DNA at specific sites, thereby producing double-strand breaks. This cellular system is commonly used to study DSBs in a sequence-specific manner (Aymard et al, 2014; Iacovoni et al, 2010). Indeed, we detected a significant HDAC1 ChIP signal at two different AsiSI cut sites, upon damage induction, and the levels of HDAC1 were significantly decreased following GATAD2B depletion (Fig. 2E,F).

KDM5A and ZMYND8 are two histone modifiers that play important roles in the DDR by interacting with PARP1 and recruiting the NuRD complex to DNA damage sites. KDM5A is a histone demethylase that binds to PAR chains through a non-canonical region and represses transcription to facilitate HR repair (Kumbhar et al, 2021). ZMYND8 is a bromodomain (BRD) protein that recognizes acetylated chromatin and mediates a transcription-associated DDR pathway targeting actively transcribed regions. ZMYND8 also regulates PAR levels at DNA breaks by recruiting ARH3 and modulates the recruitment of BRCA1 and RAD51, two key factors for HR repair (Gong et al, 2015; Gong et al, 2017;

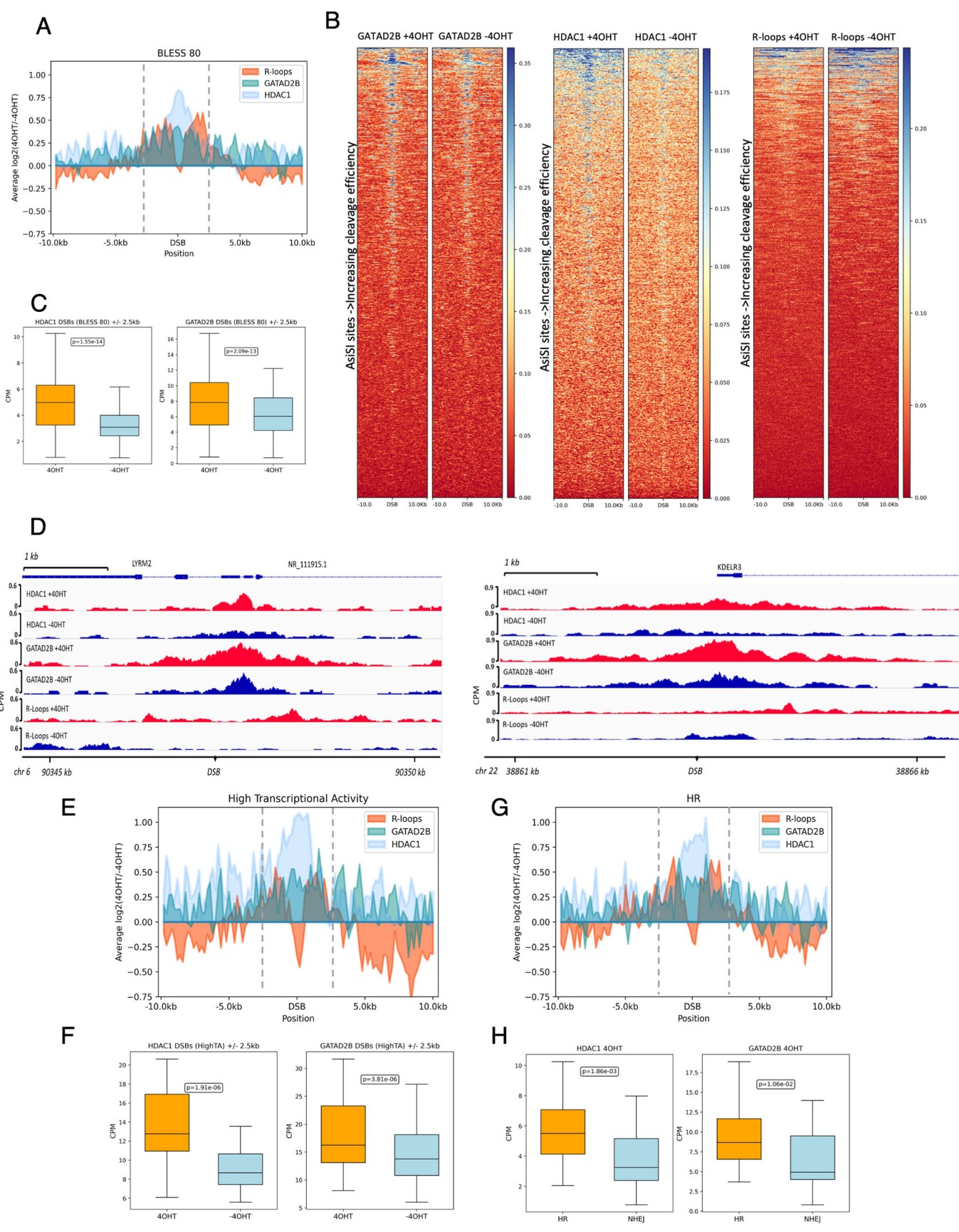

**Figure 3.  GATAD2B and HDAC1 form a boundary around DSBs in transcriptionally active and HR-prone loci.**

(A) Metagene profile showing ChIP-seq enrichment of GATAD2B, HDAC1 and R-loops between 4 h 4OHT-treated and untreated DIvA cells at 80 AsiSI cut sites (as defined by BLESS technique) over a 20 kb window. Values are presented as log2 ratios. (B) Heatmaps showing GATAD2B and HDAC1 ChIP-seq and DRIP-seq read count over a 20-kb window centered on the DSB before (−4OHT) and after (+4OHT) DSB induction. DSBs are sorted according to decreasing cleavage efficiency, as defined by BLESS technique. See text for references. (C) Box plot graph showing levels of HDAC1 and GATAD2B ChIP-seq reads mapping to AsiSI sites over a 5-kb window in the presence or absence of 4OHT. Wilcoxon two-sample test was used to determine statistical significance with $P$ value cut-off set at 0.01. $n = 3$. Box plots consist of the median (central line), the 25th and 75th percentiles (box) and the highest/lowest represented by whiskers. (D) IGV screenshot showing GATAD2B and MBD3 ChIP-Seq and DRIP-seq reads before (−4OHT) and after damage induction (+4OHT) at selected AsiSI site. Values are presented as log2 ratios. (E) Metagene profile showing ChIP-seq enrichment of GATAD2B, HDAC1 and R-loops at AsiSI cut sites between 4 h 4OHT-treated and untreated DIvA cells in highly transcribed regions over a 20 kb window. Values are presented as log2 ratios. (F) Box plot graph showing levels of HDAC1 and GATAD2B ChIP-seq reads mapping to AsiSI sites over a 5 kb window in loci with high transcriptional activity. Wilcoxon two-sample test was used to determine statistical significance with $P$ value cut-off set at 0.01. Boxplots consist of the median (central line), the 25th and 75th percentiles (box) and the highest/lowest represented by whiskers. (G) Metagene profile showing ChIP-seq enrichment of GATAD2B, HDAC1 and R-loops between 4 h 4OHT-treated and untreated DIvA cells at HR-prone AsiSI cut sites over a 20 kb window. Values are presented as log2 ratios. HR- and NHEJ-prone DSBs were defined based upon the RAD51/XRCC4 binding ratio. $n = 3$. (H) Box plot graph showing levels of HDAC1 and GATAD2B ChIP-seq reads over a 5 kb window mapping to AsiSI sites in HR- or NHEJ-prone loci. Wilcoxon two-sample test was used to determine statistical significance with $P$ value cut-off set at 0.01. $n = 3$. Box plots consist of the median (central line), the 25th and 75th percentiles (box) and the highest/lowest represented by whiskers.

Kumbhar et al, 2021; Spruijt et al, 2016). Therefore, we investigated whether these two proteins might be involved in the recruitment of the GATAD2B–NuRD to DSBs. PLA using antibodies against HDAC1 and γH2AX in cells depleted of MBD3, GATAD2B, ZMYND8, and KDM5A, revealed that knockdown (KD) of ZMYND8, as well as KD of GATAD2B and MBD3 KD, significantly reduced the number of PLA foci (Fig. EV2B). To validate the specific recruitment of GATAD2B, MBD3 and HDAC1 to DSBs, we employed PLA using antibodies against γH2AX and HOXD11, a chromatin-associated transcription factor not involved in DDR. We did not detect PLA foci under any condition tested, including control, IR, depletion of GATAD2B or MBD3, transcription inhibition by Ttriptolide or DRB and overexpression of RNAseH1 (Fig. EV2C). Furthermore, we sought to determine whether any of these conditions affect the levels of γH2AX. Immunofluorescence using the γH2AX antibody revealed increased levels of γH2AX upon DNA damage, but these levels were not significantly altered in control, IR treated, depleted of GATAD2B or MBD3, transcription inhibited by triptolide or DRB, or in cells overexpressing RNAseH1 (Fig. EV2D).

The S9.6 antibody is recognized for its ability to bind not only to DNA:RNA hybrids but also to dsRNA (Bou-Nader et al, 2022). In fact, dsRNA at DSBs has been previously reported (Burger et al, 2019; Burger et al, 2017). Therefore, we tested whether GATAD2B could be recruited to DSBs by dsRNA. We performed GATAD2B/γH2AX PLA in control cells, irradiated cells and cells treated with dsRNA-specific RNAse III, and cells depleted of Dicer, an enzyme that cleaves endogenous dsRNA (Burger et al, 2017; White et al, 2014). PARP1 inhibitor was used as a positive control, as we observed it affects the binding of GATAD2B to DNA:RNA hybrids (Fig. 1G). Irradiation led to the formation of GATAD2B/γH2AX PLA foci, whose numbers were unaffected by RNAse III treatment (cleavage of dsRNA) or by the depletion of Dicer (which increased levels of dsRNA). In the same experiment, PARP1 inhibition significantly reduced the number of GATAD2B/γH2AX PLA foci (Fig. EV3A). Furthermore, inhibition of PARP1 also reduced the number of MBD3/γH2AX and HDAC1/γH2AX PLA foci (Fig. EV3B,C). These data suggest that these NuRD subunits are part of the same complex. To investigate whether GATAD2B can bind to other NuRD subunits, we performed GATAD2B/CHD4 PLA and observed a reduced number of foci upon irradiation. In

contrast, GATAD2B/MTA1 PLA showed an increased number of foci upon irradiation (Fig. EV3D,E).

Our results suggest that the GATAD2B–NuRD complex, without CHD4, is recruited to DSBs in a transcription, DNA:RNA hybrid and PARP1-dependent manner and facilitates HDAC1 recruitment to DSBs. ZMYND8 is also required for HDAC1 recruitment to DSBs, likely through its role in transcription-associated DDR and binding to PAR.

## GATAD2B and HDAC1 are associated with transcriptionally active and HR-prone DSBs

To investigate the genome-wide distribution and profile of the GATAD2B–NuRD complex at DSBs, we employed DIvA cell line (Aymard et al, 2014; Iacovoni et al, 2010) and performed chromatin immunoprecipitation with antibodies against GATAD2B and HDAC1, followed by next-generation sequencing (ChIP-seq). The reproducibility of ChIP-seq data across all three replicates was confirmed by Principal Component Analysis (PCA) and show the reproducibility of the data (Appendix Fig. S2A).

DSBs (as cut sites) were identified and mapped by BLESS technique (Clouaire et al, 2018a). ChIP-seq analysis revealed the enrichment of GATAD2B and HDAC1 at DSBs, when a cut was induced by 4OHT treatment (Fig. 3A–D; Appendix Fig. S2B–D; levels of input signals were used as a control, Appendix Fig. S3A,D). The accumulation of both proteins was most evident at DSBs (as defined by BLESS signal), extending over a 2.5-kb window around the AsiSI cutting sites (Fig. 3A–D; Appendix Fig. S2D,E). We then aligned the GATAD2B and HDAC1 ChIP-seq profiles with DRIP-seq data that indicates the presence of DNA:RNA hybrids near DSBs (Cohen et al, 2018). DRIP-seq technique employs S9.6 antibody pull-down followed by DNA sequencing. Due to the cleavage of DNA at breaks, DRIP signals are undetectable exactly at the breakpoints, resulting in a dip in DRIP sequencing reads, unlike the ChIP-seq approach. Overlaying our ChIP-seq and with the DRIP-seq metagene profiles, we found that DNA:RNA hybrids were in the vicinity and partially overlapped with the GATAD2B, HDAC1 peaks. Interestingly, all metagene profiles showed a decline in intensity ~2.5 kb from the breaks (Fig. 3A; Appendix Fig. S2D,E). Heatmap analysis revealed that the recruitment of the GATAD2B–NuRD complex correlated with AsiSI cutting

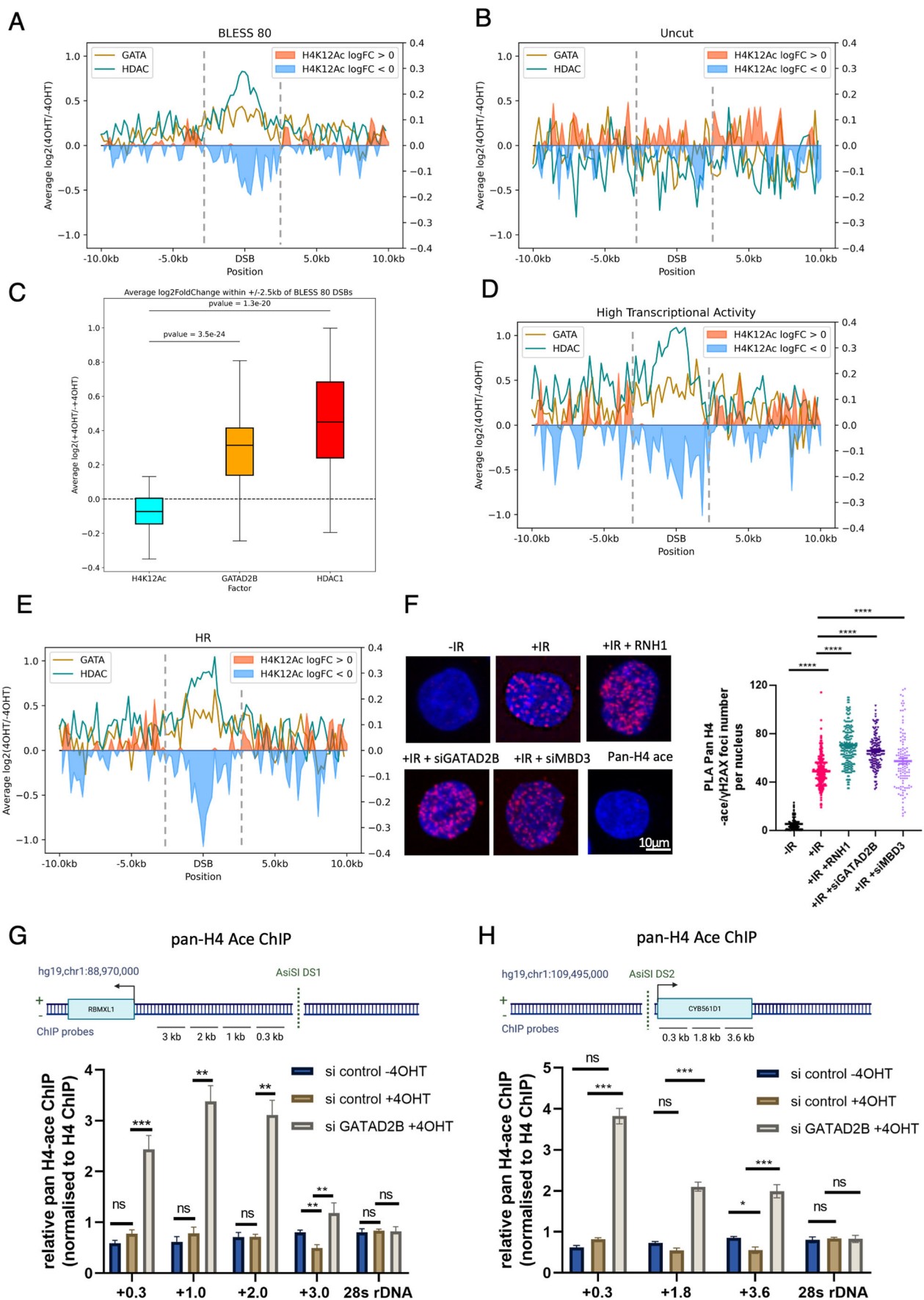

**Figure 4.  GATAD2B–NuRD complex promotes histone deacetylation at DSBs.**

(A) Metagene profile showing ChIP-seq enrichment of GATAD2B, HDAC1 and H4K12ac between 4 h 4OHT-treated and untreated DIvA cells at 80 cut AsiSI sites (as defined by BLESS technique) over a 20-kb window. Values are presented as log$_2$ ratios. (B) Metagene profile showing ChIP-seq enrichment of GATAD2B, HDAC1 and H4K12ac between 4 h 4OHT-treated and untreated DIvA cells at uncut AsiSI sites (as defined by BLESS technique) over a 20 kb window. Values are presented as log$_2$ ratios. (C) Box plot showing average log$_2$FoldChange within 2.5 kb around cut AsiSI for H4K12ac, GATAD2B and HDAC1 ChIP-seq counts. Wilcoxon two-sample test was used to determine statistical significance with $P$ value cut-off set at 0.01. $n = 3$. Box plots consist of the median (central line), the 25th and 75th percentiles (box) and the highest/lowest represented by whiskers. (D) Metagene profile showing ChIP-seq enrichment of GATAD2B, HDAC1 and H4K12ac between 4 h 4OHT-treated and untreated DIvA cells at 80 cut AsiSI sites in highly transcribed regions over a 20 kb window. Values are presented as log$_2$ ratios. (E) Metagene profile showing ChIP-seq enrichment of GATAD2B, HDAC1 and H4K12ac between 4 h 4OHT-treated and untreated DIvA cells at HR-prone cut AsiSI sites over a 20-kb window. Values are presented as log$_2$ ratios. (F) PLA of pan-acetyl H4 and γH2AX in cells with or without IR followed by 20 min of recovery, and overexpression of RNAseH1 or depletion of GATAD2B and MBD3. IR = 5 Gy. Left: representative confocal microscopy images; right: quantification of left, error bar = mean ± SEM, significance was determined using nonparametric Mann–Whitney test. ****$P$ ≤ 0.0001. Scale bar = 10 μm, $n > 3$. (G) Top: drawing showing DS1 genomic region with AsiSI cut site and position of ChIP-qPCR probes. Bottom: Bar chart showing relative pan-acetyl H4 ChIP levels (normalized to H4 ChIP levels) at indicated sites next to AsiSI cut in cells with or without 4OHT and depletion of GATAD2B, error bar = mean ± SEM, significance was determined using nonparametric Mann–Whitney test. ***$P$ ≤ 0.001, **$P$ ≤ 0.01, n.s. not significant. (H) Top: drawing showing DS2 genomic region with AsiSI cut site and position of ChIP-qPCR probes. Bottom: Bar chart showing relative pan-acetyl H4 ChIP levels (normalized to H4 ChIP) at indicated sites next to AsiSI cut in cells with or without 4OHT and depletion of GATAD2B, error bar = mean ± SEM, significance was determined using nonparametric Mann–Whitney test. ***$P$ ≤ 0.001, *$P$ ≤ 0.05 n.s. not significant. Source data are available online for this figure.

efficiency, indicating that the complex preferentially assembles at more accessible AsiSI sites, (Fig. 3B; Appendix Fig. S3B). Indeed, GATAD2B and HDAC1, were more enriched at transcription active loci compared to transcriptionally repressed regions (Fig. 3E,F; Appendix Fig. S4A–C). Previous reports have demonstrated that transcription and DNA:RNA hybrids may act as a scaffold for the recruitment of HR repair pathway proteins (Aymard et al, 2014). AsiSI cutting sites in the DIvA system can be classified as HR or NHEJ-prone DSBs based on a correlation ratio using ChIP-Seq data coverage of RAD51, a well-known facilitator of HR, (AsiSI site +/−4 kb) relative to XRCC4 coverage, a known NHEJ factor (AsiSI site +/−1 kb). The top 30 sites were annotated as HR-prone due to higher RAD51 coverage, and the bottom 30 as NHEJ-prone due to a higher XRCC4 ratio (Clouaire et al, 2018a). Our ChIP-seq data indicated that GATAD2B and HDAC1 are significantly more associated with HR-prone AsiSI cut sites (Fig. 3G,H; Appendix Fig. S4D–F).

In summary, our genome-wide data indicate that the GATAD2B–NuRD complex is predominantly localized at and surrounded by DNA:RNA hybrids near DSBs, suggesting that these hybrids provide a platform for NuRD recruitment to DSBs. The GATAD2B–NuRD complex is more enriched at HR-prone DSBs and within regions of higher transcription activity.

## The GATAD2B–NuRD complex promotes histone deacetylation around DSBs

Our findings demonstrate that the binding of GATAD2B to DNA:RNA hybrids is required for the localization of the NuRD complex at DSBs. Given that the NuRD complex includes the histone deacetylase HDAC1, we examined whether its activity inversely correlates with histone acetylation levels at DSBs. Acetylation of the lysine residue on histone H4 is associated with transcriptionally active genomic regions. Interestingly, the ChIP-seq profile of acetylated H4K12 (H4K12Ac) in DIvA cells indicates the presence of this modification at uncut AsiSI sites, with levels decreasing following cut induction (Appendix Fig. S5A). The alignment of GATAD2B and HDAC1 ChIP-seq data with the H4K12Ac ChIP-seq profile (Clouaire et al, 2018a) revealed a negative correlation between H4K12Ac levels and the presence of GATAD2B or HDAC1 around DSBs (Fig. 4A–C; Appendix

Figs. S5A and S6A–C). These opposite profiles were more pronounced at sites of high transcription activity and HR-prone DSBs (Fig. 4D,E; Appendix Fig. S5B,C).

We then tested whether the levels of histone acetylation are affected in the absence of the GATAD2B–NuRD complex. Using a PLA with antibodies against γH2AX and pan-acetyl H3 or pan-acetyl H4, we discovered that the depletion of GATAD2B or MBD3 as well as overexpression of RNaseH1, significantly increased the levels of acetylated H3 or H4 following DNA damage (Figs. 4F and EV4A). Confirming this, our ChIP-qPCR analysis in DIvA U2OS cells indicated that the lack of GATAD2B results in increased levels of H4 acetylation around two DSBs (Figs. 4G,H and EV4B,C). In addition, Western blot analysis showed that depletion of GATAD2B does not alter the overall levels of H4 acetylation (Fig. EV4D).

In conclusion, the GATAD2B–NuRD complex appears to play a pivotal role in promoting histone deacetylation, thereby forming a chromatin boundary around DSBs.

## The GATAD2B–NuRD complex boundary prevents chromatin hyperrelaxation

The NuRD complex has chromatin-remodeling and deacetylation activities, that can facilitate chromatin condensation. Additionally, transcription termination is associated with repressive histone modifications and the recruitment of heterochromatin factors. To investigate whether DNA:RNA hybrids and the GATAD2B–NuRD complex influence chromatin structure following DNA damage, we employed micrococcal nuclease (MNase) sensitivity assays. MNase is an endo-exonuclease that preferentially cleaves accessible DNA linkers between nucleosomes. MNase digestion of chromatin leads to a ladder of DNA fragments that can be visualized by agarose gel electrophoresis. Each band in the pattern corresponds to DNA protected by units of nucleosomes. The fastest migrating fragment (~147 bp) represents mono-nucleosomal DNA, and larger multiples of this length indicate further nucleosomal protection (Voong et al, 2017). More open chromatin, therefore, results in a higher number of mono-nucleosomes post MNase digestion. After exposing cells to ionizing radiation (IR), we found that within 10 min, the chromatin had become more relaxed, as evidenced by an increased number of mono-nucleosomes. Interestingly,

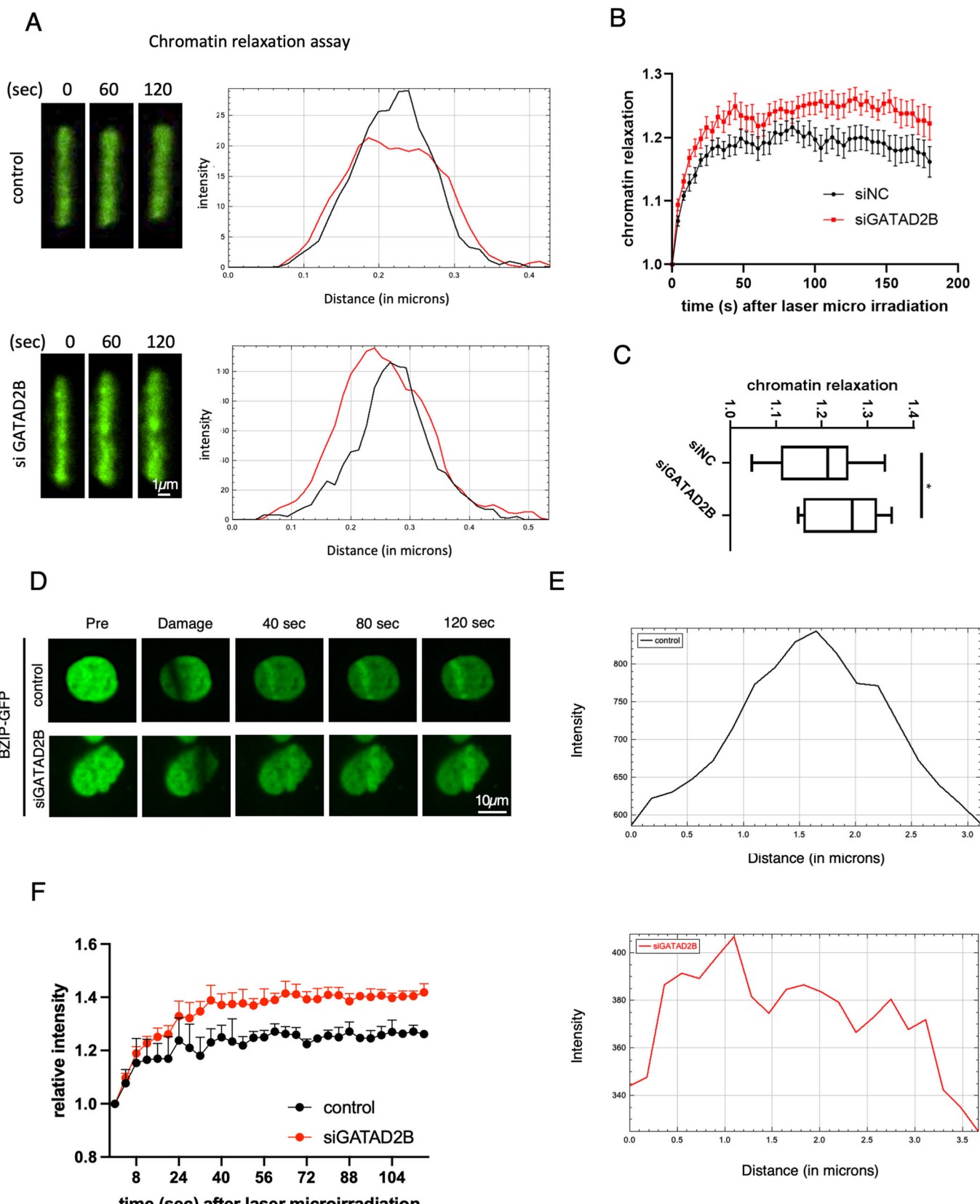

◄

**Figure 5.  GATAD2B–NuRD complex facilitates chromatin condensation near DSBs.**

(A) Left: Representative confocal images from three independent replicates of the photoconverted chromatin region at 0 s, 60 s and 120 s after irradiation at 405 nm in U2OS WT or GATAD2B KD cells. Right: Intensity profiles showing thickness of the irradiated lines at 0 s (black) and 120 s (red) after damage induction. Scale bar = 1 μm. (B) Graph showing kinetics of chromatin relaxation as measured by the thickness of the photoconverted chromatin region after irradiation in wt U2OS and GATAD2B KD cells after DNA damage induction. At least ten cells were used for quantification at indicated time points, error bar = mean ± SEM. (C) Box plot showing the levels of chromatin relaxation as measured by the thickness and width of the photoconverted chromatin region at 120 s after laser irradiation in wt and GATAD2B KD cells from (B). Significance was determined by Student *t* test: *P ≤ 0.05, *n* > 10. Box plots consist of the median (central line), the 25th and 75th percentiles (box) and the highest/lowest represented by whiskers. (D) Representative confocal images from three independent replicates showing recruitment of DNA-binding domain of BZIP from C/EBPa tagged with GFP to the sites of DNA damage induced by laser micro-irradiation at indicated time points in wt (control) and siGATAD2B cells. Scale bar = 10 μm, *n* > 3. (E) Representative plot profile showing BZIP-GFP signals across laser stripe from (D), measured in micrones. (F) Graph showing kinetics of BZIP-GFP recruitment to the sites of DNA damage induced by laser micro-irradiation at indicated time points in wt (control) and siGATAD2B cells, *n* > 10, error bar = mean ± SEM. Source data are available online for this figure.

RNaseH1 overexpression significantly increased MNase accessibility and DNA digestion following DNA damage (Fig. EV5A), suggesting that the removal of DNA:RNA hybrids may lead to increased chromatin relaxation during the early stages of DDR. In addition, in cells depleted of GATAD2B, we observed a significant increase in mono-nucleosomes, compared to control cells following DNA damage (Fig. EV5B). Interestingly, inhibition of transcription led to chromatin relaxation, but this effect did not enhance chromatin relaxation observed in GATAD2B depleted cells, suggesting that the lack of transcription and consequently lack of DNA:RNA hybrids, impair GATAD2B recruitment, resulting in chromatin relaxation (Fig. EV5C). Contrarily, depletion of CHD4 inhibited chromatin relaxation, which is consistent with previous literature (Fig. EV5D). To further validate the role of GATAD2B in chromatin remodeling upon DNA damage in vivo, we transfected U2OS cells with plasmids encoding histone H2B fused to photoactivable GFP. After laser micro-irradiation, we measured the thickness of GFP stripes as an indicator of chromatin relaxation (Smith et al, 2018). Depletion of GATAD2B significantly increased chromatin relaxation after laser micro-irradiation (Fig. 5A–C), suggesting that the GATAD2B–NuRD complex is necessary for chromatin condensation at DSBs.

To further examine the chromatin state at DSBs, we employed the BZIP DNA domain from the C/EBPa transcription factor. As C/EBPa has no established role in DDR, its BZIP domain is not anticipated to localize to DSBs for the purpose of repair. Instead, it has been used as an indicator for open/relaxed/euchromatin (Smith et al, 2019). After transiently transfecting BZIP-GFP and applying laser micro-irradiation, we monitored BZIP-GFP recruitment to DSBs in both control and siGATAD2B-treated cells. The results showed that depletion of GATAD2B leads to more relaxed chromatin at DSBs, as measured by increased intensity and breadth of BZIP-GFP stripes (Fig. 5D–F).

In conclusion, our results suggest that the GATAD2B–NuRD complex plays a critical role in limiting chromatin relaxation around DSBs.

## The GATAD2B–NuRD complex facilitates the termination of DNA end resection and promotes HR

We show that the GATAD2B–NuRD complex is associated with HR-prone DSBs. The initiation of HR is contingent upon the resection process, which generates single-stranded DNA (ssDNA) overhangs vital for D-loop formation and homology searching within the sister chromatid. In *fission yeast*, maintaining a balance

of DNA:RNA hybrids has been linked to proper termination of resection (Ohle et al, 2016). In addition, we demonstrate that the GATAD2B–NuRD complex promotes histone deacetylation and chromatin condensation. This led us to investigate whether DNA:RNA hybrids and the GATAD2B–NuRD complex also contribute to the termination of DNA end resection. Replication protein A (RPA), a ssDNA-binding protein, rapidly responds to ssDNA overhangs produced during resection. We evaluated the impact of transcription inhibition, overexpression of RNAseH1, and depletion of GATAD2B or MBD3 on phosphorylated RPA levels following ionizing radiation (IR). Our results show that inhibiting transcription, overexpressing RNaseH1, or depleting GATAD2B or MBD3 significantly increases the number of phosphorylated S4/S8 RPA32 foci (Fig. 6A). Conversely, depletion of BRCA1 results in a decreased number of pS4/S8 RPA32 foci (Fig. 6A), while CHD4 depletion results in an opposite trend, with reduced number of foci (Fig. EV5E).

Further, we observed that extended resection is indicated by an increase in the size of the phosphorylated S4/S8 RPA32 foci. Using FIJI software for foci size analysis, we determined that transcription inhibition, RNaseH1 overexpression, or GATAD2B or MBD3 depletion leads to a greater number of enlarged foci, defined by a prominence threshold of 2500 in the Find Maxima tool (Fig. EV5F).

BrdU intercalation into longer ssDNA can be detected with anti-BrdU antibodies. We employed immunofluorescence assay and showed that transcription inhibition, overexpression of RNaseH1 or depletion of GATAD2B or MBD3 enhances BrdU signal upon 5 Gy IR treatment followed by a 2-hour recovery period, but not in no damage condition. Depletion of BRCA1 upon IR treatment resulted in decreased number of BrDU foci (Figs. 6B and EV5G).

Next, we used existing RAD51 ChIP-seq data (Aymard et al, 2014) alongside our GATAD2B and HDAC1 ChIP-seq data. The metagene profiles showed a distinct co-localization of these three proteins around DSBs (Fig. 6C; Appendix Fig. S7A–K). In addition, by creating a heatmap of AsiSI sites arranged by RAD51 ChIP-seq coverage within 2.5 kb surrounding the DSBs, we observed that the GATAD2B and HDAC1 ChIP-seq coverage corresponds proportionally with the RAD51 coverage (Fig. 6D).

To directly measure DNA end resection, we employed an endonucleolytic cleavage assay (Zhou et al, 2014). Probes positioned before and after the GATAD2B boundary at two selected DSBs, revealed that GATAD2B depletion significantly extended DNA end resection beyond the chromatin boundary (Fig. 6E).

Extended resection produces longer single-stranded DNA (ssDNA) overhangs, which could trigger alternative DSB repair

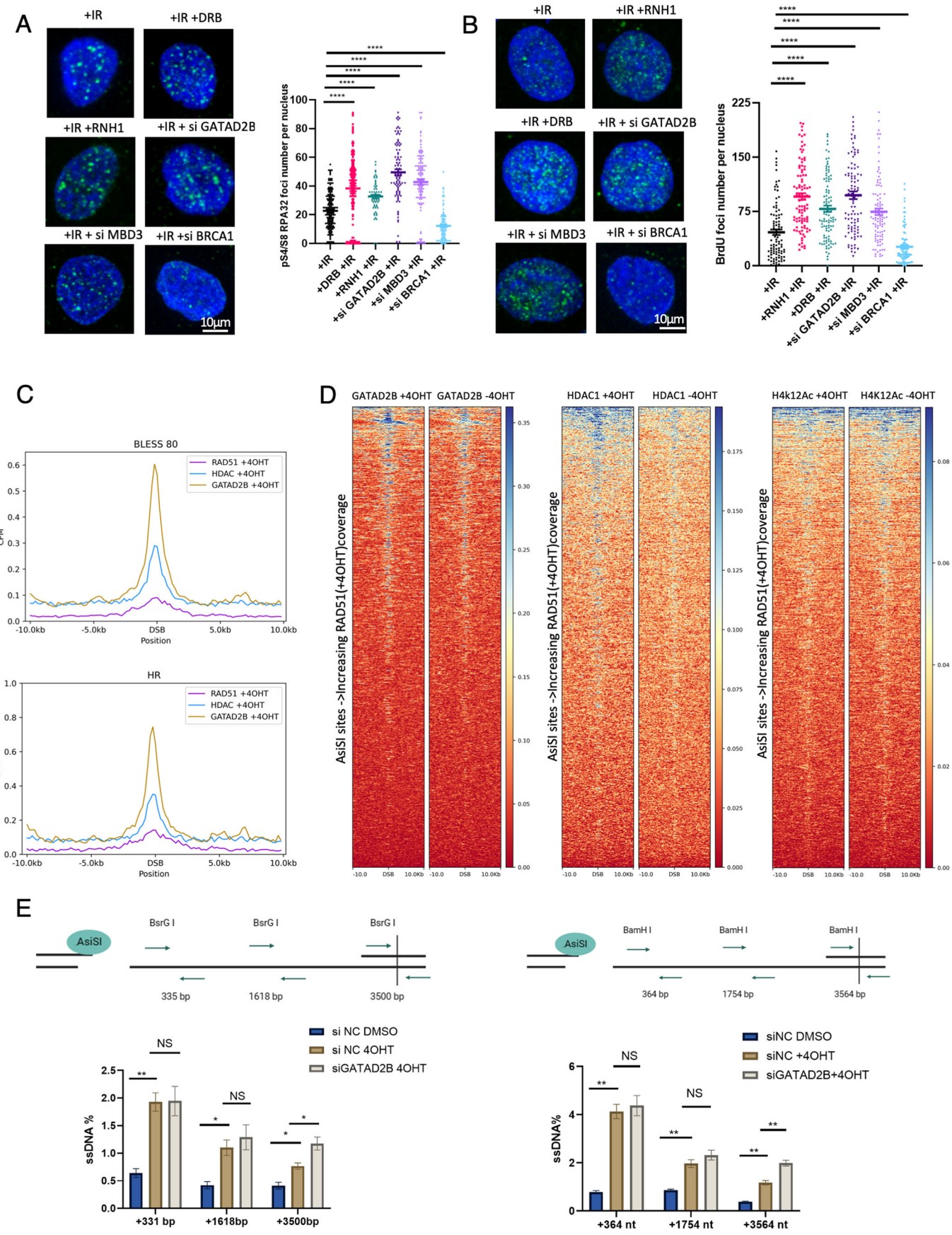

◄

**Figure 6.   GATAD2B–NuRD complex facilitates termination of the end resection.**

(A) Left: representative confocal images from three independent replicates showing immunofluorescence signals of phospho-S4/S8 RPA32 in cells treated with 5 Gy IR followed by 2 h recovery with overexpression of RNAseH1 or transcription inhibition (TLP3 or DRB) or depletion of GATAD2B and MBD3. Right: quantification of left, error bar = mean ± SEM, significance was determined using nonparametric Mann–Whitney test. ****$P \le 0.0001$, Scale bar = 10 μm, $n > 50$ cells from three biological repeats. (B) Left: representative confocal images from three independent replicates showing immunofluorescence signals of BrdU in cells treated with 5 Gy IR followed by 2 h recovery with transcription inhibition (TLP3 or DRB), overexpression of RNAseH1 or depletion of GATAD2B and MBD3. Right: quantification of left, error bar = mean ± SEM, significance was determined using nonparametric Mann–Whitney test. ****$P \le 0.0001$. Scale bar = 10 μm, $n > 50$ cells from three biological repeats. (C) Metagene profile showing ChIP-seq enrichment of GATAD2B, HDAC1 and RAD51 between 4 h 4OHT-treated and untreated DIvA cells at 80 cut AsiSI (as defined by BLESS technique) and at HR-prone cut AsiSI sites over a 20 kb window. Values are presented as log2 ratios. (D) Heatmaps showing GATAD2B, HDAC1 and H4K12Ac ChIP-seq and DRIP-seq read count over a 20-kb window centered on the DSB before (−4OHT) and after (+4OHT) DSB induction. DSBs are sorted according to decreasing coverage. See text for references. (E) The illustration explains a quantitative DNA resection assay. When U2OS cells are treated with 4OHT, DNA damage is induced by AsiSI cleavage. Subsequently, genomic DNA is collected and subjected to restriction digestion by BsrGI (indicated by vertical lines). Quantitative PCR (qPCR) primers are shown as green arrows. Amplification is possible only at sites that underwent resection before the digestion. Top: Diagram illustrating quantitative DNA resection assay. Site-specific DSBs were induced by an endonuclease AsiSI in DIvA cells after induction of 4OHT. Genomic DNA were then harvested and digested by the endonuclease BsrGI, with the distance of each recognition site to the AsiSI cut site was labeled. qPCR probes (black arrows) are designed at each side of the restriction sites. Only loci that had been resected prior to digest can be amplified. Bottom: Bar chart showing relative ssDNA levels from three independent replicates at each BsrGI or BamHI recognition loci in cells with or without 4 h 4OHT treatment and depletion of GATAD2B, NC indicates negative control, error bar = mean ± SEM, **$P \le 0.01$, *$P \le 0.05$, NS no significant. $n = 3$ biological replicates. Image was created with BioRender.com. Source data are available online for this figure.

mechanisms such as single-strand annealing (SSA) and micro-homology mediated end-joining (MMEJ), potentially resulting in genetic information loss (Zhao et al, 2020). Excessive DNA end resection may also increase ssDNA exposure, leading to replication fork collapse, HR repair deficiency and genome instability. Consequently, DNA end resection must be terminated at the point, when length of ssDNA overhang is sufficient to initiate HR repair (Chen et al, 2013). We assessed whether the GATAD2B–NuRD complex or DNA:RNA hybrids could inhibit the activation of SSA or MMEJ pathways by inducing site-specific DSBs in the 28 s rDNA region with the IPpoI-ER restriction enzyme (Berkovich et al, 2007). Activation of SSA or MMEJ repair pathways would result in loss of genetic information and reduced copy number of 28 s rDNA. Indeed, RNaseH1 overexpression or GATAD2B depletion reduced of 28 s rDNA copy number upon DSBs induction by IPpoI (Fig. 7A), indicating that the GATAD2B–NuRD complex can facilitate the termination of DNA end resection and prevent loss of genetic information.

Next, we investigated whether the GATAD2B–NuRD complex is required for effective DNA repair. We utilized U2OS DR-GFP and EJ5-GFP cell lines, which contain HR and NHEJ reporter cassettes, respectively (D'Alessandro et al, 2018). If DNA damage is induced by endonuclease cleavage; the repaired cells will generate GFP signal which can be measured and quantified by FACS. We observed a reduction in HR with GATAD2B and MBD3 depletion. ATRi and DNA-PKi served as positive controls (Fig. 7B). Furthermore, we used the HR reporter system in cells depleted of RNA helicases such as XRN2 and Senataxin, as well as RNaseH1 and detected decreased HR efficiency (Appendix Fig. S8A). More-over, using the HR reporter system in cells depleted of RNA helicases such as XRN2 and Senataxin, as well as RNaseH1, we noted a decreased HR efficiency (Appendix Fig. S8A). While GATAD2B depletion did not appear to affect NHEJ efficiency as gauged by the reporter system (Appendix Fig. S8B), this might be obscured by concurrent MMEJ repairs. Thus, we examined the number of 53BP1 foci, a marker of NHEJ, and found a significant decrease in foci in cells with depleted GATAD2B (Appendix Fig. S8C). These results suggest a role for GATAD2B in DSB repair pathways, with a preference for HR. Considering that HR primarily occurs during the S/G2 phase of the cell cycle, we analyzed the cell

cycle profiles of GATAD2B and MBD3 knockdown cells via FACS, detecting no significant alterations compared to control cells (Appendix Fig. S8D).

These data suggest that the levels of DNA:RNA hybrids around DSBs have to be tightly regulated, which is in agreement with previous studies (Ohle et al, 2016). To extend these data further, we monitored the clearance of γH2AX in cells lacking GATAD2B and MBD3 or overexpressing RNAseH1 and found delayed DNA repair as indicated by persistent levels of γH2AX (Appendix Fig. S8E,F). Finally, we employed a comet assay for a direct assessment of DNA repair efficiency. Consistently, depletion of GATAD2B and MBD3 resulted in significant accumulation of DNA tails corresponding to unrepaired damaged DNA (Fig. 7C).

Overall, our data demonstrate that the chromatin boundary, formed by the GATAD2B–NuRD complex, prevents DNA end hyper-resection and promotes efficient DSB repair.

## Discussion

The DNA damage response involves chromatin relaxation, enabling the recruitment of DDR factors to the site of DNA lesions through DSB-associated transcription (Vitor et al, 2019). However, the vicinity of DSBs also exhibits transcriptional repression and chromatin condensation, indicating a need for coordinated action between open and condensed chromatin landscapes to ensure effective DSB repair. Histone acetylation has been identified as a key regulator of chromatin structure in response to DNA damage, with various histone acetyltransferases and deacetylases observed at DSB sites (Legube and Trouche, 2003). The formation of a boundary between open and condensed chromatin appears to be essential for this coordination. We demonstrate that DNA:RNA hybrid formation at DSBs enhances the recruitment of the GATAD2B–NuRD complex, which aids in HDAC1-mediated histone deacetylation during the early stages of DDR. This action establishes a spatio-temporal boundary between relaxed and condensed chromatin, thereby promoting the termination of DNA end resection and efficient HR.

Chromatin remodeling is an important step in DDR. The relaxation of chromatin is required to allow DDR factors access to

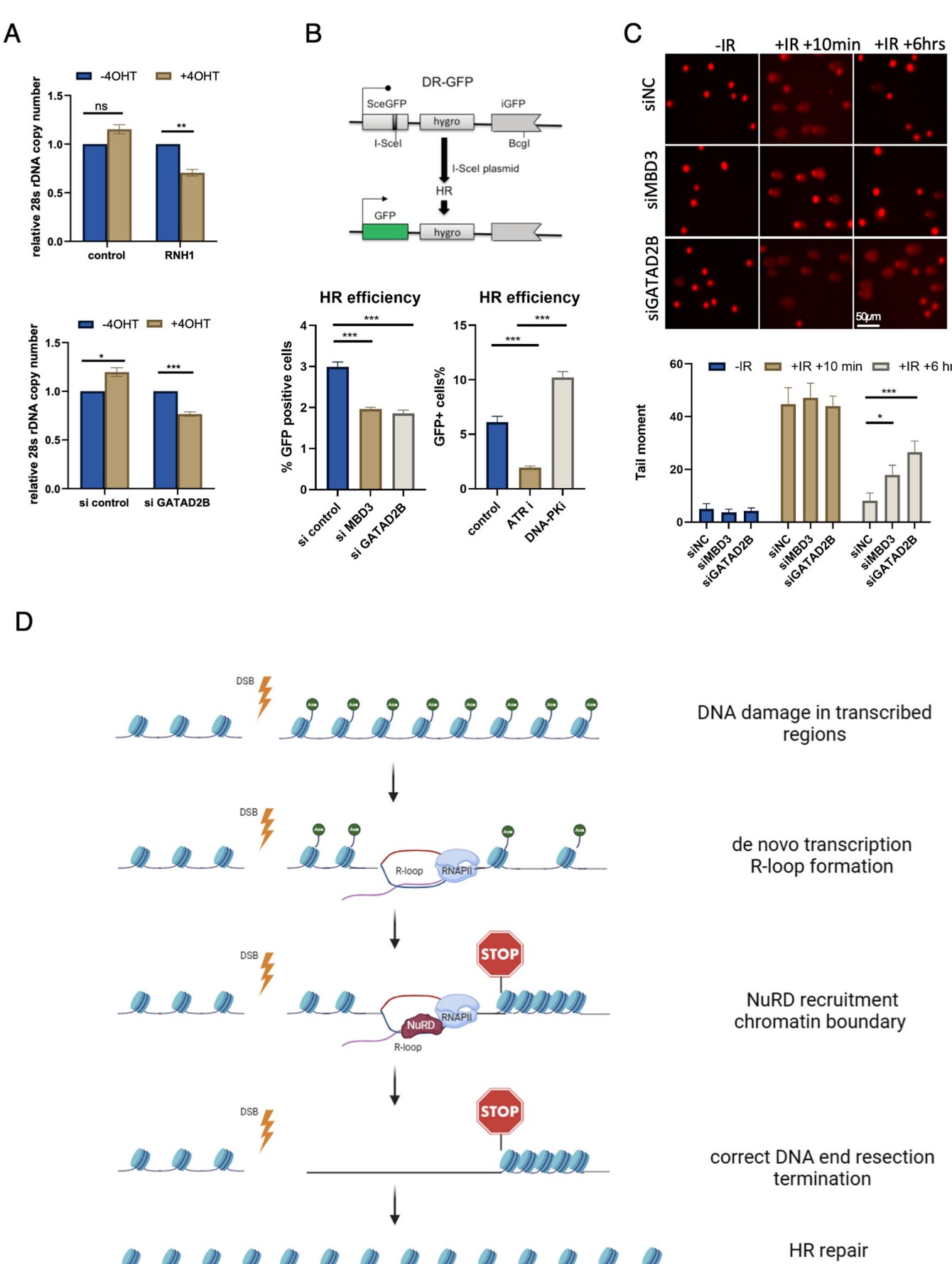

**Figure 7.   Depletion of GATAD2B–NuRD complex leads to impaired DNA damage response.**

(A) Bar charts showing change of relative 28 s rDNA copy number upon DMSO or 4OHT induction in cells expressing IPpOI normalized to GAPDH from three independent replicates, and overexpression of RNaseH (top) or knockdown of GATAD2B (bottom), ***$P \leq 0.001$, **$P \leq 0.01$, *$P \leq 0.05$, $n = 3$, error bar = mean ± SEM. (B) Top: Drawing indicating the structure of HR reporter cassette. Bottom: bar charts showing HR repair efficiency under DSB inducted by I-SceI in U2OS cells containing HR reporter, and knockdown of MBD3 and GATAD2B, inhibition of DNA-PK and ATR from three independent replicates. ***$P \leq 0.001$, $n = 3$, error bar = mean ± SEM. (C) Top: representative microscopy images from three independent replicates showing comet assay of control (siNC), or GATAD2B and MBD3 knockdown cells without IR, or irradiated with 5 Gy with 10 min recovery and 6 h recovery. Bottom: quantification of top, error bar = mean ± SEM, significance was determined using nonparametric Mann–Whitney test. ***$P \leq 0.001$, *$P \leq 0.05$, $n > 50$. (D) The model of chromatin boundary around DSBs established by GATAD2B–NuRD complex and R-loops. GATAD2B–NuRD complex associates with DSBs in a transcription and R-loop-dependent manner, creating a boundary and facilitating histone deacetylation and chromatin condensation around DSBs. Its presence near damaged loci is required for controlled end resection termination. The lack of the GATAD2B–NuRD complex leads to chromatin hyperrelaxation and extended DNA end resection, causing failure in HR repair. Image was created with BioRender.com. Source data are available online for this figure.

broken DNA ends (Murr et al, 2006; Price and D'Andrea, 2013). Heterochromatin markers such as H3K9me3 and proteins involved in chromatin condensation like HP1 and G9a were also found near DSBs (Ayrapetov et al, 2014; Kalousi et al, 2015; Yang et al, 2017). Interestingly, in non-damaged conditions, HP1 and G9a are recruited to transcription termination sites to initiate heterochromatin formation following H3K9me2 deposition induced by R-loops (Skourti-Stathaki et al, 2014). Moreover, R-loops have been implicated in the regulation of both chromatin relaxation and condensation. A fine balance of proteins and histone modifications associated with both heterochromatin and euchromatin has been observed at sequence-independent heterochromatin borders (Kimura and Horikoshi, 2004; Wang et al, 2014). It is therefore conceivable that DNA:RNA hybrids/R-loops generated at DSBs serve as boundary elements, stimulating the recruitment of the GATAD2B–NuRD complex to instigate histone deacetylation. This deacetylation might then expose lysine residues on histones for further modifications, such as methylation by G9a. This methylation could, in turn, lead to HP1 binding, which recruits SUV39H to methylate adjacent histones, thereby establishing and propagating the heterochromatin structure.

The presence of GATAD2B and HDAC1 at DSBs, which is contingent on DNA:RNA hybrid formation, correlates inversely with acetylation at histone 4 lysine 12 (H4K12ac) in the vicinity of DSBs (Clouaire et al, 2018a). This suggests that H4K12ac could be deacetylated by the NuRD complex during DNA damage response. Given that H4K12Ac is typically associated with transcriptionally active regions and that RNA polymerase II (RNAPII) pausing and *de novo* transcription at DSBs require termination (Pankotai et al, 2012), it is plausible that deacetylation of H4K12 by the NuRD complex facilitates the cessation of DSB-associated transcription.

The structural architecture of the NuRD complex encompasses two catalytic activities: ATP-dependent chromatin-remodeling enzymes, including CHD3, CHD4, and CHD5 (Allen et al, 2013), and histone deacetylases HDAC1 and HDAC2. Supplementary to these catalytic functionalities, the NuRD complex integrates methyl-CpG-binding domain MBD2 or MBD3, GATAD2A or GATAD2B, MTA1, MTA2, and MTA3, in addition to RBBP4 and RBBP7 and CDK2AP1 (Allen et al, 2013). Various specific NuRD subunits are rapidly recruited to loci of DNA damage (Chou et al, 2010; Larsen et al, 2010; Polo et al, 2010; Smeenk et al, 2010). GATAD2A and GATAD2B are two proteins that belong to the GATA zinc finger domain-containing family. These subunits define specific and mutually exclusive NuRD subcomplexes that play a crucial role in regulating the chromatin environment surrounding DSBs (Spruijt et al, 2016). ZMYND8 was found to be rapidly

recruited to DSBs, facilitating the efficient recruitment of GATAD2A and CHD4 but not GATAD2B, suggesting a sequential recruitment model for NuRD subunits.

Another member of the NuRD complex, CHD4, has been found at DSBs (Gong et al, 2015) to promote chromatin relaxation following DNA damage, through mechanisms that may or may not involve the rest of the NuRD complex (Hou et al, 2020; Smith et al, 2018). It is plausible that CHD4 (whether alone or as a component of specific NuRD complex) initially plays a vital role in facilitating the formation of an appropriate chromatin landscape necessary for transcriptional activation and DNA:RNA hybrid formation. Consequently, these hybrids could serve as a binding platform for GATAD2B–NuRD, which in turn promotes chromatin condensation and formation of a boundary, associated with negative acetylation histone profile around DSBs.

One crucial aspect of HR is DNA end resection, which benefits from a relaxed chromatin state (Costelloe et al, 2012; Pai et al, 2014). However, excessive end resection can produce long single-stranded DNA (ssDNA) overhangs triggering the activation of single-strand annealing (SSA) and microhomology-mediated end-joining (MMEJ) pathways (Zhao et al, 2020). These pathways can lead to significant loss of DNA, particularly in repetitive regions. Thus, limiting end resection to approximately 2 kilobases (kb) from DSBs is vital for accurate HR repair (Cohen et al, 2018; Mimitou et al, 2017). Our findings indicate that a chromatin boundary formed by the GATAD2B–NuRD complex is pivotal in maintaining the appropriate extent of end resection, safeguarding against the excessive loss of genetic information and ensuring accurate DNA repair.

In summary, our research elucidates the role of the GATAD2B–NuRD complex and DNA:RNA hybrids in delineating a spatial-temporal boundary between open and condensed chromatin, a finding that extends our comprehension of DNA damage response (DDR) regulation (Fig. 7D).

# Methods

## Tissue culture

All human cell lines applied in this study (HEK293T, HeLa, U2OS, DR-U2OS, EJ5-U2OS, AsiSI-ER) were cultured in DMEM media with 10–20% fetal bovine serum, L-glutamine and penicillin/ streptomycin. All human tissues were cultured under the condition 37 °C, 5% $CO_2$.

**Reagents and tools table**

| Reagent/resource | Reference or source | Identifier or catalog number |
|---|---|---|
| **Experimental models** | | |
| HeLa | ATCC | CCL-2 |
| U-2 OS cells | ATCC | HTB-96 |
| HEK293T cells | ATCC | CRL-3216 |
| U-2 OS DR-GFP cells | A gift from Xingzhi Xu | N/A |
| U-2 OS EJ5-GFP cells | A gift from Xingzhi Xu | N/A |
| AsiSI-ER UU2OS | A gift from Gaelle Legube | N/A |
| **Chemicals and reagents** | | |
| Triptolide | Cayman Chemical | CAY11973 |
| DRB | Cayman | 10010302 |
| Olaparib | Cayman | CAY-10621 |
| 4OHT | Cayman | CAY-14854 |
| 5-Bromo-2'-deoxyuridine (5-BrDU) | Sigma | B-5002 |
| RNaseH | New England Biolabs | M0297S |
| Lipofectamine 3000 | Thermo Fisher | L3000008 |
| Lipofectamine RNAiMAX | Thermo Fisher | 13778075 |
| Hoechst 33342 | Thermo Fisher | R37605 |
| Dynabeads Protein A | Thermo Fisher | 10002D |
| Micrococcal Nuclease | New England Biolabs | M0247S |
| Dynabeads Protein G | Thermo Fisher | 10004D |
| AZD5305 | Cayman | 2589531-76-8 |
| Ku-55933 | Cayman | 587871-26-9 |
| AZ20 | Selleckchem | S7050 |
| Duolink In Situ Red Starter Kit Mouse/Rabbit | Sigma | DUO92101-1KT |
| CometAssay Electrophoresis Starter Kit | Bio-Techne | 4250-050-ESK |
| **Antibodies** | | |
| Rabbit-Anti-GATAD2B | Abcam | Ab76925 |
| Rabbit-Anti-MBD3 | Abcam | Ab157464 |
| Mouse Anti-HDAC1 | Santa Cruz | Sc-81598 |
| Anti-phospho-Histone H2A.X (Ser139), clone JBW301 | Merck Millipore | 05-636 |
| Anti-Phospho-Histone H2A.X (Ser139) (20E3) Rabbit mAb | Cell Signaling Technology | 9718 S |
| Anti-DNA-RNA Hybrid Antibody, clone S9.6, mouse monoclonal | Merck | MABE1095 |
| Anti-DNA-RNA hybrid antibody, clone S9.6, rabbit monoclonal | Absolute Antibody | Ab01137-23.0 |
| Histone H3ac (pan-acetyl) antibody, Rabbit polyclonal | Active Motif | 39140 |
| Histone H4ac (pan-acetyl) antibody, Rabbit polyclonal | Active Motif | 39043 |
| Anti-Histone H3 | Abcam | Ab1791 |
| Anti-Histone H4 Polyclonal antibody | Proteintech | 16047-1-AP |
| Anti-beta Tubulin antibody | Abcam | Ab6046 |
| Anti-RNA polymerase II CTD repeat YSPTSPS antibody | Abcam | Ab26721 |
| Anti-RNA polymerase II CTD repeat YSPTSPS (phospho S2) antibody | Abcam | Ab5095 |
| Anti- XRN2 Antibody (H3) | Santa Cruz | sc-365258 |
| Anti-BrdU antibody, Mouse monoclonal | Sigma | B-8434 |
| Anti-Rad51 Antibody | Abcam | Ab63801 |

| Reagent/resource | Reference or source | Identifier or catalog number |
|---|---|---|
| Anti-V5 tag antibody [SV5-P-K] | Abcam | Ab206566 |
| Donkey Anti-Rabbit IgG (H + L) Highly Cross-Adsorbed Secondary Antibody, Alexa Fluor™ 488 | Thermo Fisher | A-21206 |
| Donkey anti-Mouse IgG (H + L) Highly Cross-Adsorbed Secondary Antibody, Alexa Fluor™ 555 | Thermo Fisher | A-31570 |
| Rabbit-anti-Phospho RPA32 (S4/S8) Antibody | Bethyl Laboratories | A300-245A |
| Anti-RNA Helicase A (DHX9) | Abcam | ab26271 |
| Mouse anti-CHD4 | Abcam | Ab70469 |
| Moust anti-MTA1 | Santa Cruz | Sc-133138 |
| Mouse Anti-53BP1 | BD Biosciences | AB_399824 |
| **Recombinant DNA** | | |
| ppyCAG_RNaseH1_WT | Addgene | #111905 |
| pCMV-mCherry-RNaseH1 | This Study | N/A |
| pCMV-GFP-MBD3 | This Study | N/A |
| pCMV-GFP-GATAD2B | This Study | N/A |
| pH2B-PAGFP | (8) | N/A |
| peGFP-BZIP | | N/A |
| **Primers** | | |
| 28s-F | TTCCCTCCGAAGTTTCCCTC | |
| 28s-R | ACTAGGCACTCGCATTCCAC | |
| DS2 + 0.3-F | CCAGCAGTAAAGGGGAGACAGA | |
| DS2 + 0.3-R | CTGTTCAATCGTCTGCCCTTC | |
| DS2 + 1.8-F | GAAGCCATCCTACTCTTCTCACCT | |
| DS2 + 1.8-R | GCTGGAGATGATGAAGCCCA | |
| DS2 + 3.6-F | GCCCAGCTAAGATCTTCCTTCA | |
| DS2 + 3.6-R | CTCCTTTGCCCTGAGAAGTGA | |
| DS1 + 0.3-F | AGGACTGGTTTGCAAGGATG | |
| DS1 + 0.3-R | ACCCCCATCTCAAATGACAA | |
| DS1 + 1.0-F | AGGAATTGACTGCGGTGTTC | |
| DS1 + 1.0-R | GGGGAGGAGGAAAGGTGTAG | |
| DS1 + 2.0-F | GCCATAACAGAGGGTGGAAA | |
| DS1 + 2.0-R | AACTTTAGGATGGGGCTGCT | |
| DS1 + 3.0-F | TGTAGCCACAGTTTGCCTGT | |
| DS1 + 3.0-R | CTCCTCTATTGTCACCTGGAAGAC | |
| DS1 + 5.0-F | GGCCAACATCCCTGATGACTAC | |
| DS1 + 5.0-R | CACCCTTGCCAGCATTTGTT | |
| MACROD2 pro-F | ACGCAGCACAGTCCTTTGG | |
| MACROD2 pro-R | AGGACCTGAATTCTGTGGTGG | |
| GAPDH PRO-F | ATCCAAGCGTGTAAGGGTCC | |
| GAPDH PRO-R | TAGGGGGGAAGGGACTGAGA | |
| DSB1-335-F | GAATCGGATGTATGCGACTGATC | |
| DSB1-335-R | TTCCAAAGTTATTCCAACCCGAT | |
| DSB1-1618-F | TGAGGAGGTGACATTAGAACTCAGA | |
| DSB1-1618-R | AGGACTCACTTACACGGCCTTT | |
| DSB1-3500-F | TCCTAGCCAGATAATAATAGCTATACAAACA | |

| Reagent/resource | Reference or source | Identifier or catalog number |
|---|---|---|
| DSB1-3500-R | TGAATAGACAGACAACAGATAAATGAGACA | |
| DSB2-364-F | CCAGCAGTAAAGGGGAGACAGA | |
| DSB2-364-R | CTGTTCAATCGTCTGCCCTTC | |
| DSB2-1754-F | GAAGCCATCCTACTCTTCTCACCT | |
| DSB2-1754-R | GCTGGAGATGATGAAGCCCA | |
| DSB2-3564-F | GCCCAGCTAAGATCTTCCTTCA | |
| DSB2-3564-R | CTCCTTTGCCCTGAGAAGTGA | |
| siRNA | | |
| SiGATAD2B siGENOME SMARTPOOL | Dharmacon | M-013892-01-0010 |
| SiMBD3 siGENOME SMARTPOOL | Dharmacon | M-013616-01-0010 |
| ON-TARGETplus Non-targeting Control | Dharmacon | D-001810-01-20 |
| SiXRN2 siGENOME SMARTPOOL | Dharmacon | M-017622-01-0005 |
| SiSETX siGENOME SMARTPOOL | Dharmacon | M-021420-00-0005 |
| SiZMYND8 siGENOME SMARTPOOL | Dharmacon | M-017354-00-0005 |
| SiKDM5A siGENOME SMARTPOOL | Dharmacon | M-003297-02-0005 |
| SiCHD4 siGENOME SMARTPOOL | Dharmacon | L-009774-00-0005 |
| SiRNase H1 siGENOME SMARTPOOL | Dharmacon | M-012595-01-0005 |

## S9.6 immunoprecipitation

The protocol for this assay was described previously. Briefly, nuclei from ionized radiated or non-treated HEK293T cells were extracted by cell lysis buffer (5 mM PIPES pH 8.0, 0.5% NP-40 and 85 mM KCl) on ice for 10 min. Isolated nuclei were then resuspended in nuclear lysis buffer (10 mM Tris-HCl pH 7.5, 200 mM NaCl, 2.5 mM MgCl$_2$, 0.2% sodium deoxycholate [NaDOC], 0.1% SDS, 0.05% sodium lauroyl sarcosinate [Na sarkosyl] and 0.5% Triton X-100), followed by 10 min sonication (Diagenode Bioruptor). The extracts were then precleared, after which S9.6 antibodies were added for immunoprecipitation. Protein G dynabeads were then applied to bind with the antibody, followed by five times washing and elution.

## Mass spectrometry

Beads were subjected to a series of washes, including two washes in cold dilution buffer and two washes in cold PBS. The elution of proteins from the beads was carried out using 0.2 M glycine (pH 2) for 7 min at 4 °C, followed by neutralization with 1 M Tris (pH 8). This elution step was repeated twice to ensure efficient protein recovery. For in-solution protein samples, denaturation was performed using 4 M urea in ammonium bicarbonate buffer (100 mM) at room temperature for 10 min. Subsequently, cysteine residues were reduced using TCEP (10 mM) for 30 min at room temperature, followed by alkylation with 2-Chloroacetamide (50 mM) for 30 min at room temperature in the dark. Predigestion was carried out with LysC enzyme (1 g/100 g of sample) for 2 h at 37 °C. This was followed by overnight digestion with trypsin enzyme (1 g/40 g of sample) at 37 °C. The urea concentration was reduced to 2 M in ammonium bicarbonate buffer (100 mM), and calcium chloride was added at a final concentration of 2 mM before the trypsin digestion. On the next day, trypsin digestion was stopped by the addition of formic acid (5%). The digested peptides were centrifuged at 13,200 rpm at 4 °C for 30 min to remove any undigested material. The supernatant containing the peptides was loaded onto a handmade C18 stage tip, which had been pre-activated with 100% acetonitrile, and centrifuged at 4000 rpm at room temperature. The peptides were washed twice with TFA 0.1% and then eluted using 50% acetonitrile/0.1% TFA. The eluted peptides were dried using a speed vacuum. Prior to liquid chromatography coupled with tandem mass spectrometry analysis (LC-MS/MS), the peptides were resuspended in 2% acetonitrile/ 0.1% formic acid. Peptides were then trapped on an Acclaim™ PepMap™ 100 C18 HPLC Columns (PepMapC18; 300 μm × 5 mm, 5-μm particle size, Thermo Fischer) using solvent A (0.1% Formic Acid in water) at a pressure of 60 bar and separated on an Ultimate 3000 UHPLC system (Thermo Fischer Scientific) coupled to a QExactive mass spectrometer (Thermo Fischer Scientific). The peptides were separated on an Easy-Spray PepMap RSLC column (75 μm i.d. × 2 μm × 50 mm, 100 Å, Thermo Fisher) and then electrospray directly into an QExactive mass spectrometer (Thermo Fischer Scientific) through an EASY-Spray nano-electrospray ion source (Thermo Fischer Scientific) using a linear gradient (length: 60 min, 5% to 35% solvent B (0.1% formic acid in acetonitrile and 5% dimethyl sulfoxide), flow rate: 250 nL/min). The raw data was acquired on the mass spectrometer in a data-dependent mode (DDA). Full scan MS spectra were acquired in the Orbitrap (scan range 380–1800 $m/z$, resolution 70,000, AGC target 3e6, maximum injection time 100 ms). After the MS scans, the 15 most intense peaks were selected for HCD fragmentation at 28% of normalized collision energy. HCD spectra were also acquired in the Orbitrap (resolution 17500, AGC target 1e5, maximum injection time 128 ms) with first fixed mass at 100 $m/z$.

## Treatment of inhibitors, transfection of plasmids, and siRNA

When indicated, cells were treated with 1 μM triptolide or 100 μM 5,6-dichloro-1-beta-D-ribofuranosylbenzimidazole (DRB) to inhibit transcription 2 h before IR treatment or other assays. In the RNAseH overexpression group, cells were transfected with plasmids encoding RNAseH1 using Lipofectamine 3000 48 h prior to the subsequent step. For siRNA transfection, siRNA was delivered into cells via Lipofectamine RNAiMAX 48 h before IR treatment or other specified assays. For inhibitors, Cells were treated AZD5305 (10 nM, 6 h) to inhibit PARP1, Ku-55933 (10 μM 2 h) to inhibit ATM and AZ20 (500 nM 2 h) to inhibit ATR.

## Proximity ligation assay (PLA)

These experiments were performed according to manufacturers' instructions. Specifically, cells were fixed by 4% paraformaldehyde solution at room temperature for 10 min, followed by PBS washing and 10 min lysis by 0.2% Triton X-100. Cells were then blocked in PLA blocking buffer at 37 °C for 1 h, after which primary antibodies were diluted and incubated with samples at cold room overnight. Those primary antibodies were then captured by PLA probes which were then ligated, followed by amplification by polymerase. For S9.6-related PLA, cells were pre-extracted by CSK buffers (10 mM PIPES, 100 mM NaCl, 300 mM Sucrose,3 mM MgCl$_2$, 1 mM EGTA, 0.7% Triton X-100, 1× protease inhibitor cocktail) at cold room for 15 min prior to fixation according to a previous report, while 1 h incubation at room temperature with RNase treatment solution (0.1% BSA, 3 mM MgCl$_2$, 1:200(v/v) RNAse T1, 1:200 (v/v) ShortCut RNAse III in PBS) was required for each sample after fixation.

## Laser micro-irradiation

Cells were transfected with related GFP or mCherry-tagged plasmids 48 h before experiments, and then seeded on glass-bottomed dish (NEST) and pre-treated with 10 μM BrDU 24 h prior to irradiation. UV laser (16 Hz pulse, 40% laser output) generated from the Micropoint System (Andor) was then applied to induce DNA damage. Live-cell images were then taken by a Nikon A1 confocal imaging systems for every 5 or 10 s. Images were then analyzed by ImageJ and intensities of damage, non-damage and background areas were measured respectively to calculate relative intensities of stripped areas.

## Chromatin immunoprecipitation (ChIP)

AsiSI-ER U2OS cells were treated 400 nM 4-Hydroxytamoxifen (4OHT) 4 h before 10 min fixation by 1% formaldehyde solution. Fixed cells were then quenched by 125 mM glycine at room temperature for 10 min and scrapped at cold room. Later, tissues were lysed in cell lysis buffer mentioned above and nuclei lysis buffer (50 mM Tris-HCl pH 8.0, 1% SDS and 10 mM EDTA) respectively on ice and sonicated for 15 min. Samples were then spun for 10 min at 13,000 rpm to remove cell debris. The supernatant was the precleared by protein G dynabeads and diluted by dilution buffer (16.7 mM Tris-HCl pH 8.0, 0.01% SDS, 1% Triton X-100, 167 mM NaCl, 1 mM EDTA) in a 1:4 ratio. Diluted samples were then aliqoted into several ChIP samples. Antibodies and beads were then added for immunoprecipitation, and beads were washed five times before elution in ChIP elution buffer (1% SDS and 100 mM NaHCO3) at room temperature for 30 min. Eluted samples were then de-crosslinked at 65 °C overnight and immunoprecipitated DNAs were then extracted by using Phenol-Chloroform extraction methods for following qPCR and sequencing experiments. For primers, see the Reagents and Tools table.

## MNase assay

Cells were collected in clean Eppendorf tubes, lysed in buffer A (50 mM Tris-HCl pH 8.0, 0.1% Triton X-100, 150 mM NaCl, 1 mM EDTA) to extract chromatin. Chromatin pellets were then resuspended with digestion buffer (15 mM Tris-HCl pH 7.5, 60 mM KCl, 15 mM NaCl, 0.25 M sucrose, 1 mM CaCl$_2$, 0.5 mM DTT) with 100 units of MNase (NEB), incubated at 37 °C for 5–10 min. Digested DNA was then purified by Phenol-Chloroform extraction methods and separated by 1.2% agarose gel electrophoresis. Each representative image was selected from three biological repeats.

## Chromatin relaxation and DNA accessibility assay

The chromatin relaxation assay was performed as in Smith et al, 2023. This assay uses PAGFP tagged histone H2B to mark a chromatin region by local photoactivation. This happens at the same time as laser irradiation at 405 nm causes DNA damage. The assay measures the thickness of the photoactivated line to estimate how much the chromatin condenses at the damaged sites.

BZIP DNA domain from C/EBPa transcription factor was fused to GFP (Smith et al, 2023). Cells were transiently transfected and subjected to laser micro-irradiation and monitored BZIP-GFP recruitment to DSBs in control and siGATAD2B cells.

## Single-stranded DNA resection assay

A single-stranded DNA resection assay was performed as described in a previous work (Zhou et al, 2014). Briefly, Control (NC) or GATAD2B siRNA were transfected using Lipofectamine RNAi Max (Life Technologies) for 72 h prior to induction of site-specific double-strand breaks (DSBs) in DIvA cells by incubation with 500 nM 4OHT for 4 h. Genomic DNA was then extracted from these cells using the Genomic DNA purification Kit (New England BioLabs). Following genomic DNA extraction, 1000 ng of genomic DNA was digested using 10 units of BsrGI or BamHI at 37 °C overnight. The percentage of single-stranded DNA (ssDNA) was then measured and calculated by qPCR. $\Delta$CT is defined as the difference in CT value between a digested sample and its undigested counterpart, and the percentage of ssDNA was calculated as follows: % ssDNA = $1/[2^{(\Delta CT - 1)} + 0.5] * 100$.

## HR and NHEJ reporter assay

DR-U2OS or EJ5-U2OS reporter cell lines were transfected siRNA by Lipofectamine RNAi Max (Life Technology) 48 h prior to I-SceI lentivirus transduction. After infection by lentivirus containing I-SceI, reporter cell lines were cultured at 37 °C for 48 h before trypsinized and resuspended by PBS. Samples were then analyzed by a BD Accuri C6 flow cytometer.

## Immunofluorescence

Cells were seeded on glass-bottomed dish or coverslips 24 h prior to ionic irradiation, followed by CSK pre-extraction and fixation which were mentioned above. Coverslips were then blocked with 3% BSA at 37 °C for 1 h, after which samples would be incubated with primary antibodies overnight at cold room and secondary antibodies at 37 °C for 1 h subsequently. For BrdU IF, cells were incubated with 20 μM BrdU 24 h prior to 5 Gy IR treatment, followed by 2 h recovery before the CSK extraction step. Cells were mounted on slides with Fluoroshield Mounting Medium with DAPI (Abcam). Images were then acquired by an Olympus Fluoview FV1200 confocal microscope.

## Site-specific DSBs induction by IPpoI and measurement of 28 s rDNA copy number

HeLa cells were transfected with plasmids encoding IPpoI-ER 36 h before treatment with 2 μM 4OHT or DMSO for 3 h. Subsequently, the cells were cultured in DMEM media without 4OHT for an additional 5 h. Genomic DNA was then isolated from these cells by phenol-chloroform extraction. Following this, relative 28 S rDNA copy numbers were quantified by qPCR and normalized using GAPDH as a reference.

## Comet assay

The assay was performed according to manufacturers' instructions (bio-techne). Briefly, $5 \times 10^3$ cells in 5 μl PBS were mixed with 50 μl molten LMAgarose at 37 °C, and spread onto CometSlide. Comet-Slides were then transferred to 4 °C, before lysed in lysis buffer (2.5 M NaCl, 100 mM Na$_2$EDTA, 10 mM Tris, with final pH 10.0) with 10% DMSO and 1% Triton X-100 at 4 °C for at least 2 h. Samples were then immersed into electrophoresis buffer (300 mM NaOH, 1 mM EDTA) to unwind for 30 min. The electrophoresis was performed under 21 V, 200 mA condition at cold room. Slides were then washed 4 times and stained by Propidium Iodide for 30 min before imaging on a fluorescence microscope. For each condition, at least 50 cells were quantified by ImageJ. Tail moment, which is the combination of tail length and intensity of the tail, was quantified and calculated by OpenComet.

## Cell cycle analysis

Cells were harvested 2 h after 5 Gy irradiation. 1 h prior to harvest, cells were incubated with 10 μM EdU. Harvested cell pellets were washed twice with PBS, followed by fixation in 70% EtOH at 4 °C for 2 h. Cells were then permeabilized with 0.1% Triton X-100 in 1% BSA at 4 °C for 15 min, followed by two washes with 0.1% Tween-20 in 1% BSA. Subsequently, cells were incubated with 2 mM CuSO4, 10 mM Sodium Ascorbate, and 10 μM Alexa Fluor 555 in the dark at 37 °C for 1 h, followed by 1 h of DAPI staining at room temperature. Samples were then analyzed using a BD Accuri C6 flow cytometer.

For all reagents used in this study, see Reagent and Tools Table.

## Bioinformatic analyses

### ChIP-Seq data processing

GATAD2B + 4OHT, GATAD2B -4OHT, HDAC1 + 4OHT, HDAC -4OHT samples were sequenced using Illumina NextSeq 500 (single end 151 bp reads). Read quality of the fastq files was checked using

FastQC (https://www.bioinformatics.babraham.ac.uk/projects/fastqc/) before and after adapter trimming. The trimmed reads were then mapped to hg19 genome following the standard ChIP-seq pipeline. The classic ChIP-seq pipeline consists of BWA (http://bio-bwa.sourceforge.net/) for alignment and samtools (http://www.htslib.org/) for duplicate removal (rmdup), sorting (sort) and indexing (index). deepTools (https://deeptools.readthedocs.io/en/develop/) (Ramirez et al, 2016) bamCompare was used to calculate log2 fold change at each nucleotide position between +4OHT and -4OHT for GATAD2B and HDAC1 samples. For peak calling, we have first extracted peaks with Input as control using MACS2 software. We then compared the replicates using ChIP QC software. PCA plots comparing coverage across consensus peaks show that at least two replicates per condition cluster together. For the downstream analysis, we used one of the clustering replicates for each condition.

Read count normalization was applied as part of bamCompare to normalize for sequencing depth between samples. H4K12Ac + 4OHT and H4K12Ac -4OHT ChIP-seq samples from (Clouaire et al, 2018b) (E-MTAB-5817) was processed using the same approach for downstream analysis plots.

RNA Pol II S5P ChIP from a previous study (Cohen et al, 2018) (E-MTAB-6318) was processed using the classical ChIP pipeline as well. Read count coverage was calculated for each the 80 inducible DSB site (+/−5kb) using bedtools multicov (https://bedtools.readthedocs.io/en/latest/) (Quinlan and Hall, 2010). The top 20 sites with highest coverage were annotated as high transcriptional activity and the bottom 20 as low transcriptional activity.

### BLESS Seq data processing

BLESS Seq (E-MTAB-5817) was processed using the same protocol as detailed in previous publication (Clouaire et al, 2018a). Read count coverage was calculated for all annotated DSBs (+/−500bp) using bedtools multicov. The sites were then ordered based on read count coverage for representing cleavage efficiency of DSB sites.

### DRIP-Seq data processing

DRIP-Seq samples +/− 4OHT from a previous research work (Cohen et al, 2018) (E-MTAB-6318) was processed using the classical ChIP pipeline. Log2 Fold Change was calculated using bamCompare at each nucleotide position with read count normalization. bamCompare outputs were exported as bigwig files for further downstream analysis.

## Metagene profiles

computeMatrix operation of deepTools was used to calculate the average profile of log2Fold Change ( + 4OHT/-4OHT) from the bigwig files generated using bamCompare. The bin size was set to 200 and region length was set to +/−10kb with reference to DSB. The average profile matrices created for GATAD2B, HDAC1 and DRIP-Seq were then combined and plotted using matplotlib package of python.

## BoxPlots

Read count coverage around DSBs (+/−2.5 kb) was calculated using bedtools multicov and then normalized to CPM using total read count of respective samples. The plots were made with matplotlib package of python. Wilcoxon two-sample test was used to determine statistical significance with $P$ value cut-off set at 0.01. Statistical tests were run using the Scipy Python package.

## IGV profiles and heatmaps

Coverage files containing CPM normalized read count per nucleotide position was generated for each sample using deeptools bamCoverage. The coverage files were exported in bigwig format and loaded into IGV browser (https://software.broadinstitute.org/software/igv/) for visualizing the normalized read density around break sites.

The coverage files were also used to make the heatmaps comparing cleavage efficiency and read density of HDAC1, GATAD2B, R-loops and H4K12Ac around DSB in 4OHT-treated samples. plotHeatmap function of deepTools was used to make the heatmaps with regions set as all annotated DSBs arranged in ascending order of cleavage efficiency.

## Quantification and statistical analysis

For IF and PLA, at least 50 cells were analyzed. Foci numbers were quantified by cell profiler software. For laser, fluorescence intensity was measured by software imageJ. Unless indicated, two-tailed Student's *t* tests were applied for statistical analysis, *$P < 0.05$, **$P < 0.01$, ***$P < 0.001$ and ****$P < 0.0001$.

## Data availability

The mass spectrometry proteomics data have been deposited to the ProteomeXchange Consortium via the PRIDE partner repository with the dataset identifier PXD042271. ChIP-seq datasets are deposited in GEO database under accession number GSE230710.

The source data of this paper are collected in the following database record: biostudies:S-SCDT-10_1038-S44318-024-00111-7.

## Peer review information

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

## Acknowledgements

The authors would like to thank all the members of the Gullerova and Zhu labs for their help and advice throughout this study. The authors are grateful to Dr. Rebecca Smith for her help with chromatin relaxation assays. The authors also thank to Dr. Jun Zhang for his help with microscopy imaging and stimulating scientific discussion and Xingkai He with her assistance with sample preparation for qRT-PCRs. The authors are grateful to Dr. Gaelle Legube for providing HR and NHEJ-prone AsiSI cut site genome coordinates. This work was supported by the Senior Research Fellowship by Cancer Research UK [grant number BVR01170], EPA Trust Fund [BVR01670], and Lee Placito Fund awarded to MG. This study was supported by the National Natural Science Foundation of China (grant numbers 32090030 and 32090033). The Science and Technology Program of the Guangdong Province in China (2017B030301016), and the Shenzhen Municipal Commission of Science and Technology Innovation (JCYJ20200109114214463).

## Author contributions

**Zhichao Liu**: Conceptualization; Formal analysis; Validation; Investigation; Methodology; Writing—original draft. **Kamal Ajit**: Software; Investigation; Methodology. **Yupei Wu**: Investigation. **Wei-Guo Zhu**: Conceptualization; Resources; Supervision; Funding acquisition. **Monika Gullerova**: Conceptualization; Resources; Supervision; Funding acquisition; Investigation; Methodology; Project administration; Writing—review and editing.

Source data underlying figure panels in this paper may have individual authorship assigned. Where available, figure panel/source data authorship is listed in the following database record: biostudies:S-SCDT-10_1038-S44318-024-00111-7.

## Disclosure and competing interests statement

The authors declare no competing interests.

# Expanded View Figures

**Figure EV1.  The GATAD2B-NuRD complex binds to DNA:RNA hybrids upon DNA damage.**

(A) Representative confocal images showing single antibody controls for PLA shown in Fig. 1. Scale bar = 10 µm. (B) Representative confocal images from three independent replicates showing PLA of HDAC1 and S9.6 in cells with or without IR and overexpression of RNAseH1 Left: representative confocal microscopy images and single antibody PLA control; right: quantification of left, error bar = mean ± SEM, significance was determined using nonparametric Mann–Whitney test. ****$P \leq 0.0001$. Scale bar = 10 µm, $n > 50$ cells from three biological repeats. (C) Representative confocal images from three independent replicates showing immunofluorescence of γH2AX staining upon laser-induced DNA damage. Cells were fixed ~2 min after irradiation of laser. Nuclei were stained by DAPI. Scale bar = 10 µm, $n > 3$. (D) Laser stripping of MBD3-GFP cells with or without treatment with transcription inhibitors triptolide (TPL3) and DRB. Representative spinning disk confocal microscopy images from three independent replicates and quantification ($n \geq 10$) showing GFP signals before and after laser striping at indicated time points; error bar = mean ± SEM. Scale bar = 10 µm, $n > 3$. (E) Laser stripping of MBD3-GFP cells with or without transiently expression of RNAseH1-RFP plasmid. Representative spinning disk confocal microscopy images from three independent replicates and quantification ($n \geq 10$) showing GFP and RFP signals before and after laser striping at indicated time points; error bar = mean ± SEM. Scale bar = 10 µm, $n > 3$. (F) Quantification ($n \geq 10$) showing GATAD2B-GFP and RNH1-mCherry signals before and after laser striping at indicated time points; error bar = mean ± SEM. $n = 3$ biological repeats. (G) ChIP-qPCR bar charts showing levels of HDAC1 (left) and GATAD2B (right) at three genes known to be bound by NuRD complex in non-damage condition in cells treated with Triptolide or overexpressing RNaseH1, $n = 3$, error bar = mean ± SEM. (H) Representative confocal images from three independent replicates showing PLA of MBD3 and γH2AX in cells with or without IR followed by 20 min recovery, and transcription inhibition (TLP3 or DRB) or overexpression of RNAseH1. IR = 5 Gy. Left: representative confocal microscopy images; right: quantification of left, error bar = mean ± SEM, significance was determined using nonparametric Mann–Whitney test. ****$P \leq 0.0001$. Scale bar = 10 µm, $n > 3$.

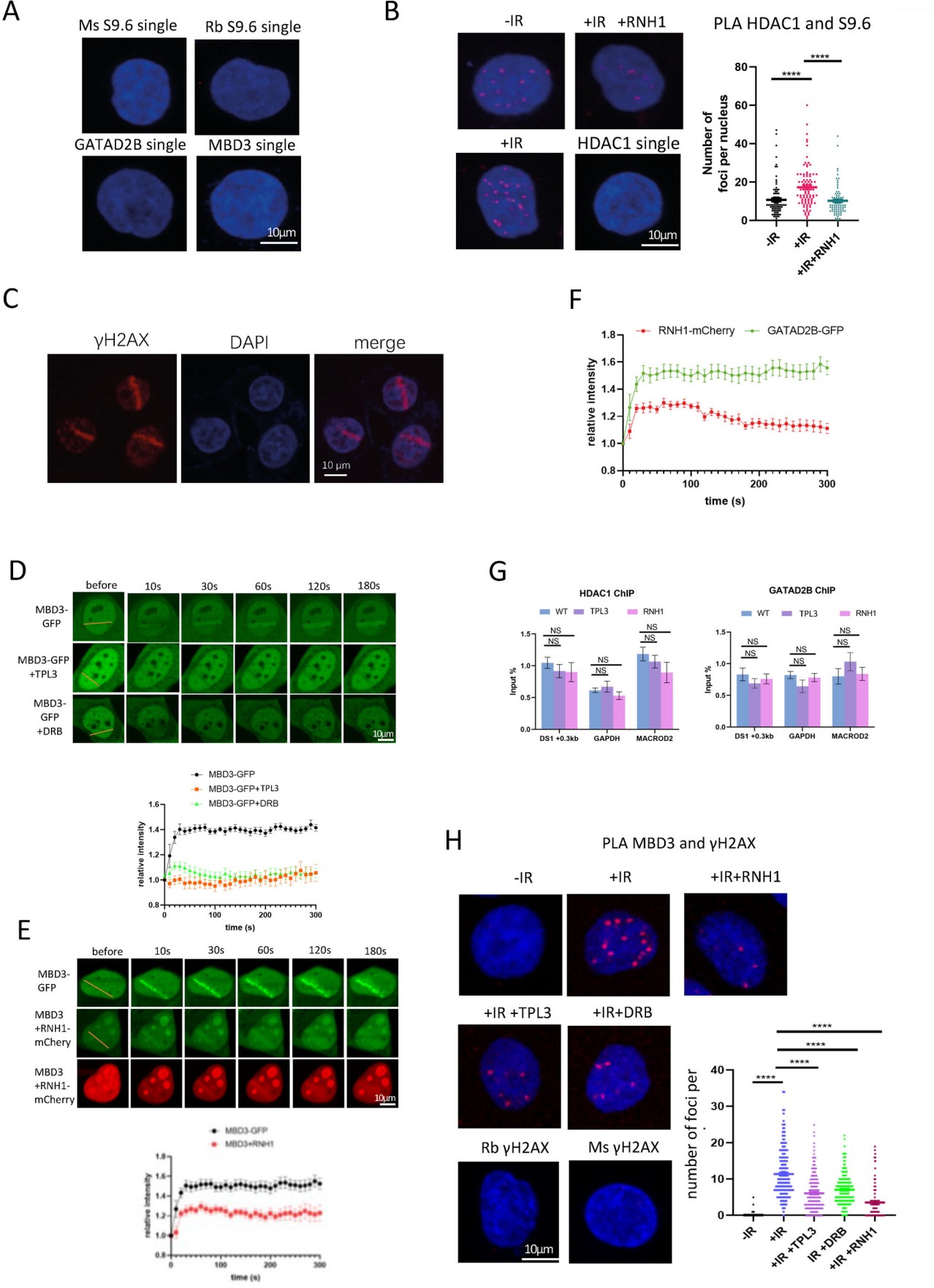

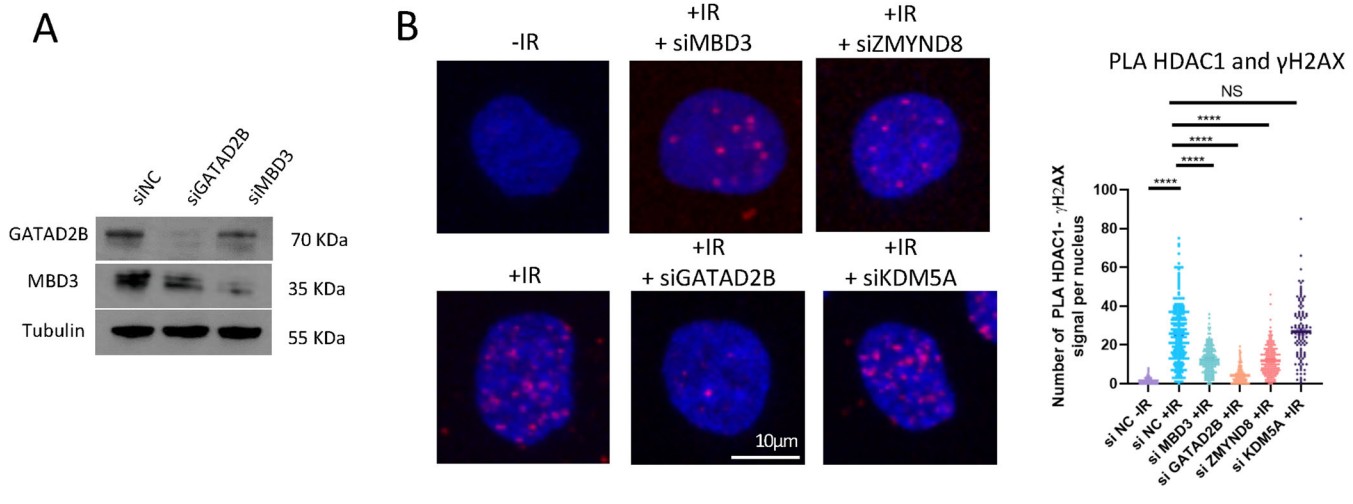

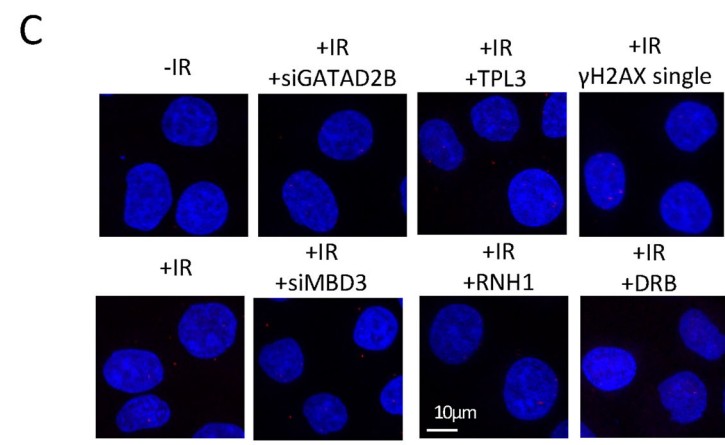

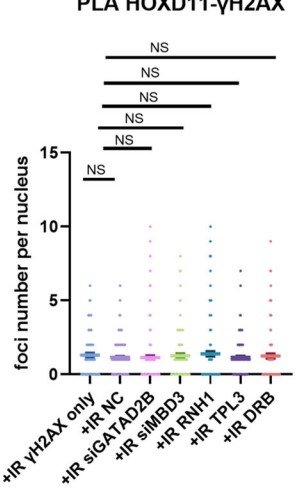

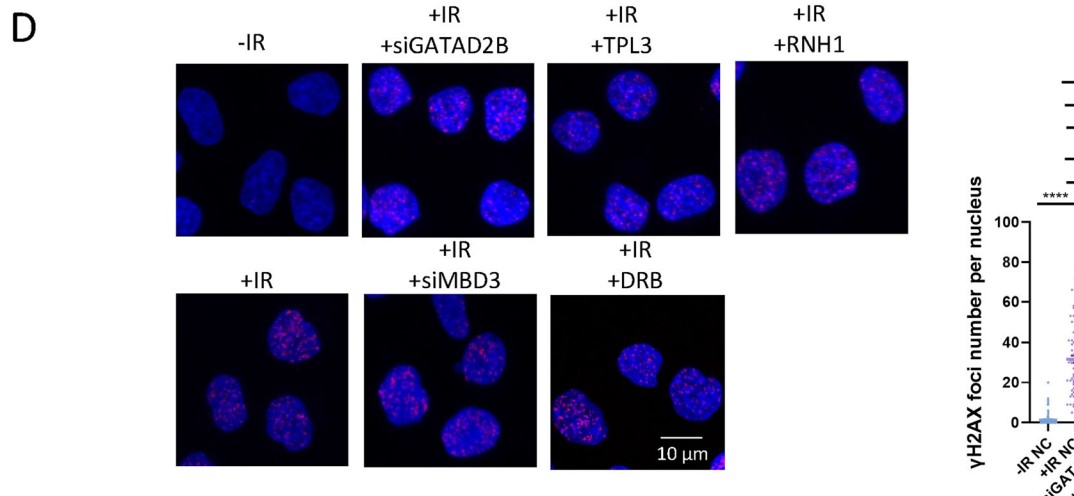

◀  **Figure EV2.  The GATAD2B-NuRD complex specifically localises to DSBs.**

(A) Western blot showing efficiency of siRNA mediated knockdown of GATAD2B and MBD3, NC negative control. Tubulin was used as loading control.
(B) Representative confocal images from three independent replicates showing PLA of HDAC1 and γH2AX in cells with or without IR and depleted of ZMYND8 or KDM5A, IR = 5 Gy. Left: representative confocal microscopy images; right: quantification of left, error bar = mean ± SEM, significance was determined using nonparametric Mann–Whitney test. ****$P$ ≤ 0.0001. Scale bar = 10 μm, $n$ > 50 cells from 3 biological repeats. (C) Representative confocal images from three independent replicates showing PLA of HOXD11 and γH2AX in cells with or without IR followed by 20 min recovery, and transcription inhibition (TLP3 or DRB), overexpression of RNAseH1 or depletion of GATAD2B and MBD3, IR = 5 Gy. Left: representative confocal microscopy images; right: quantification of left, error bar = mean ± SEM, significance was determined using nonparametric Mann–Whitney test. ****$P$ ≤ 0.0001. Scale bar = 10 μm, $n$ > 50 cells from three biological repeats. (D) Immunofluorescence of γH2AX in cells with 5 Gy IR followed by 20 min recovery, and transcription inhibition (TLP3 or DRB), overexpression of RNAseH1 or depletion of GATAD2B and MBD3. Left: representative confocal microscopy images; right: quantification of left, error bar = mean ± SEM, significance was determined using nonparametric Mann–Whitney test. ****$P$ ≤ 0.0001, NS no significance. Scale bar = 10 μm, $n$ > 50 cells from three biological repeats.

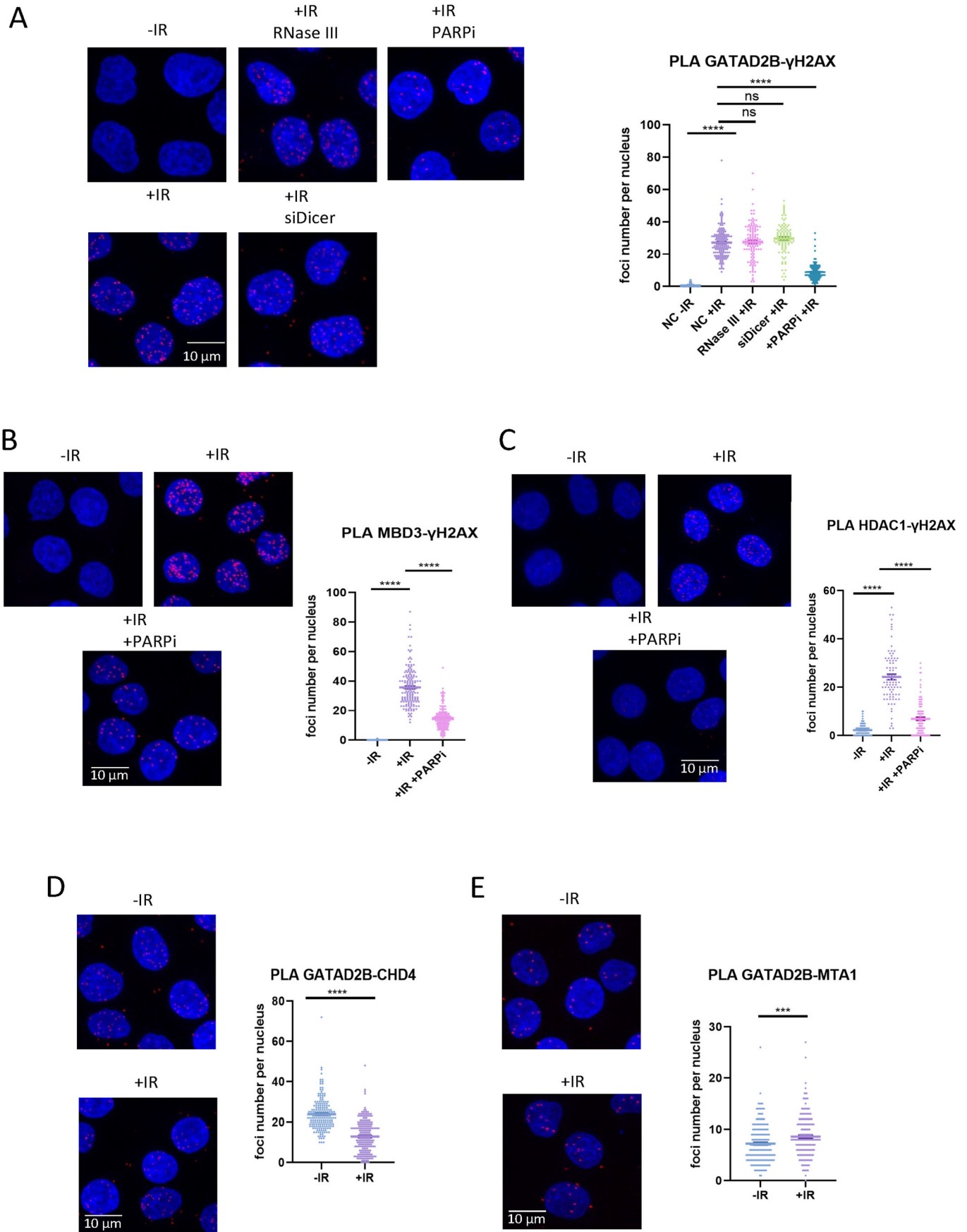

◀ **Figure EV3.   The GATAD2B-NuRD localises to DSBs in PARP1-dependent manner.**

(**A**) PLA of GATAD2B and γH2AX in cells with or without IR treatment followed by 20 min recovery and treatment of the PARP inhibitor or RNase III or knockdown of Dicer. IR = 5 Gy. Left: representative confocal microscopy images from three independent replicates; right: quantification of left, error bar = mean ± SEM, significance was determined using nonparametric Mann–Whitney test. ****$P \le$ 0.0001. Scale bar = 10 μm, $n > 50$ cells from three biological repeats. (**B**) PLA of MBD3 and γH2AX in cells with or without IR treatment followed by 20 min recovery and treatment of the PARP inhibitor. IR = 5 Gy. Left: representative confocal microscopy images from three independent replicates; right: quantification of left, error bar = mean ± SEM, significance was determined using nonparametric Mann–Whitney test. ****$P \le$ 0.0001. Scale bar = 10 μm, $n > 50$ cells from three biological repeats. (**C**) PLA of HDAC1 and γH2AX in cells with or without IR treatment followed by 20 min recovery and treatment of the PARP inhibitor. IR = 5 Gy. Left: representative confocal microscopy images from three independent replicates; right: quantification of left, error bar = mean ± SEM, significance was determined using nonparametric Mann–Whitney test. ****$P \le$ 0.0001. Scale bar = 10 μm, $n > 50$ cells from three biological repeats. (**D**) PLA of GATAD2B and CHD4 in cells with or without IR treatment followed by 20 min recovery. IR = 5 Gy. Left: representative confocal microscopy images from three independent replicates; right: quantification of left, error bar = mean ± SEM, significance was determined using nonparametric Mann–Whitney test. ****$P \le$ 0.0001. Scale bar = 10 μm, $n > 50$ cells from 3 biological repeats. (**E**) PLA of GATAD2B and MTA1 in cells with or without IR treatment followed by 20 min recovery. IR = 5 Gy. Left: representative confocal microscopy images from three independent replicates; right: quantification of left, error bar = mean ± SEM, significance was determined using nonparametric Mann–Whitney test. ***$P \le$ 0.001. Scale bar = 10 μm, $n > 50$ cells from three biological repeats.

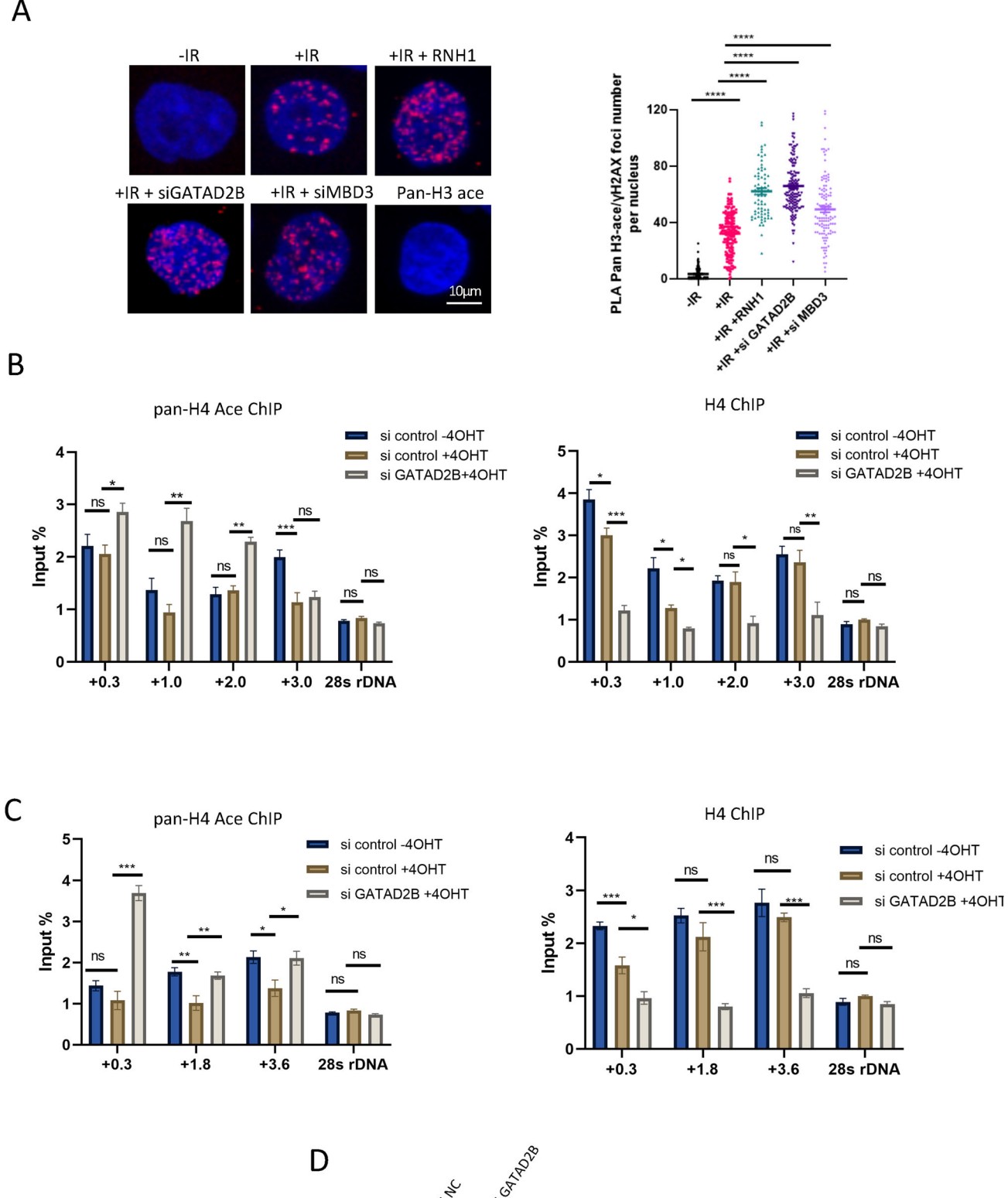

**Figure EV4. The lack of GATAD2B-NuRD complex leads to histone hyper-acetylation at DSBs.**

(A) PLA of pan-acetyl H3 and γH2AX in cells with or without IR followed by 20 min recovery, and overexpression of RNAseH1 or depletion of GATAD2B and MBD3. IR = 5 Gy. Left: representative confocal microscopy images from three independent replicates; right: quantification of left, error bar = mean ± SEM, significance was determined using nonparametric Mann–Whitney test. ****$P \leq 0.0001$. Scale bar = 10 μm, $n > 50$ cells from three biological repeats. (B) Bar chart showing pan-acetyl H4 (left) and histone H4 (right) ChIP levels at indicated sites (related to Fig. 4E) next to AsiSI cut in cells with or without 4OHT and depletion of GATAD2B, error bar = mean ± SEM, significance was determined using nonparametric Mann–Whitney test. ***$P \leq 0.001$, **$P \leq 0.01$, n.s. not significant, $n = 3$. (C) Bar chart showing pan-acetyl H4 (left) and histone H4 (right) ChIP levels at indicated sites (related to Fig. 4F) next to AsiSI cut in cells with or without 4OHT and depletion of GATAD2B, error bar = mean ± SEM, significance was determined using nonparametric Mann–Whitney test. ***$P \leq 0.001$, **$P \leq 0.01$, n.s. not significant, $n = 3$. (D) Western blot showing effect of knockdown of GATAD2B on histone H4 acetylation levels, NC, negative control. Histone H4 was used as loading control.

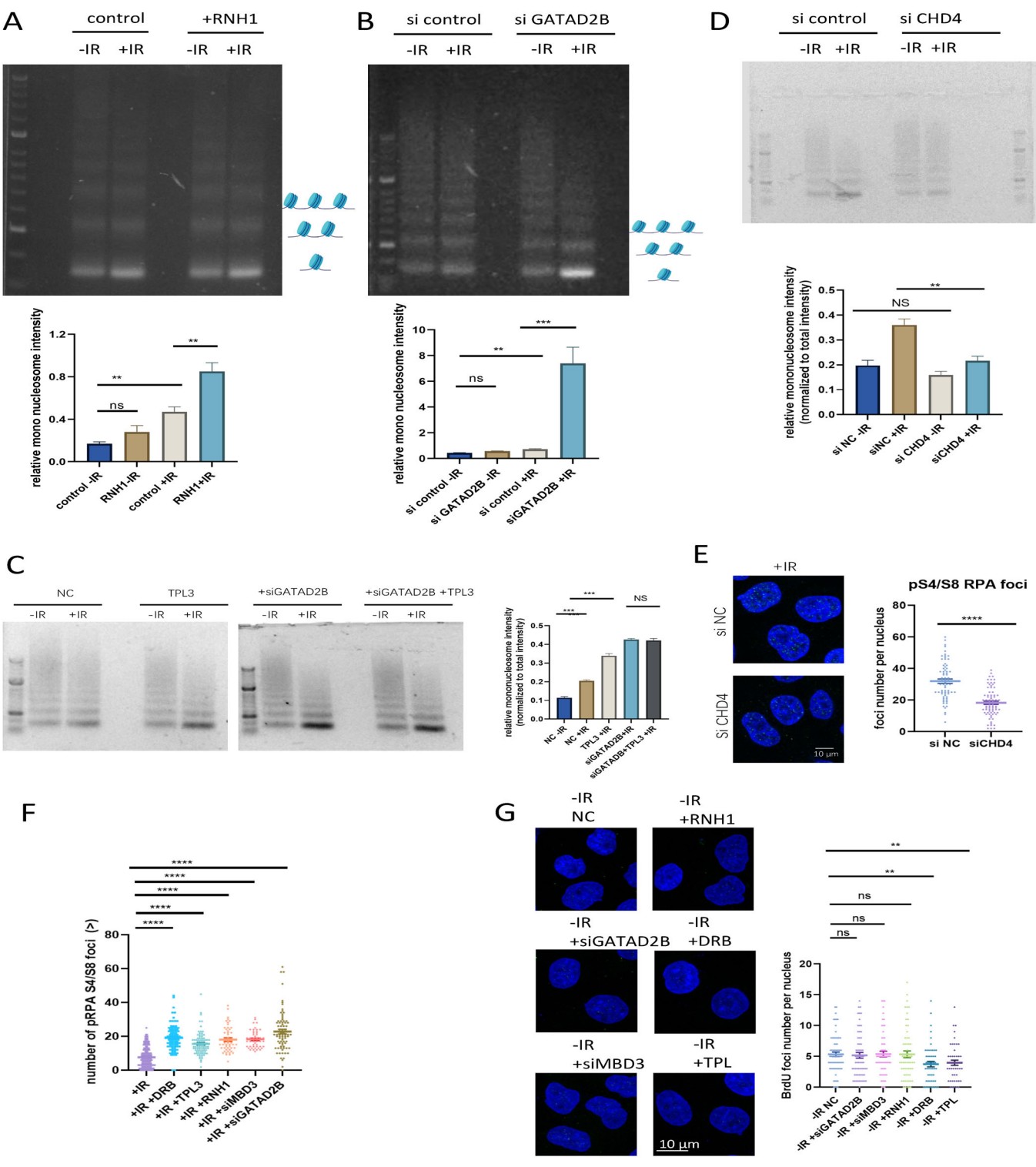

◄ **Figure EV5.   The GATAD2B-NuRD complex prevents chromatin hyper-relaxation.**

(A) Top: Representative DNA gel from three independent replicates showing nucleosome profile of wt and cells overexpressing RNaseH, with or without 5 Gy IR followed by 15 min recovery, after MNase treatment and DNA extraction. Bottom: Bar chart showing relative mononucleosome intensity of each lane. Significance was determined by Student *t* test: **$P \leq$ 0.01, n.s. not significant, error bar = mean ± SEM. (B) Top: Representative DNA gel from three independent replicates showing nucleosome profile of wt and cells depleted of GATAD2B, with or without 5 Gy IR followed by 15 min recovery, after MNase treatment and DNA extraction. Bottom: Bar chart showing relative mononucleosome intensity of each lane. Significance was determined by Student *t* test: ***$P \leq$ 0.001, **$P \leq$ 0.01, n.s. not significant, error bar = mean ± SEM. (C) Left: Representative DNA gel from three independent replicates showing nucleosome profile of wt and cells depleted of GATAD2B, with or without triptolide treatment and in combination, with or without 5 Gy IR followed by 15 min recovery, after MNase treatment and DNA extraction. Right: Bar chart showing relative mononucleosome intensity of each lane. Significance was determined by Student *t* test: ***$P \leq$ 0.001, n.s. not significant, error bar = mean ± SEM. (D) Top: Representative DNA gel from three independent replicates showing nucleosome profile of wt and cells depleted of CHD4, with or without 5 Gy IR followed by 30 min recovery, after MNase treatment and DNA extraction. Bottom: Bar chart showing relative mononucleosome intensity of each lane. Significance was determined by Student *t* test: **$P \leq$ 0.01, n.s. not significant, error bar = mean ± SEM. (E) Left: representative confocal images showing immunofluorescence signals of phospho-S4/S8 RPA32 in cells treated with 5 Gy IR followed by 2 h recovery with knockdown of CHD4. Right: quantification of left, error bar = mean ± SEM, significance was determined using nonparametric Mann–Whitney test. ****$P \leq$ 0.0001. Scale bar = 10 μm, *n* > 50 cells from three biological repeats. (F) Analysis and quantification of larger phospho-S4/S8 RPA32 foci, as defined by defined by a prominence threshold of 2500 in the Find Maxima tool, in cells treated with 5 Gy IR followed by 2 h recovery with overexpression of RNAseH1 or transcription inhibition (TLP3 or DRB) or depletion of GATAD2B and MBD3. Error bar = mean ± SEM, significance was determined using nonparametric Mann–Whitney test. ****$P \leq$ 0.0001, Scale bar = 10 μm, *n* > 50 cells from three biological repeats. (G) Representative confocal images from three independent replicates showing immunofluorescence signals of BrdU in cells upon no damage conditions with transcription inhibition (TLP3 or DRB), overexpression of RNAseH1 or depletion of GATAD2B and MBD3. Right: quantification of left, error bar = mean ± SEM, significance was determined using nonparametric Mann–Whitney test. **$P \leq$ 0.01, ns, no significance. Scale bar = 10 μm, *n* > 50 cells from three biological repeats.

