## [Peer Review File · The EMBO Journal]

The GATAD2B-NuRD complex drives DNA:RNA hybrid-dependent chromatin boundary formation upon DNA damage

Zhichao Liu, Kamal Ajit, Yupei Wu, Wei-Guo Zhu, and Monika Gullerova

Corresponding author(s): Monika Gullerova (monika.gullerova@path.ox.ac.uk)

Review Timeline:

Submission Date:	10th Nov 23
Editorial Decision:	19th Dec 23
Revision Received:	18th Mar 24
Editorial Decision:	16th Apr 24
Revision Received:	17th Apr 24
Accepted:	18th Apr 24

Editor: Hartmut Vodermaier

Transaction Report:

Dear Monika,

Thank you for submitting your manuscript on a NuRD complex role in setting up chromatin boundaries at DNA double strand breaks in response to R-loop formation. The study has now been reviewed by three expert referees, whose comments I am copying below for your information. As you will see, the reviewers appreciate the comprehensive analyses and the overall interest of the work, but also remain unconvinced that all main conclusions are decisively supported by the current data. Especially referees 1 and 2 raise a number of overlapping key issues, including unresolved discrepancies with published literature, possibility of alternative explanations, uncertainty about bona fide core NuRD and R-loop roles; as well as over-reliance on, or insufficient control of, potentially artifact-prone experimental approaches. Since it is not clear if and how these issues could be adequately addressed during a regular round of major revision, I would be -prior to taking a final decision in this case- interested in considering a tentative point-by-point response, detailing how you might address the main concerns of the referees, should you be given the opportunity to revise this work for The EMBO Journal. Based on such a revision proposal, I could then determine whether a major revision for The EMBO Journal would seem realistic, or whether a less substantively revised version might at least be suitable for one of our sister journals. I'd also be happy to talk through such a revision proposal with you if needed.

Looking forward to hearing from you,

Best regards,

Hartmut

Referee #1 (Report for Author)

The manuscript from Liu et al addresses a potential role for the GATAD2B containing NuRD complex in DNA repair. They use a proteomics-based approach to identify GATAD2B and MBD3 in pull-down samples using an antibody (S9.6) that can recognise RNA:DNA hybrids as well as dsRNA. They show that GATAD2B is recruited to DNA breaks and this is impaired when transcription is inhibited or RNaseH1 is overexpressed, suggesting that recruitment is driven by R loops. They show that HDAC1 is recruited to DNA breaks and this is reduced when GATAD2B is depleted. They perform ChIP-seq of GATAD2B and HDAC1 and show that the proteins accumulate at DNA breaks and that there is a relationship with actively transcribed genes,

which are preferentially repaired by HR. Consistent with this, there is an HR defect in GATAD2B depleted cells. They use existing datasets to explore a potential relationship with R loops and H4K12ac. They use PLA assays to suggest that depletion of GATAD2B or MBD3 leads to increased H3 and H4 acetylation around DNA breaks. They then look at chromatin relaxation after DNA damage using microscopy based approaches and conclude that GATAD2B prevents relaxation. Finally, they test the effect of GATAD2B or MBD3 depletion on end resection and show that there is an increase in phosphorylated RPA foci, suggesting that these proteins normally function to impede resection at DNA breaks.

This work is consistent with some previously published reports implicating the NuRD complex in DNA double strand break responses: it has been shown that NuRD is recruited to DNA breaks and promotes repair by HR. The work here potentially extends the understanding of how NuRD functions by showing a dependence on transcription and R loops for recruitment, although how this fits with previous data showing recruitment via other mechanisms is not clear.

Importantly, however, some of the data presented here directly contrast with existing literature. Specifically, there is data to suggest that the NuRD complex promotes resection (PMID: 22219182, PMID: 26546801) and promotes chromatin relaxation (as mentioned in the discussion). In addition, previous work showed that there is a role for NuRD in regulating DNA damage signaling and checkpoint responses (for example, PMID: 20805324 and 20805320 and 20693977). These functions can indirectly impact on downstream responses, making interpretation of some of the data presented here difficult and limits the impact of the findings.

There are a few potential reasons for the conflicting data. One is that GATAD2B is not functioning in the context of NuRD in these activities (most - but not all- of the literature focuses on CHD4 or HDAC subunits of NuRD). It is worth noting that CHD4 was not present in the irradiated R loop associated factors (in fact, appeared to be associated in unirradiated controls), and no other NuRD subunits were obviously present in these data. Other possibilities include experimental design and technical issues. While the discussion mentions the relaxation issue, the explanation doesn't fit well with the data, and it is important to address this experimentally before this study is published. In addition, the differences in resection with previous reports is not well addressed in the discussion.

Major issues:

1. To address the possibility that the data presented here are specific to the GATAD2B and/or MBD3 proteins, depletion of CHD4 should be performed in parallel for key experiments. Particularly the experiments that contradict previous reports - RPA foci and changes in chromatin compaction.
2. The impact of NuRD on DNA damage responses includes changes in checkpoint responses and signalling, so timepoints become important for these assays. A FACS profile should be included for the time/doses used to show that depletion of GATAD2B or MBD3 isn't indirectly

affecting these responses through changes in cell cycle profile.

3. The manuscript relies heavily on PLA, which is notorious for artefacts. One major issue is apparent co-localization after irradiation with gH2AX, when this will occur by chance with any abundant chromatin protein when doses are high. At the very least, the number of gH2AX foci present in these cells at the indicated time/doses should be reported (and representative images shown). Ideally, additional controls (eg an abundant chromatin protein that is not involved in these responses) could be added.

4. The raw mass spec data should be provided.

5. To really conclude that these proteins are important for resection termination, a time course and/or measurement of resected chromatin should be performed. The number of RPA or BrdU foci could reflect differences in resolution and rates of repair, and give no direct readout of resection distance.

Minor issues:

1. The discussion should be expanded to include more of the existing literature on this complex.

2. The figure legends should have more information. For example, it would be very helpful to know what time points and IR doses were used in the immunofluorescence experiments in Figure 6 (this is true for all the legends in the manuscript).

Referee #2 (Report for Author)

Liu et al. have studied the regulation of chromatin condensation at DNA double-strand breaks (DSBs) in human cells. R-loops commonly occur in the genome and are known to influence important biological processes such as transcription termination and chromatin compaction, but their possible role in chromatin regulation at DSBs had not been investigated. In the present study, Liu et al. used the S9.6 antibody against DNA:RNA hybrids to immunoprecipitate proteins that bind to the hybrids and identified two subunits of the NuRD complex, GATAD2B and MBD3. Following this initial observation, the authors have thoroughly studied the recruitment of GATAD2B to DSBs, its role in promoting HDAC1-mediated histone deacetylation and preventing chromatin relaxation, and its involvement in the regulation of DNA end resection.

The model presented by the authors is that R-loops formed at DNA DSBs recruit the NuRD complex, which leads to the establishment of a chromatin domain characterized by reduced levels of H4K21 acetylation that restricts DNA end resection and in this way prevents mutagenic repair reactions. The topic is timely and very interesting. This is a thorough and well done study that makes use of a diversity of relevant methodologies. The experiments are in general well designed and well presented, including correct controls. The authors show that GATAD2B contributes to chromatin regulation at DSBs and plays an important role in maintaining genome integrity by promoting HR. However, the study of the involvement of R-loops in this process is based on experiments that can be interpreted in alternative ways, as described in more detail below. The authors also propose the establishment of a chromatin boundary

between open and condensed chromatin that promotes resection termination, but this point is not sufficiently supported by the data presented. Moreover, some of the results presented are in conflict with previously published data. In summary, although the study is original, well done and interesting, some of the main conclusions should be better supported experimentally and the points below should be addressed to clarify inconsistencies and increase the quality of the study.

Major concerns

1. One of the major claims of the study is that R-loops formed around DSBs promote the recruitment of the GATAD2B-NuRD complex. One concern is that the S9.6 antibody is not strictly specific for DNA:RNA hybrids: it also recognizes dsRNA, as shown for example by Hartono et al., *J Mol Biol* 2018. The Gullerova lab has previously shown that dsRNAs are produced at DSBs (Burger et al. 2019), which raises the possibility that some of the signals detected are not R-loops but dsRNA. In this context, the limited effect of RNase H in some of the experiments becomes a serious concern. See below, point 3 below.

2. Fig. 2C-D present PLA experiments with anti-gamma-H2AX and either anti-GATAD2B or HDAC1. Changes in PLA signal could be due to changed phosphorylation of gamma-H2AX, not necessarily increased or decreased recruitment of GATAD2B or HDAC1. The authors should show that the levels of gamma-H2AX are not changed in by the different treatments included in this experiment.

3. Fig. 2C (and to a lesser extent also 2D): the S9.6 signal is to a large extent (perhaps 70%?) resistant to RNase H digestion as shown by the comparison +IR and +IR +RNH1. Therefore it is hard to conclude that the effects observed in cells treated with transcription inhibitors reveal a role for R-loops in the recruitment of GATAD2B, as concluded by the authors. Can the authors rule out the involvement of dsRNA? Alternatively, have the authors considered the possibility that dsRNAs are involved in the recruitment of GATAD2B?

4. The authors propose that R-loops formed at DSBs recruit a NuRD complex. However, in the original article from the Legube lab, DRIP-seq signals are observed over a window of similar size in both 4OHT-treated and 4OHT-untreated cells (see Fig 2d in Cohen et al <https://doi.org/10.1038/s41467-018-02894-w>). If the R-loops are already formed in the absence of cut (probably because cut AsiSI sites in DiVA cells are mostly near/at transcription units), what prevents R-loops from recruiting GATAD2B and HDAC1 in the absence of DNA damage?

The authors should also explain why the R-loop occupancy shown in Fig. 3D is so different from that shown in the Cohen et al. paper. Is this a representative example?

5. Fig. 3: the idea of a boundary as proposed by the authors is very appealing. A concern here is

that the boundaries are neither characterized nor discussed in detail, and the data are not fully consistent.

a. The manuscript would benefit from a more detailed description of the boundaries based on the ChIP-seq and DRIP-seq signals, and a model could be presented for the contribution of each protein to the boundary. In the present version, the authors simply state that both HDAC1 and GATAD2B spread across an average window of 2.5 kb, which is not accurate. The plots presented in EV5B show that HDAC1 extends beyond 2.5 kb, while GATAD2B is restricted to a smaller region. Why do they show somewhat different distributions if they are both recruited to DSBs as subunits of a NuRD complex? Which of the two proteins defines the boundary? Or is the boundary defined by the deacetylated H4K12?

b. Quantitative measurements of resection at individual DSBs showing that resection terminates at the limit of the deacetylated region would reinforce the idea of a boundary.

c. The GATAD2B, HDAC1 and H4K12ac profiles around HR sites are similar to those observed at NHEJ sites (compare fig. 4E with EV8C). Is this consistent with the model presented by the authors?

There might be technical reasons that explain the differences and apparent inconsistencies mentioned above. In any case, these questions should be clarified.

6. Fig. 5A-C: These experiments are not easy to follow, mainly because they are poorly described/presented. For example:

a. In the 5A plots, the scales for control and siGATAD2B are different which makes comparisons difficult. It would also help to explain in the figure itself what the red and black lines are.

b. In 5B, it is not clear whether the plot shows measurements of one representative stripe (if so, what are the error bars?) or average of several stripes (if so, how many and how were these selected?).

c. Unclear how the data was processed to produce the box plot shown in 5C.

7. The MNase experiments are very interesting and conclusive. Two additional experiments could be envisioned to strengthen this part of the study:

a. To establish a link between chromatin relaxation and HR, the authors could consider repeating MNase experiments in DIVA cells followed by purification and sequencing of mononucleosome DNA. An enrichment of HR-prone DSBs in the mononucleosomes would provide strong support for the link between R-loop and GATAD2B-mediated chromatin condensation and HR.

b. The authors could repeat the MNase in siGATAD2B-treated cells. According to the model presented by the authors, if transcription is inhibited, DNA:RNA hybrids will not be formed and GATAD2B will not be recruited. Therefore, transcription inhibition per se will induce relaxation but will not affect the result observed in siGATAD2B.

8. The experiment presented in Figure 6B is convincing, but it is important to show the BrdU stainings also for non-irradiated siGATAD2B samples to establish whether the role of GATAD2B

in regulating chromatin compaction is general or related to the DNA damage response. Moreover, to further strengthen the conclusion that GATAD2B contributes to restrict DNA end resection, the authors could carry out qPCR-based measurements of resection using the method described by Zhou et al, doi: 10.1093/nar/gkt1309.

9. The reporter experiments presented in EV13B show that GATAD2B depletion does not inhibit NHEJ. Isn't this result in contradiction with GATAD2B being needed to restrict resection? I would argue that deregulated resection resulting in extended ssDNA tails, as proposed by the authors, would strongly inhibit NHEJ. The authors should clarify this point.

10. The title contains several elements that are problematic:

- a. It is correct that both GATAD2B and MBD3 are components of NuRD, but the experiments presented show the involvement of the individual components, not the requirement of a NuRD complex. Is there enough experimental evidence to rule out the existence of a free GATAD2B pool that could help recruit HDAC1 independently of the NuRD complex?
- b. A second problem with the title is the involvement of R-loops, as discussed in point 3 above.
- c. In its present form, the manuscript shows a region with reduced H4K12ac around DSBs, but does not provide enough information to support the claim that this region is limited by boundaries, as discussed above in point 5.

Minor points

1. In the ChIP-seq data analysis sections, the authors state that they only used one of the replicates for each condition for downstream analysis. Was there any particular reason for using only one instead of averages of all replicates, or selecting high-confidence peaks?
2. The experiment presented in EV2B is important to support the conclusion and could be moved to the main figure.
3. Strictly speaking, the methods used in the paper allow the detection of DNA-RNA hybrids, not necessarily R-loops (including three nucleic acid strands). For the sake of accuracy, the authors should revise the text and refer to DNA:RNA hybrids instead of R-loops when relevant.
4. Fig. 1: why did the authors use to irradiate the cells with 10 Gy? This is a very high dose. Most human cell lines have LD50s around 3-5 Gy.
5. The lettering in some of the figure panels is too small (for ex, it is almost impossible to read the x and y axes of the heat maps).
6. Mass spectrometry, PCR-based analysis of copy number and quantitative analysis of COMET tails must be described in the Methods section. This information is totally missing in the present version of the manuscript.
7. EV5A: Not all the samples are visible in the PCA plot, probably because they are on top of each other. Would it be possible to choose a different format to better show the result?
8. In general, the information provided in figure legends is scarce. For widely used methodologies this may not be a problem, but the lack of details makes interpretation difficult in the case of more specialized approaches, as described above for Fig. 5B-C. The experiments

should be better described so that the reader can understand what has been done. Among other things, figure legends should indicate the number of replicates for each experiment and whether the replicates are biological or technical.

9. On page 6, the authors claim that "the subunits of the GATAD2B-NuRD complex preferentially bind R-loops upon DNA damage in PARP1 dependent manner". However, the effect of a PARP1i is only studied for GATAD2B (EV2B). Either the remaining subunits should be tested (at least HDAC1 and MBD3) or the conclusion restricted to the tested subunit.

10. In Fig. 6D, the authors conclude that the CHIP-seq coverage for GATAD2B and HDAC1 is proportional to RAD51 coverage, but they do not provide any quantitative data in support of this statement. A correlation analysis of CHIP-seq coverage should be relatively easy to perform and would provide a more accurate description of the relationships among these proteins at DSBs.

11. The quantification of gamma-H2AX foci is missing in EV13D (RNase H1).

12. In the drawing presented in Fig. 7, NuRD binds to the ssDNA in the R-loop, which is not in agreement with the data.

13. Figure EV8B-C are not mentioned in the text.

14. A few typos/language issues:

- p. 5, line 5 from the top: "...hyper-relaxation and the consequently to DNA ... "
- p. 6, line 3 from the bottom: "... GFP-GATAD2B and GFP-MBD3 cells respectively,....."
- p. 8, line 2 at the top, I would say "producing DSBs" instead of "simulating double strand breaks"
- p.11, line 7 from the top: "... complex and could affect chromatin ..."
- Legend to Fig. 5B is written as 5D.
- The legend to fig. 3F is not correct. The low transcription is shown in a EV figure instead.
- I am not a native speaker, but I believe that some parts of the text would benefit from language editing to ensure correct use of articles.

Referee #3 (Report for Author)

In this study, Liu et al. investigate the mechanisms that regulate open and condensed chromatin architecture at DNA double-strand breaks (DSBs). They observe an association of the GATAD2B-NuRD complex with DSBs, dependent on active transcription and the formation of R-loops-non-canonical nucleic acid structures that form co-transcriptionally. This association promotes histone deacetylation and chromatin condensation at the DNA break, regulating the transition between open and closed chromatin. The authors demonstrate that this transition is necessary to restrict DNA end resection. The lack of the GATAD2B-NuRD complex causes hyper-relaxation of chromatin and extended DNA end resection, leading to a failure in repairing DSBs by homologous recombination (HR). Overall, this study presents a compelling model that links transcription, R-loop formation, chromatin dynamics, and DNA end resection, shedding light on a novel mechanism operating during DSB repair. I have just a few comments that could contribute to further strengthening this excellent piece of work.

1 - Cytoplasmic foci are visible in some PLA images obtained with the S9.6 Ab (e.g. Fig 1F, EV2...). How abundant are they and what is their significance? The authors should consider presenting PLA images that encompass more than one cell for clarity.

2 - The authors should include a control experiment, such as gammaH2AX immunofluorescence, to demonstrate that laser irradiation indeed induces DSBs along the stripe, providing additional support for the study's findings.

3 - Some of the original literature supporting the rationale of the present study are not properly cited (e.g. not all papers originally showing a link between active transcription and the DSB repair pathways - a central aspect of this study - are cited. Same for the evidence that transcription initiates at DSBs).

4 - The observation that RNaseH overexpression significantly increases MNase accessibility raises the hypothesis that MNase might encounter difficulty digesting DNA engaged in the DNA:RNA hybrid moiety. Could this be the case? Additional clarification on this point would strengthen the study's findings.

Dear Hartmut,

Thank you for handling our submission. I appreciate your concerns. However, we are confident that we can address all reviewer's comments in satisfactory way. Please see attached revision proposal.

In particular:

1. unresolved discrepancies with published literature:

Most of the published literature focuses on CHD4 and CHD4-NuRD complex and their roles in HR. In contrast, we investigate the role of the GATADA2B-NuRD complex in DNA repair. We propose that these two complexes work independently and consecutively at DSBs. Furthermore, it has been suggested that CHD4 can function independently of NuRD complex (PMID 23697937).

We will support our hypothesis further with additional data, specifically demonstrating the interaction between GATAD2B and other subunits of the NuRD complex (please see attached revision proposal).

2. possibility of alternative explanations

We will address this with additional data summarised in attached revision proposal.

3. uncertainty about bona fide core NuRD and R-loop roles

Additional data will clarify this point.

4. over-reliance on, or insufficient control of, potentially artifact-prone experimental approaches

We appreciate concerns about PLA approach. However, we always provide single antibody controls and additional experimental approaches: PLA between GATAD2B/MBD3 and S9.6 supports our mass spectroscopy data. PLA between GATAD2B/MBD3 and gH2AX support laser stripping and CHIP-seq. PLA between pan-H4ace and gH2AX is supported by CHIP-seq, and CHIP-qPCR.

PLA is widely used by many scientist across the world.

I hope that you will find the attached revision proposal useful for your final decision and that you will support the revision of our study for the EMBO journal.

Please do not hesitate to contact me if you have any further concerns.

Kind Regards

Monika

Dr Monika Gullerova

monika.gullerova@path.ox.ac.uk

Cancer Research UK Senior Research Fellow
Professor of Molecular Medicine
Lee Placito Fellow in Medicine at Wadham College
Sir William Dunn School of Pathology
University of Oxford
South Parks Road
Oxford, OX1 3RE, UK
Tel.: 0044 1865 285658

Referee #1 (Report for Author)

The manuscript from Liu et al addresses a potential role for the GATAD2B containing NuRD complex in DNA repair. They use a proteomics-based approach to identify GATAD2B and MBD3 in pull-down samples using an antibody (S9.6) that can recognise RNA:DNA hybrids as well as dsRNA. They show that GATAD2B is recruited to DNA breaks and this is impaired when transcription is inhibited or RNaseH1 is overexpressed, suggesting that recruitment is driven by R loops. They show that HDAC1 is recruited to DNA breaks and this is reduced when GATAD2B is depleted. They perform ChIP-seq of GATAD2B and HDAC1 and show that the proteins accumulate at DNA breaks and that there is a relationship with actively transcribed genes, which are preferentially repaired by HR. Consistent with this, there is an HR defect in GATAD2B depleted cells. They use existing datasets to explore a potential relationship with R loops and H4K12ac. They use PLA assays to suggest that depletion of GATAD2B or MBD3 leads to increased H3 and H4 acetylation around DNA breaks. They then look at chromatin relaxation after DNA damage using microscopy based approaches and conclude that GATAD2B prevents relaxation. Finally, they test the effect of GATAD2B or MBD3 depletion on end resection and show that there is an increase in phosphorylated RPA foci, suggesting that these proteins normally function to impede resection at DNA breaks.

This work is consistent with some previously published reports implicating the NuRD complex in DNA double strand break responses: it has been shown that NuRD is recruited to DNA breaks and promotes repair by HR. The work here potentially extends the understanding of how NuRD functions by showing a dependence on transcription and R loops for recruitment, although how this fits with previous data showing recruitment via other mechanisms is not clear.

Importantly, however, some of the data presented here directly contrast with existing literature. Specifically, there is data to suggest that the NuRD complex promotes resection (PMID: 22219182, PMID: 26546801) and promotes chromatin relaxation (as mentioned in the discussion).

In PMID22219182, depletion of CHD4 impairs phosphorylation of RPA and recruitment of BRCA1 and BRIT1 to DSBs.

In PMID: 26546801 depletion of CDH4 impairs HR (reporters) and recruitment of RPA to DSBs. Neither of these two publications investigate GATAD2B nor show direct experimental evidence for the role of CHD4 in end resection termination.

In addition, previous work showed that there is a role for NuRD in regulating DNA damage signaling and checkpoint responses (for example, PMID: 20805324 and 20805320 and 20693977). These functions can indirectly impact on downstream responses, making interpretation of some of the data presented here difficult and limits the impact of the findings.

In PMID: 20805324 CHD4 becomes transiently immobilized on chromatin after IR. Knockdown of CHD4 triggers enhanced Cdc25A degradation and p21(Cip1) accumulation, which lead to more pronounced cyclin-dependent kinase inhibition and extended cell cycle delay. At DNA double-strand breaks, depletion of CHD4 disrupts the chromatin response at the level of the RNF168 ubiquitin ligase, which in turn impairs local ubiquitylation and BRCA1 assembly.

In PMID 20805320 CHD4 stimulates RNF8/RNF168-dependent formation of ubiquitin conjugates to facilitate the accrual of RNF168 and BRCA1. Finally, we show that CHD4 promotes DSB repair and checkpoint activation in response to IR.

There are a few potential reasons for the conflicting data. One is that GATAD2B is not functioning in the context of NuRD in these activities (most - but not all- of the literature focuses on CHD4 or HDAC subunits of NuRD). It is worth noting that CHD4 was not present in the irradiated R loop associated factors (in fact, appeared to be associated in unirradiated controls), and no other NuRD subunits were obviously present in these data.

Indeed, we propose that GATAD2B functions in a specific NuRD complex, which is different from CHD4 containing NuRD complex as characterised above.

We will perform PLA between GATAD2B and CHD4 to support our model.

Furthermore, it has been suggested that CHD4 can function independently of NuRD complex (PMID 23697937).

Other possibilities include experimental design and technical issues. While the discussion mentions the relaxation issue, the explanation doesn't fit well with the data, and it is important to address this experimentally before this study is published. In addition, the differences in resection with previous reports is not well addressed in the discussion.

Major issues:

1. To address the possibility that the data presented here are specific to the GATAD2B and/or MBD3 proteins, depletion of CHD4 should be performed in parallel for key experiments. Particularly the experiments that contradict previous reports - RPA foci and changes in chromatin compaction.

We will include CHD4 KD controls for RPA foci and chromatin compaction.

2. The impact of NuRD on DNA damage responses includes changes in checkpoint responses and signalling, so timepoints become important for these assays. A FACS profile should be included for the time/doses used to show that depletion of GATAD2B or MBD3 isn't indirectly affecting these responses through changes in cell cycle profile.

We will include FACS profile for GATAD3B and MBD3 KD for conditions used.

3. The manuscript relies heavily on PLA, which is notorious for artefacts. One major issue is apparent co-localization after irradiation with gH2AX, when this will occur by chance with any abundant chromatin protein when doses are high. At the very least, the number of gH2AX foci present in these cells at the indicated time/doses should be reported (and representative images shown). Ideally, additional controls (eg an abundant chromatin protein that is not involved in these responses) could be added.

We will quantify gH2AX foci for conditions used and include PLA using antibodies against gH2AX and selected splicing factor (to be selected based on available antibodies suitable for PLA) in all condition tested, including GATAD2B and MBD3 KD, treatments with transcription inhibitors such as triptolide and DRB and RNase H overexpression.

4. The raw mass spec data should be provided.

The mass spectrometry proteomics raw data have been deposited to the ProteomeXchange Consortium via the PRIDE partner repository with the dataset identifier PXD042271".

Data can be accessed via reviewer account details: Username: reviewer_pxd042271@ebi.ac.uk Password: r9fVIgEC.

This is included in Data availability section. Both raw and processed data were deposited.

5. To really conclude that these proteins are important for resection termination, a time course and/or measurement of resected chromatin should be performed. The number of RPA or BrdU foci could reflect differences in resolution and rates of repair, and give no direct readout of resection distance.

We will employ endonuclease-based end resection assay (used before for example Zhou et al, doi: 10.1093/nar/gkt1309 and suggested by reviewer 2) and RPA chromatin immunoprecipitation.

Minor issues:

1. The discussion should be expanded to include more of the existing literature on this complex.

2. The figure legends should have more information. For example, it would be very helpful to know what time points and IR doses were used in the immunofluorescence experiments in Figure 6 (this is true for all the legends in the manuscript).

All of these will be addressed.

Referee #2 (Report for Author)

Liu et al. have studied the regulation of chromatin condensation at DNA double-strand breaks (DSBs) in human cells. R-loops commonly occur in the genome and are known to influence important biological processes such as transcription termination and chromatin compaction, but their possible role in chromatin regulation at DSBs had not been investigated. In the present study, Liu et al. used the S9.6 antibody against DNA:RNA hybrids to immunoprecipitate proteins that bind to the hybrids and identified two subunits of the NuRD complex, GATAD2B and MBD3. Following this initial observation, the authors have thoroughly studied the recruitment of GATAD2B to DSBs, its role in promoting HDAC1-mediated histone deacetylation and preventing chromatin relaxation, and its involvement in the regulation of DNA end resection.

The model presented by the authors is that R-loops formed at DNA DSBs recruit the NuRD complex, which leads to the establishment of a chromatin domain characterized by reduced levels of H4K21 acetylation that restricts DNA end resection and in this way prevents mutagenic repair reactions. The topic is timely and very interesting. This is a thorough and well done study that makes use of a diversity of relevant methodologies. The experiments are in general well designed and well presented, including correct controls. The authors show that GATAD2B contributes to chromatin regulation at DSBs and plays an important role in maintaining genome integrity by promoting HR. However, the study of the involvement of R-loops in this process is based on experiments that can be interpreted in alternative ways, as described in more detail below. The authors also propose the establishment of a chromatin boundary between open and condensed chromatin that promotes resection termination, but this point is not sufficiently supported by the data presented. Moreover, some of the results presented are in conflict with previously published data. In summary, although the study is original, well done and interesting, some of the main conclusions should be better supported experimentally and the points below should be addressed to clarify inconsistencies and increase the quality of the study.

We are grateful to this reviewer for overall positive valuation of our work. Below we propose to address specific points in detail:

Major concerns

1. One of the major claims of the study is that R-loops formed around DSBs promote the recruitment of the GATAD2B-NuRD complex. One concern is that the S9.6 antibody is not strictly specific for DNA:RNA hybrids: it also recognizes dsRNA, as shown for example by Hartono et al., J Mol Biol 2018. The Gullerova lab has previously shown that dsRNAs are produced at DSBs (Burger et al. 2019), which raises the possibility that some of the signals detected are not R-loops but dsRNA. In this context, the limited effect of RNase H in some of the experiments becomes a serious concern. See below, point 3 below.

We will repeat experiments showing GATAD2B at DSBs with additional control. Specifically, with treatment with RNases, such as RNase A, T and III, which are used to remove RNA species such as single and double stranded RNA, but not DNA:RNA hybrids.

2. Fig. 2C-D present PLA experiments with anti-gamma-H2AX and either anti-GATAD2B or HDAC1. Changes in PLA signal could be due to changed phosphorylation of gamma-H2AX, not necessarily increased or decreased recruitment of GATAD2B or HDAC1. The authors should show that the levels of gamma-H2AX are not changed in by the different treatments included in this experiment.

We will quantify gH2AX foci for conditions used.

3. Fig. 2C (and to a lesser extent also 2D): the S9.6 signal is to a large extent (perhaps 70%?) resistant to RNase H digestion as shown by the comparison +IR and +IR +RNH1. Therefore it is hard to conclude that the effects observed in cells treated with transcription inhibitors reveal a role for R-loops in the recruitment of GATAD2B, as concluded by the authors. Can the authors rule out the involvement of dsRNA? Alternatively, have the authors considered the possibility that dsRNAs are involved in the recruitment of GATAD2B?

We will repeat experiments showing GATAD2B at DSBs with additional control. Specifically, with treatment with RNases, such as RNase A, T and III, which are used to remove RNA species

such as single and double stranded RNA, but not DNA:RNA hybrids. Alternatively we will also test this experiment in Dicer KD cells.

4. The authors propose that R-loops formed at DSBs recruit a NuRD complex. However, in the original article from the Legube lab, DRIP-seq signals are observed over a window of similar size in both 4OHT-treated and 4OHT-untreated cells (see Fig 2d in Cohen et al <https://doi.org/10.1038/s41467-018-02894-w>). If the R-loops are already formed in the absence of cut (probably because cut AsiSI sites in DiVA cells are mostly near/at transcription units), what prevents R-loops from recruiting GATAD2B and HDAC1 in the absence of DNA damage?

We showed that GATAD2B binding to R-loops is dependent on PARP1 (Figure EV2B).

Therefore, it is possible that GATAD2B-NuRD binding to R-loops associated with DSBs is stabilised by PARP1, which make it specific to damage conditions.

To support this we will add GATAD2B, MBD3 and HDAC1 PLA with gH2AX in cells treated with PARP1 inhibitor.

The authors should also explain why the R-loop occupancy shown in Fig. 3D is so different from that shown in the Cohen et al. paper. Is this a representative example?

We were trying to use examples of DSBs in other than promoter regions (as NuRD is known to bind to promoter regions in non-damage conditions). We will provide additional snapshots of the promoter regions such as:

5. Fig. 3: the idea of a boundary as proposed by the authors is very appealing. A concern here is that the boundaries are neither characterized nor discussed in detail, and the data are not fully consistent.

a. The manuscript would benefit from a more detailed description of the boundaries based on the ChIP-seq and DRIP-seq signals, and a model could be presented for the contribution of each protein to the boundary. In the present version, the authors simply state that both HDAC1 and GATAD2B spread across an average window of 2.5 kb, which is not accurate. The plots presented in EV5B show that HDAC1 extends beyond 2.5 kb, while GATAD2B is restricted to a smaller region. Why do they show somewhat different distributions if they are both recruited to DSBs as subunits of a NuRD complex? Which of the two proteins defines the boundary? Or is the boundary defined by the deacetylated H4K12?

We will provide more detailed explanation of the boundary. It should be noted that the extent of HDAC1 beyond 2.5kb is minimal and could be simply caused by the different affinity of the antibodies. We will provide box plots showing significance of the GATAD2B and HDAC1 within 2.5kb window and beyond.

b. Quantitative measurements of resection at individual DSBs showing that resection terminates at the limit of the deacetylated region would reinforce the idea of a boundary.

We will employ endonuclease-based end resection assay (used before for example Zhou et al, doi: 10.1093/nar/gkt1309 and suggested by reviewer 2) and RPA chromatin immunoprecipitation

at three selected DSBs.

c. The GATAD2B, HDAC1 and H4K12ac profiles around HR sites are similar to those observed at NHEJ sites (compare fig. 4E with EV8C). Is this consistent with the model presented by the authors?

We propose that GATAD2B-NuRD is promoting histone de-acetylation and chromatin compaction. This role is applicable to both HR and NHEJ DSBs. At HR DSBs, this role of NuRD and the boundary also contribute to accurate end resection termination. Hence this is compatible with our model.

There might be technical reasons that explain the differences and apparent inconsistencies mentioned above. In any case, these questions should be clarified.

Please see above.

6. Fig. 5A-C: These experiments are not easy to follow, mainly because they are poorly described/presented. For example:

a. In the 5A plots, the scales for control and siGATAD2B are different which makes comparisons difficult. It would also help to explain in the figure itself what the red and black lines are.

b. In 5B, it is not clear whether the plot shows measurements of one representative stripe (if so, what are the error bars?) or average of several stripes (if so, how many and how were these selected?).

c. Unclear how the data was processed to produce the box plot shown in 5C.

We apologise for this. We will provide detailed description of these data.

7. The MNase experiments are very interesting and conclusive. Two additional experiments could be envisioned to strengthen this part of the study:

a. To establish a link between chromatin relaxation and HR, the authors could consider repeating MNase experiments in DIVA cells followed by purification and sequencing of mononucleosome DNA. An enrichment of HR-prone DSBs in the mononucleosomes would provide strong support for the link between R-loop and GATAD2B-mediated chromatin condensation and HR.

As mentioned above, we propose that GATAD2B-NuRD can function (chromatin condensation) at both types of DSBs, therefore we don't think that this experiment would provide us with meaningful data. Most likely we would detect chromatin from both NHEJ and HR prone DSBs.

b. The authors could repeat the MNase in siGATAD2B-treated cells. According to the model presented by the authors, if transcription is inhibited, DNA:RNA hybrids will not be formed and GATAD2B will not be recruited. Therefore, transcription inhibition per se will induce relaxation but will not affect the result observed in siGATAD2B.

If we understood this comment correctly, we propose to repeat MNase assay in siGATAD2B and siGATAD2B treated with transcription inhibitors (with all controls).

8. The experiment presented in Figure 6B is convincing, but it is important to show the BrdU stainings also for non-irradiated siGATAD2B samples to establish whether the role of GATAD2B in regulating chromatin compaction is general or related to the DNA damage response. Moreover, to further strengthen the conclusion that GATAD2B contributes to restrict DNA end resection, the authors could carry out qPCR-based measurements of resection using the method described by Zhou et al, doi: 10.1093/nar/gkt1309.

We will add BrD staining in non-damage conditions.

We will employ endonuclease-based end resection assay (used before for example Zhou et al, doi: 10.1093/nar/gkt1309 and suggested by reviewer 2) and RPA chromatin immunoprecipitation at three selected DSBs.

9. The reporter experiments presented in EV13B show that GATAD2B depletion does not inhibit NHEJ. Isn't this result in contradiction with GATAD2B being needed to restrict resection? I would

argue that deregulated resection resulting in extended ssDNA tails, as proposed by the authors, would strongly inhibit NHEJ. The authors should clarify this point.

We propose that de-regulated end resection would lead to repair by MMEJ, which would be detected by NHEJ reporter. To clarify this point we will add IF experiment showing XRCC4 foci (with gH2AX) in control and GATAD2B KD cells.

10. The title contains several elements that are problematic:

a. It is correct that both GATAD2B and MBD3 are components of NuRD, but the experiments presented show the involvement of the individual components, not the requirement of a NuRD complex. Is there enough experimental evidence to rule out the existence of a free GATAD2B pool that could help recruit HDAC1 independently of the NuRD complex?

We will perform PLA between GATAD2B and CHD4, MTA and RBBP subunits (subject to available antibodies suitable for PLA) of NuRD complex in non-damage and damage conditions.

b. A second problem with the title is the involvement of R-loops, as discussed in point 3 above. This will be clarified with the experiments proposed above in first 2 points.

c. In its present form, the manuscript shows a region with reduced H4K12ac around DSBs, but does not provide enough information to support the claim that this region is limited by boundaries, as discussed above in point 5.

This will be clarified with the experiments proposed above in point 5.

Minor points

1. In the ChIP-seq data analysis sections, the authors state that they only used one of the replicates for each condition for downstream analysis. Was there any particular reason for using only one instead of averages of all replicates, or selecting high-confidence peaks?

2. The experiment presented in EV2B is important to support the conclusion and could be moved to the main figure.

3. Strictly speaking, the methods used in the paper allow the detection of DNA-RNA hybrids, not necessarily R-loops (including three nucleic acid strands). For the sake of accuracy, the authors should revise the text and refer to DNA:RNA hybrids instead of R-loops when relevant.

4. Fig. 1: why did the authors use to irradiate the cells with 10 Gy? This is a very high dose. Most human cell lines have LD50s around 3-5 Gy.

5. The lettering in some of the figure panels is too small (for ex, it is almost impossible to read the x and y axes of the heat maps).

6. Mass spectrometry, PCR-based analysis of copy number and quantitative analysis of COMET tails must be described in the Methods section. This information is totally missing in the present version of the manuscript.

7. EV5A: Not all the samples are visible in the PCA plot, probably because they are on top of each other. Would it be possible to choose a different format to better show the result?

8. In general, the information provided in figure legends is scarce. For widely used methodologies this may not be a problem, but the lack of details makes interpretation difficult in the case of more specialized approaches, as described above for Fig. 5B-C. The experiments should be better described so that the reader can understand what has been done. Among other things, figure legends should indicate the number of replicates for each experiment and whether the replicates are biological or technical.

9. On page 6, the authors claim that "the subunits of the GATAD2B-NuRD complex preferentially bind R-loops upon DNA damage in PARP1 dependent manner". However, the effect of a PARP1i is only studied for GATAD2B (EV2B). Either the remaining subunits should be tested (at least HDAC1 and MBD3) or the conclusion restricted to the tested subunit.

10. In Fig. 6D, the authors conclude that the ChIP-seq coverage for GATAD2B and HDAC1 is proportional to RAD51 coverage, but they do not provide any quantitative data in support of this statement. A correlation analysis of ChIP-seq coverage should be relatively easy to perform and would provide a more accurate description of the relationships among these proteins at DSBs.

11. The quantification of gamma-H2AX foci is missing in EV13D (RNase H1).

12. In the drawing presented in Fig. 7, NuRD binds to the ssDNA in the R-loop, which is not in agreement with the data.

13. Figure EV8B-C are not mentioned in the text.

14. A few typos/language issues:

- p. 5, line 5 from the top: "...hyper-relaxation and the consequently to DNA ... "
- p. 6, line 3 from the bottom: "... GFP-GATAD2B and GFP-MBD3 cells respectively,....."
- p. 8, line 2 at the top, I would say "producing DSBs" instead of "simulating double strand breaks"
- p.11, line 7 from the top: "... complex and could affect chromatin ..."
- Legend to Fig. 5B is written as 5D.
- The legend to fig. 3F is not correct. The low transcription is shown in a EV figure instead.
- I am not a native speaker, but I believe that some parts of the text would benefit from language editing to ensure correct use of articles.

All minor comments will be addressed.

Referee #3 (Report for Author)

In this study, Liu et al. investigate the mechanisms that regulate open and condensed chromatin architecture at DNA double-strand breaks (DSBs). They observe an association of the GATAD2B-NuRD complex with DSBs, dependent on active transcription and the formation of R-loops-non-canonical nucleic acid structures that form co-transcriptionally. This association promotes histone deacetylation and chromatin condensation at the DNA break, regulating the transition between open and closed chromatin. The authors demonstrate that this transition is necessary to restrict DNA end resection. The lack of the GATAD2B-NuRD complex causes hyper-relaxation of chromatin and extended DNA end resection, leading to a failure in repairing DSBs by homologous recombination (HR). Overall, this study presents a compelling model that links transcription, R-loop formation, chromatin dynamics, and DNA end resection, shedding light on a novel mechanism operating during DSB repair. I have just a few comments that could contribute to further strengthening this excellent piece of work.

We thank this reviewer for such positive valuation of our work.

1 - Cytoplasmic foci are visible in some PLA images obtained with the S9.6 Ab (e.g. Fig 1F, EV2...). How abundant are they and what is their significance? The authors should consider presenting PLA images that encompass more than one cell for clarity.

These barely visible rare cytoplasmic foci and most likely background. We will show more than 1 cell in our PLA images.

2 - The authors should include a control experiment, such as gammaH2AX immunofluorescence, to demonstrate that laser irradiation indeed induces DSBs along the stripe, providing additional support for the study's findings.

We will add this control experiment.

3 - Some of the original literature supporting the rationale of the present study are not properly cited (e.g. not all papers originally showing a link between active transcription and the DSB repair pathways - a central aspect of this study - are cited. Same for the evidence that transcription initiates at DSBs).

We will fix this in the text.

4 - The observation that RNaseH overexpression significantly increases MNase accessibility raises the hypothesis that MNase might encounter difficulty digesting DNA engaged in the DNA:RNA hybrid moiety. Could this be the case? Additional clarification on this point would strengthen the study's findings.

Data from proposed experiments for reviewers 1 and 2 would contribute to clarification of this point.

Prof. Monika Gullerova
University of Oxford
Sir William Dunn School of Pathology
South Parks Road
Oxford OX1 3RE
United Kingdom

19th Dec 2023

Re: EMBOJ-2023-116131
GATAD2B containing NuRD complex drives R-loop dependent chromatin boundary formation at double strand breaks

Dear Monika,

Thank you for sending me the detailed tentative responses and revision plan for your recent EMBO Journal submission on R-loop-dependent GATAD2B-NuRD complex roles at double-strand breaks, which I now had a chance to carefully consider. I am happy to see that you seem to be in a good position to respond to the key concerns raised by the referees, and appreciate that your answers may well clarify most of them. I would therefore invite you to prepare a revised version of the study, modified and extended along the lines suggested in your proposal. Please do keep in mind that in light of our single-major-revision-round policy, it will be important to convince the critical referees with the additional data and clarifications at this stage. Should you need an extended revision period for this, please simply let me know, and as always, competing manuscript published during the course of this revision will not affect our final decision on your study.

Please also note the additional information and more detailed guidelines on how to prepare a revision below (and in our online Guide to Authors) - closely adhering to them shall greatly facilitate the editorial process at the time of resubmission.

Thank you again for the opportunity to consider this work, and I look forward to receiving your revised manuscript in due time.

With kind regards,

Hartmut

9) Digital image enhancement is acceptable practice, as long as it accurately represents the original data and conforms to community standards. If a figure has been subjected to significant electronic manipulation, this must be clearly noted in the figure legend and/or the 'Materials and Methods' section. The editors reserve the right to request original versions of figures and the original images that were used to assemble the figure. Finally, we generally encourage uploading of numerical as well as gel/blot image source data; for details see: embopress.org/page/journal/14602075/authorguide#sourcedata

At EMBO Press, we ask authors to provide source data for the main manuscript figures. Our source data coordinator will contact you to discuss which figure panels we would need source data for and will also provide you with helpful tips on how to upload and organize the files.

In the interest of ensuring the conceptual advance provided by the work, we recommend submitting a revision within 3 months (18th Mar 2024). Please discuss the revision progress ahead of this time with the editor if you require more time to complete the revisions.

Rebuttal letter for EMBOJ-2023-116131

Referee #1 (Report for Author)

The manuscript from Liu et al addresses a potential role for the GATAD2B containing NuRD complex in DNA repair. They use a proteomics-based approach to identify GATAD2B and MBD3 in pull-down samples using an antibody (S9.6) that can recognise RNA:DNA hybrids as well as dsRNA. They show that GATAD2B is recruited to DNA breaks and this is impaired when transcription is inhibited or RNaseH1 is overexpressed, suggesting that recruitment is driven by R loops. They show that HDAC1 is recruited to DNA breaks and this is reduced when GATAD2B is depleted. They perform ChIP-seq of GATAD2B and HDAC1 and show that the proteins accumulate at DNA breaks and that there is a relationship with actively transcribed genes, which are preferentially repaired by HR. Consistent with this, there is an HR defect in GATAD2B depleted cells. They use existing datasets to explore a potential relationship with R loops and H4K12ac. They use PLA assays to suggest that depletion of GATAD2B or MBD3 leads to increased H3 and H4 acetylation around DNA breaks. They then look at chromatin relaxation after DNA damage using microscopy based approaches and conclude that GATAD2B prevents relaxation. Finally, they test the effect of GATAD2B or MBD3 depletion on end resection and show that there is an increase in phosphorylated RPA foci, suggesting that these proteins normally function to impede resection at DNA breaks.

This work is consistent with some previously published reports implicating the NuRD complex in DNA double strand break responses: it has been shown that NuRD is recruited to DNA breaks and promotes repair by HR. The work here potentially extends the understanding of how NuRD functions by showing a dependence on transcription and R loops for recruitment, although how this fits with previous data showing recruitment via other mechanisms is not clear.

We thank this reviewer for their comments and suggestions. Below, individual points are addressed in detail. We hope that providing an explanation of how our work fits within the published literature and additional experimental data will make this manuscript a suitable contribution to the field of the NuRD complex and DNA repair.

Importantly, however, some of the data presented here directly contrast with existing literature. Specifically, there is data to suggest that the NuRD complex promotes resection (PMID: 22219182, PMID: 26546801) and promotes chromatin relaxation (as mentioned in the discussion).

In addition, previous work showed that there is a role for NuRD in regulating DNA damage signaling and checkpoint responses (for example, PMID: 20805324 and 20805320 and 20693977). These functions can indirectly impact on downstream responses, making interpretation of some of the data presented here difficult and limits the impact of the findings. There are a few potential reasons for the conflicting data. One is that GATAD2B is not functioning in the context of NuRD in these activities (most - but not all- of the literature focuses on CHD4 or HDAC subunits of NuRD). It is worth noting that CHD4 was not present in the irradiated R loop associated factors (in fact, appeared to be associated in unirradiated controls), and no other NuRD subunits were obviously present in these data.

The NuRD complex, as highlighted by the studies referenced in PMIDs 22219182 and 26546801, includes the CHD4 subunit, which is instrumental in promoting end resection and chromatin relaxation. These activities are crucial for facilitating the repair of DSBs by repair proteins. Further, studies cited in PMIDs 20805324, 20805320, and 20693977 underscore the role of the NuRD complex, including CHD4, in regulating DNA damage signalling and checkpoint responses. These processes are integral to the initial recognition of DNA damage and precede the mechanical aspects of DNA repair.

The structural architecture of the NuRD complex includes two catalytic diverse groups of enzymes: ATP-dependent chromatin-remodeling enzymes, including CHD3, CHD4, and CHD5(1), and histone deacetylases HDAC1 and HDAC2. Additionally, the NuRD complex incorporates methyl-

CpG-binding domain (MBD)2 or MBD3, GATAD2A and GATAD2B, MTA1, MTA2, and MTA3, along with RBBP4 and RBBP7, and CDK2AP1(1). Various specific NuRD subunits are rapidly recruited to the sites of DNA damage(2-5). GATAD2A and GATAD2B, belonging to the GATA zinc finger domain-containing family, define **specific and mutually exclusive NuRD** sub-complexes that play a crucial role in modulating the chromatin environment surrounding DSBs(6). ZMYND8 was found to be rapidly recruited to DSBs, promoting the efficient recruitment of GATAD2A and CHD4 but not GATAD2B. The recruitment of MBD2 was also dependent on ZMYND8-GATAD2A interaction, suggesting a sequential recruitment model for NuRD subunits.

Furthermore, this reviewer noted that CHD4 was not present in the irradiated R-loop-associated factors and seemed to be associated in un-irradiated controls. We have now further reinforced this observation by directly testing the interaction between GATAD2B and CHD4, demonstrating that they indeed separate upon DNA damage (unlike GATAD2B and MTA1, which was used as a control) (EV Figure 3 D and E).

Therefore, it is crucial to acknowledge the multifaceted nature of the NuRD complex and the diverse functions of its sub-complexes when assessing its impact on DNA repair.

We propose that GATAD2B functions within a distinct NuRD sub-complex, differing from the CHD4-containing NuRD sub-complex as characterized previously. Upon DNA damage, transcription and chromatin relaxation are necessary early steps leading to repair, facilitated by the CHD4-NuRD complex. Eventually, transcription at DSBs and end resection must cease, in a process facilitated by the GATAD2B-NuRD complex. We believe that our findings integrate well with the existing literature and contribute to understanding the sequence of events. This information has been incorporated into the Discussion section."

Other possibilities include experimental design and technical issues. While the discussion mentions the relaxation issue, the explanation doesn't fit well with the data, and it is important to address this experimentally before this study is published. In addition, the differences in resection with previous reports is not well addressed in the discussion.

Major issues:

1. To address the possibility that the data presented here are specific to the GATAD2B and/or MBD3 proteins, depletion of CHD4 should be performed in parallel for key experiments. Particularly the experiments that contradict previous reports - RPA foci and changes in chromatin compaction.

To address this comment, we have performed CHD4 KD and tested for a number of pS4/S8 RPA foci in damage conditions. Depletion of CHD4 led to decreased number of RPA foci (EV Figure 5E). These data further suggest that the CHD4-NuRD and the GATAD2B-NuRD function as separate subcomplexes.

2. The impact of NuRD on DNA damage responses includes changes in checkpoint responses and signalling, so timepoints become important for these assays. A FACS profile should be included for the time/doses used to show that depletion of GATAD2B or MBD3 isn't indirectly affecting these responses through changes in cell cycle profile.

To address this point, we have performed the FACS analysis of GATAD2B and MBD3 depleted cells, harvested 2hrs after 5Gy IR treatment. We did not detect significant effect of GATAD2B or MBD3 depletion on cell cycle. These data are now shown in Supplementary Figure 8D.

3. The manuscript relies heavily on PLA, which is notorious for artefacts. One major issue is apparent co-localization after irradiation with gH2AX, when this will occur by chance with any abundant chromatin protein when doses are high. At the very least, the number of gH2AX foci present in these cells at the indicated time/doses should be reported (and representative images shown). Ideally, additional controls (eg an abundant chromatin protein that is not involved in these responses) could be added.

As suggested, we have performed IF and quantified the number of gH2AX foci in all conditions used, and show now that this is not affected in any of the conditions (siMBD3, siGATAD2B, RNaseH, TPL3 or DRB). See EV Figure 2D.

Additionally, we performed PLA using antibodies against gH2AX and HOXD11, which is a chromatin associated transcription factor not known to function at DSBs, in all condition tested, including GATAD2B and MBD3 KD, treatments with transcription inhibitors such as triptolide and DRB and RNase H overexpression. We failed to detect HOXD11's proximity to gH2AX in any of the conditions tested (EV Figure 2C). These data suggest that GATAD2B recruitment to DSBs is specific.

4. The raw mass spec data should be provided.

The mass spectrometry proteomics raw data have been deposited to the ProteomeXchange Consortium via the PRIDE partner repository with the dataset identifier PXD042271".

Data can be accessed via reviewer account details: Username: reviewer_pxd042271@ebi.ac.uk Password: r9fVIgEC.

This is included in Data availability section. Both raw and processed data were deposited to PRIDE.

5. To really conclude that these proteins are important for resection termination, a time course and/or measurement of resected chromatin should be performed. The number of RPA or BrdU foci could reflect differences in resolution and rates of repair, and give no direct readout of resection distance.

To address this comment, we employed endonuclease-based end resection assay (used before for example Zhou et al, doi: 10.1093/nar/gkt1309 and suggested by reviewer 2) to measure resected chromatin. This assay confirmed that depletion of GATAD2B led to significantly extended resection.

These data are now part of Figure 6E.

Minor issues:

1. The discussion should be expanded to include more of the existing literature on this complex. We expanded discussion accordingly.

2. The figure legends should have more information. For example, it would be very helpful to know what time points and IR doses were used in the immunofluorescence experiments in Figure 6 (this is true for all the legends in the manuscript).

We have revised all the figure legends adding more information.

Referee #2 (Report for Author)

Liu et al. have studied the regulation of chromatin condensation at DNA double-strand breaks (DSBs) in human cells. R-loops commonly occur in the genome and are known to influence important biological processes such as transcription termination and chromatin compaction, but their possible role in chromatin regulation at DSBs had not been investigated. In the present study, Liu et al. used the S9.6 antibody against DNA:RNA hybrids to immunoprecipitate proteins that bind to the hybrids and identified two subunits of the NuRD complex, GATAD2B and MBD3. Following this initial observation, the authors have thoroughly studied the recruitment of GATAD2B to DSBs, its role in promoting HDAC1-mediated histone deacetylation and preventing chromatin relaxation, and its involvement in the regulation of DNA end resection.

The model presented by the authors is that R-loops formed at DNA DSBs recruit the NuRD complex, which leads to the establishment of a chromatin domain characterized by reduced levels of H4K21 acetylation that restricts DNA end resection and in this way prevents mutagenic repair reactions. The topic is timely and very interesting. This is a thorough and well done study that makes use of a diversity of relevant methodologies. The experiments are in general well designed and well presented, including correct controls. The authors show that GATAD2B contributes to chromatin regulation at DSBs and plays an important role in maintaining genome

integrity by promoting HR. However, the study of the involvement of R-loops in this process is based on experiments that can be interpreted in alternative ways, as described in more detail below. The authors also propose the establishment of a chromatin boundary between open and condensed chromatin that promotes resection termination, but this point is not sufficiently supported by the data presented. Moreover, some of the results presented are in conflict with previously published data. In summary, although the study is original, well done and interesting, some of the main conclusions should be better supported experimentally and the points below should be addressed to clarify inconsistencies and increase the quality of the study.

We are grateful to this reviewer for overall positive valuation of our work. Below we address their specific points in detail.

Major concerns

1. One of the major claims of the study is that R-loops formed around DSBs promote the recruitment of the GATAD2B-NuRD complex. One concern is that the S9.6 antibody is not strictly specific for DNA:RNA hybrids: it also recognizes dsRNA, as shown for example by Hartono et al., J Mol Biol 2018. The Gullerova lab has previously shown that dsRNAs are produced at DSBs (Burger et al. 2019), which raises the possibility that some of the signals detected are not R-loops but dsRNA. In this context, the limited effect of RNase H in some of the experiments becomes a serious concern. See below, point 3 below.

We agree with this valid point. A recent report suggested that S9.6 antibody has robust specificity for DNA:RNA hybrids over dsRNA(7).

However, to further address concerns regarding the specificity of the S9.6 antibody for DNA:RNA hybrids versus dsRNA, we conducted additional controls within our GATAD2B/ γ H2AX PLA assays. We specifically utilized RNase III treatment, which selectively cleaves dsRNA without affecting DNA:RNA hybrids. Should dsRNA play a role in GATAD2B's recruitment to DSBs, we would anticipate observing diminished GATAD2B at DSBs post-RNase III treatment. Additionally, we conducted experiments in cells depleted of Dicer, a condition under which endogenously produced dsRNA would be expected to accumulate. If dsRNA contributes to GATAD2B recruitment to DSBs, an increased presence of GATAD2B at DSBs in this context would be predicted. As an additional control, we employed PARP1 inhibitors, given our previous demonstration that PARP1 is necessary for GATAD2B's association with S9.6 (Figure 1G) and for the recruitment of MBD3 and HDAC1 to DSBs (EV Figure 3B and C). Our experiments showed that neither RNase III treatment nor Dicer depletion impacted GATAD2B recruitment to DSBs. Conversely, inhibiting PARP1 led to a reduced number of GATAD2B/ γ H2AX PLA foci. These new results are now presented in EV Figure 3A."

2. Fig. 2C-D present PLA experiments with anti- γ H2AX and either anti-GATAD2B or HDAC1. Changes in PLA signal could be due to changed phosphorylation of γ H2AX, not necessarily increased or decreased recruitment of GATAD2B or HDAC1. The authors should show that the levels of γ H2AX are not changed in by the different treatments included in this experiment.

We appreciate this comment, it is a useful control. We have now performed γ H2AX IF in all conditions used and show that the levels of γ H2AX remain the same across all conditions tested. These new data are now presented in EV Figure 2D.

3. Fig. 2C (and to a lesser extent also 2D): the S9.6 signal is to a large extent (perhaps 70%?) resistant to RNase H digestion as shown by the comparison +IR and +IR +RNH1. Therefore it is hard to conclude that the effects observed in cells treated with transcription inhibitors reveal a role for R-loops in the recruitment of GATAD2B, as concluded by the authors. Can the authors rule out the involvement of dsRNA? Alternatively, have the authors considered the possibility that dsRNAs are involved in the recruitment of GATAD2B?

Please see the reply to point 1. The residual S9.6 signal in cells overexpressing RNaseH1 might be caused by their incomplete resolution. It is plausible that DNA:RNA hybrids associated with DSBs might be less accessible due to various proteins binding to them. The effect of RNaseH1 on GATAD2B or HDAC1 recruitment to DSBs, is regardless significant.

4. The authors propose that R-loops formed at DSBs recruit a NuRD complex. However, in the original article from the Legube lab, DRIP-seq signals are observed over a window of similar size in both 4OHT-treated and 4OHT-untreated cells (see Fig 2d in Cohen et al <https://doi.org/10.1038/s41467-018-02894-w>). If the R-loops are already formed in the absence of cut (probably because cut AsiSI sites in DiVA cells are mostly near/at transcription units), what prevents R-loops from recruiting GATAD2B and HDAC1 in the absence of DNA damage?

This is an interesting point. It is correct that most of the AsiSI cut sites are located near promoters of protein coding genes(8). Also R-loops are associated with promoters of genes in non-damage conditions(9). We have shown that GATAD2B binding to R-loops (Figure 1G) and its proximity to gH2AX (EV Figure 3A, B and C) is PARP1-dependent.

PARP1 has been implicated in NuRD recruitment to DSBs before(10). PARP1 can also directly bind to R-loops(11,12) and PARP1 is recruited to DSBs(13). Therefore, it is plausible that PARP1, associated with DSBs contributes to definition of R-loops that NuRD is recruited to.

The authors should also explain why the R-loop occupancy shown in Fig. 3D is so different from that shown in the Cohen et al. paper. Is this a representative example?

Cohen et al., used snapshots of AsiSI sites localised in promoters (Cohen et al., Fig 2a). Indeed, they did detect some DRIP signal in -4OHT samples, but this was significantly increased in +4OHT samples.

Interestingly, for their manual validation in Fig. 2e, they chose a site where DRIP signal in -4OHT is minimal.

We used examples of DSBs in other than promoter regions (as NuRD is known to bind to promoter regions in non-damage conditions).

For consistency, we added additional snapshots of the AsiSI in promoter regions to Figure 3D.

5. Fig. 3: the idea of a boundary as proposed by the authors is very appealing. A concern here is that the boundaries are neither characterized nor discussed in detail, and the data are not fully consistent.

a. The manuscript would benefit from a more detailed description of the boundaries based on the CHIP-seq and DRIP-seq signals, and a model could be presented for the contribution of each protein to the boundary. In the present version, the authors simply state that both HDAC1 and GATAD2B spread across an average window of 2.5 kb, which is not accurate. The plots presented in EV5B show that HDAC1 extends beyond 2.5 kb, while GATAD2B is restricted to a smaller region. Why do they show somewhat different distributions if they are both recruited to DSBs as subunits of a NuRD complex? Which of the two proteins defines the boundary? Or is the boundary defined by the deacetylated H4K12?

The NuRD-mediated boundary exhibits spatiotemporal characteristics. We observed significant damage-induced enrichment of GATAD2B and HDAC1 within the ± 2.5 kb window surrounding DSBs. Addressing the reviewer's comments regarding potential extension beyond this boundary, we analyzed the distribution of HDAC1 and GATAD2B in 2.5 kb intervals radiating from the DSB. Our findings indicate that both HDAC1 and GATAD2B show a significant increase in enrichment from the 5 kb to 2.5 kb region away from the DSB upon damage, though not as pronounced as within the immediate 0 to 2.5 kb vicinity of the DSB (see Fig S2E). Given this pattern for both HDAC1 and GATAD2B, we suggest that they are co-recruited as NuRD complex constituents and are predominantly concentrated within the 2.5 kb area nearest to the DSBs, forming what we term the chromatin 'boundary'.

It is important to recognize that these metagene profiles reflect the average across 80 sites; therefore, the observed enrichment of GATAD2B or HDAC1 beyond the average boundary may arise from a limited number of locations. We also anticipate that the boundary is not an absolute demarcation but rather a zone within the chromatin. This spatial boundary is associated with a transition in H4K12Ac occupancy, showing a negative correlation within the ± 2.5 kb range. The formation of the GATAD2B-NuRD boundary is sequential, following chromatin relaxation and transcription. R-loops, established behind paused RNAPII, provide a scaffold for GATAD2B-NuRD to facilitate transcription termination, end resection, and chromatin compaction. We have incorporated a more comprehensive description of the boundary concept into the Discussion section.

b. Quantitative measurements of resection at individual DSBs showing that resection terminates at the limit of the deacetylated region would reinforce the idea of a boundary.

To address this comment, we employed endonuclease-based end resection assay (used before for example Zhou et al, doi: 10.1093/nar/gkt1309) to measure resected chromatin before and after the boundary. This assay confirmed that depletion of GATAD2B led to significantly extended resection beyond the boundary.

These data are now part of Figure 6E.

c. The GATAD2B, HDAC1 and H4K12ac profiles around HR sites are similar to those observed at NHEJ sites (compare fig. 4E with EV8C). Is this consistent with the model presented by the authors?

We propose that GATAD2B-NuRD is promoting histone de-acetylation and chromatin compaction. This role is applicable to both HR and NHEJ DSBs. At HR DSBs, this role of NuRD and the boundary also contribute to accurate end resection termination. Hence this is compatible with our model.

There might be technical reasons that explain the differences and apparent inconsistencies mentioned above. In any case, these questions should be clarified.

Please see above.

6. Fig. 5A-C: These experiments are not easy to follow, mainly because they are poorly described/presented. For example:

a. In the 5A plots, the scales for control and siGATAD2B are different which makes comparisons difficult. It would also help to explain in the figure itself what the red and black lines are.

We appreciate this comment. The scale is produced by the software in Fiji/ImageJ and is set automatically. We have now added further information for the corresponding Figure legend.

b. In 5B, it is not clear whether the plot shows measurements of one representative stripe (if so, what are the error bars?) or average of several stripes (if so, how many and how were these selected?).

We used at least 10 cells for quantification at indicated time points. This information is now added to the Figure legend.

c. Unclear how the data was processed to produce the box plot shown in 5C.

We now provided this information in corresponding Figure legend.

7. The MNase experiments are very interesting and conclusive. Two additional experiments could be envisioned to strengthen this part of the study:

a. To establish a link between chromatin relaxation and HR, the authors could consider repeating MNase experiments in DIVA cells followed by purification and sequencing of mononucleosome DNA. An enrichment of HR-prone DSBs in the mononucleosomes would provide strong support for the link between R-loop and GATAD2B-mediated chromatin condensation and HR.

As previously stated, we suggest that GATAD2B-NuRD is involved in chromatin condensation associated with both non-homologous end joining (NHEJ) and homologous recombination (HR) repair pathways at DNA double-strand breaks (DSBs). It is probable that chromatin associated with both NHEJ and HR-prone DSBs would be detected in our assays. After thorough consideration of the scope of new experiments conducted, technical challenge and "value for money" we came to conclusion that prioritizing other controls requested here, was of greater importance. Please see below.

b. The authors could repeat the MNase in siGATAD2B-treated cells. According to the model presented by the authors, if transcription is inhibited, DNA:RNA hybrids will not be formed and GATAD2B will not be recruited. Therefore, transcription inhibition per se will induce relaxation but will not affect the result observed in siGATAD2B.

This is an interesting point. As suggested, we performed MNase assay in control cells, cells treated with triptolide, cells depleted of GATAD2B and in cells with combined treatment with triptolide and depleted of GATAD2B. All of these in no damage and damage conditions. We observed chromatin relaxation in IR treated cells, which was further enhanced by triptolide treatment and depletion of GATAD2B. Combined treatment with triptolide and siGATAD2B did not further increase chromatin relaxation above the one detected in cells depleted of GATAD2B. These data are now shown in EV Figure 5C.

8. The experiment presented in Figure 6B is convincing, but it is important to show the BrdU stainings also for non-irradiated siGATAD2B samples to establish whether the role of GATAD2B in regulating chromatin compaction is general or related to the DNA damage response.

To address this point, we performed BrD staining in non-damage conditions. Depletion of GATAD2B or MBD3 or overexpression of RNaseH1 did not have any effect on BrD levels in no damage conditions (EV Figure 5G). These data further support the idea that GATAD2B-NuRD regulates chromatin compaction upon DNA damage.

Moreover, to further strengthen the conclusion that GATAD2B contributes to restrict DNA end resection, the authors could carry out qPCR-based measurements of resection using the method described by Zhou et al, doi: 10.1093/nar/gkt1309.

To address this comment, we employed endonuclease-based end resection assay (used before Zhou et al, doi: 10.1093/nar/gkt1309) to measure resected chromatin before and after the boundary. This assay confirmed that depletion of GATAD2B led to significantly extended resection beyond the boundary.

These data are now part of Figure 6E.

9. The reporter experiments presented in EV13B show that GATAD2B depletion does not inhibit NHEJ. Isn't this result in contradiction with GATAD2B being needed to restrict resection? I would argue that deregulated resection resulting in extended ssDNA tails, as proposed by the authors, would strongly inhibit NHEJ. The authors should clarify this point.

We propose that de-regulated end resection would lead to repair by MMEJ, which would be detected by NHEJ reporter and hence would mask the effect of GATAD2B depletion on NHEJ. To clarify this point, we now added IF experiment showing 53BP1 foci (with gH2AX) in control and GATAD2B KD cells. Depletion of GATAD2B impaired 53BP1 foci formation and hence we conclude that GATAD2B-NuRD can act within both repair pathways. These data are now shown in Supplementary Figure 8C.

10. The title contains several elements that are problematic:

a. It is correct that both GATAD2B and MBD3 are components of NuRD, but the experiments presented show the involvement of the individual components, not the requirement of a NuRD complex. Is there enough experimental evidence to rule out the existence of a free GATAD2B pool that could help recruit HDAC1 independently of the NuRD complex?

To address this point, we performed PLA between GATAD2B and CHD4 and MTA subunits of NuRD complex in non-damage and damage conditions. We found that GATAD2B and CHD4 interact with each other less upon DNA damage, whilst the proximity of GATAD2B to MTA1 is increased upon DNA damage. These data (EV Figure 3D and E) further support the hypothesis that GATAD2B functions within specific NuRD complex.

b. A second problem with the title is the involvement of R-loops, as discussed in point 3 above. We addressed this issue in point 1.

c. In its present form, the manuscript shows a region with reduced H4K12ac around DSBs, but does not provide enough information to support the claim that this region is limited by boundaries, as discussed above in point 5.

We clarified this issue in point 5.

Minor points

1. In the ChIP-seq data analysis sections, the authors state that they only used one of the replicates for each condition for downstream analysis. Was there any particular reason for using only one instead of averages of all replicates, or selecting high-confidence peaks?

We thank this reviewer for this point. Generally, both ways, merging or analysing individual repeats of ChIP-seq can be found across published literature with no obvious preference for neither of them.

We chose not to merge biological replicates for two reasons. First reason is to evaluate our hypothesis independently in all replicates. Quantitative analysis of metagene profile of HDAC1 and GATAD2B in the other two replicates also showed significant increase in coverage in 2.5kb flank region of BLESS 80 sites upon tamoxifen treatment. We have now added these data to Figure S2 B and C. Second reason was that PCA plot (Fig. S2A) of all significant peaks showed clustering of one of HDAC1 replicates with input sample. However, this did not impact our downstream analysis since we focussed on the BLESS 80 sites from two reproducible repeats, where the input did not show increase in enrichment upon damage induction using tamoxifen treatment (Fig. S2D). Furthermore, we were able to validate our hypothesis in all three replicates showing that there is statistically significant enrichment of NuRD complex near DSBs upon cleavage induction using tamoxifen. We were mostly interested in 2.5kb flanks surrounding annotated BLESS 80 sites and directly compared the coverage of GATAD2B/HDAC1 in those

regions between damage (+4OHT) and no damage sample (-4OHT). Same approach was also employed in previous studies (14) where they used AsiSI model systems and ChIP-Seq to explore histone modifications and enrichment of DNA repair protein near annotated DSBs.

2. The experiment presented in EV2B is important to support the conclusion and could be moved to the main figure.

We have moved these data to main Figure 1G.

3. Strictly speaking, the methods used in the paper allow the detection of DNA-RNA hybrids, not necessarily R-loops (including three nucleic acid strands). For the sake of accuracy, the authors should revise the text and refer to DNA:RNA hybrids instead of R-loops when relevant.

We appreciate this comment. It is really hard to distinguish between DNA:RNA hybrids and R-loops as two most common approaches for their detection/resolution, S9.6 antibody and RNaseH1 overexpression, are relevant to both structures. Furthermore, DNA:RNA hybrids and R-loops and used interchangeably in many publications in this field. One such example can be this study, showing the specificity of S9.6 antibody to various substrates(7).

However, we have changed R-loops to DNA:RNA hybrids were relevant throughout the text.

4. Fig. 1: why did the authors use to irradiate the cells with 10 Gy? This is a very high dose. Most human cell lines have LD50s around 3-5 Gy.

For most of our experiments we used 5Gy. Only for the mass spectroscopy we used 10Gy. As we collected samples shortly after DNA damage induction, we wanted to induce significant amount of damage to increase the detection of as many S9.6 proteins as possible. As we selected GATAD2B and MBD3 for further analysis, in various experimental conditions these were validated using 5Gy.

5. The lettering in some of the figure panels is too small (for ex, it is almost impossible to read the x and y axes of the heat maps).

We apologise for these. We made them bigger.

6. Mass spectrometry, PCR-based analysis of copy number and quantitative analysis of COMET tails must be described in the Methods section. This information is totally missing in the present version of the manuscript.

We apologise for this. Description of these methods was added to the Methods section.

7. EV5A: Not all the samples are visible in the PCA plot, probably because they are on top of each other. Would it be possible to choose a different format to better show the result?

We thank the reviewer for raising a valid concern about the readability of PCA plot. The PCA plot was generated using CHIPQC R package, unfortunately this package does not include parameters to change format of the image. CHIPQC also does not output the raw files from where one could generate the PCA plots separately, hence we are sorry that the plot cannot be improved. However, we now provide a more detailed description of the PCA plot as part of the Figure legend, to guide the readers: "HDAC1: Damage replicates are all clustered together appearing as a single black point. The Control:No Damage i.e Input samples from no damage are also clustered together appearing as a single orange point. One of the HDAC1:No Damage clustered with the Control:No Damage samples, see orange and pink points close together".

We hope this provides more clarifications.

8. In general, the information provided in figure legends is scarce. For widely used methodologies this may not be a problem, but the lack of details makes interpretation difficult in the case of more specialized approaches, as described above for Fig. 5B-C. The experiments should be better described so that the reader can understand what has been done. Among other things, figure legends should indicate the number of replicates for each experiment and whether the replicates are biological or technical.

We have amended Figure legends and provided more information.

9. On page 6, the authors claim that "the subunits of the GATAD2B-NuRD complex preferentially

bind R-loops upon DNA damage in PARP1 dependent manner". However, the effect of a PARP1i is only studied for GATAD2B (EV2B). Either the remaining subunits should be tested (at least HDAC1 and MBD3) or the conclusion restricted to the tested subunit.

We have changed this sentence and refer only to tested subunits.

10. In Fig. 6D, the authors conclude that the ChIP-seq coverage for GATAD2B and HDAC1 is proportional to RAD51 coverage, but they do not provide any quantitative data in support of this statement. A correlation analysis of ChIP-seq coverage should be relatively easy to perform and would provide a more accurate description of the relationships among these proteins at DSBs. We thank the reviewer for suggesting correlation analysis to check the veracity of our hypothesis. As per the suggestion, we compared the GATAD2B and HDAC1 coverage to RAD51 coverage in 2.5kb flanking region of BLESS 80 sites upon tamoxifen treatment. Pearson's Correlation test shows significant positive correlation ($p > 0.5$, P val < 0.001) between GATAD2B and RAD51 as well as HDAC1 and RAD51 as represented in the scatter plot (Fig S7G). Overall, we can conclude that GATAD2B and HDAC1 coverage are proportional to RAD51 coverage near DSBs.

11. The quantification of gamma-H2AX foci is missing in EV13D (RNase H1).

This has been added now. See new Supplementary Figure 8F.

12. In the drawing presented in Fig. 7, NuRD binds to the ssDNA in the R-loop, which is not in agreement with the data.

This has been amended.

13. Figure EV8B-C are not mentioned in the text.

This has been fixed.

14. A few typos/language issues:

- p. 5, line 5 from the top: "...hyper-relaxation and the consequently to DNA ... "
- p. 6, line 3 from the bottom: "... GFP-GATAD2B and GFP-MBD3 cells respectively,....."
- p. 8, line 2 at the top, I would say "producing DSBs" instead of "simulating double strand breaks"
- p.11, line 7 from the top: "... complex and could affect chromatin ..."
- Legend to Fig. 5B is written as 5D.
- The legend to fig. 3F is not correct. The low transcription is shown in a EV figure instead.
- I am not a native speaker, but I believe that some parts of the text would benefit from language editing to ensure correct use of articles.

We would like to thank the reviewer for highlighting these points. All minor comments have been addressed, and we have revised the entire manuscript accordingly.

Referee #3 (Report for Author)

In this study, Liu et al. investigate the mechanisms that regulate open and condensed chromatin architecture at DNA double-strand breaks (DSBs). They observe an association of the GATAD2B-NuRD complex with DSBs, dependent on active transcription and the formation of R-loops-non-canonical nucleic acid structures that form co-transcriptionally. This association promotes histone deacetylation and chromatin condensation at the DNA break, regulating the transition between open and closed chromatin. The authors demonstrate that this transition is necessary to restrict DNA end resection. The lack of the GATAD2B-NuRD complex causes hyper-relaxation of chromatin and extended DNA end resection, leading to a failure in repairing DSBs by homologous recombination (HR). Overall, this study presents a compelling model that links transcription, R-loop formation, chromatin dynamics, and DNA end resection, shedding light on a novel mechanism operating during DSB repair. I have just a few comments that could contribute to further strengthening this excellent piece of work.

We would like to thank this reviewer for such positive valuation of our work.

1 - Cytoplasmic foci are visible in some PLA images obtained with the S9.6 Ab (e.g. Fig 1F, EV2...). How abundant are they and what is their significance? The authors should consider presenting PLA images that encompass more than one cell for clarity.

We have double-checked all images acquired for this experiment and can confirm that these cytoplasmic foci are rare and very weak in comparison to "real" PLA foci (see nuclear foci) and most likely are a background. Furthermore, our Cell profiler pipeline only quantifies foci that are above certain size threshold.

2 - The authors should include a control experiment, such as gammaH2AX immunofluorescence, to demonstrate that laser irradiation indeed induces DSBs along the stripe, providing additional support for the study's findings.

We have added this control experiment. This is now shown in EV Figure 1C.

3 - Some of the original literature supporting the rationale of the present study are not properly cited (e.g. not all papers originally showing a link between active transcription and the DSB repair pathways - a central aspect of this study - are cited. Same for the evidence that transcription initiates at DSBs).

We apologise for this and have added more references to the Introduction.

4 - The observation that RNaseH overexpression significantly increases MNase accessibility raises the hypothesis that MNase might encounter difficulty digesting DNA engaged in the DNA:RNA hybrid moiety. Could this be the case? Additional clarification on this point would strengthen the study's findings.

We thank this reviewer for this interesting observation. If DNA:RNA hybrids were difficult to digest by MNase, segments with lengths not equivalent to $n \times$ mono-nucleosome would likely be detected on the gel. However, we did not observe these DNA fragments on gels. Furthermore, we observed chromatin relaxation in samples from cells depleted of GATAD2B. In these samples the levels of DNA:RNA hybrids would not be affected. Therefore, it is unlikely that DNA:RNA hybrids would significantly affect MNase digestion activities.

1. Allen, H.F., Wade, P.A. and Kutateladze, T.G. (2013) The NuRD architecture. *Cell Mol Life Sci*, **70**, 3513-3524.
2. Smeenk, G., Wiegant, W.W., Vrolijk, H., Solari, A.P., Pastink, A. and van Attikum, H. (2010) The NuRD chromatin-remodeling complex regulates signaling and repair of DNA damage. *J Cell Biol*, **190**, 741-749.
3. Polo, S.E., Kaidi, A., Baskcomb, L., Galanty, Y. and Jackson, S.P. (2010) Regulation of DNA-damage responses and cell-cycle progression by the chromatin remodelling factor CHD4. *EMBO J*, **29**, 3130-3139.
4. Chou, D.M., Adamson, B., Dephoure, N.E., Tan, X., Nottke, A.C., Hurov, K.E., Gygi, S.P., Colaiacovo, M.P. and Elledge, S.J. (2010) A chromatin localization screen reveals poly (ADP ribose)-regulated recruitment of the repressive polycomb and NuRD complexes to sites of DNA damage. *Proc Natl Acad Sci U S A*, **107**, 18475-18480.
5. Larsen, D.H., Poinsignon, C., Gudjonsson, T., Dinant, C., Payne, M.R., Hari, F.J., Rendtlew Danielsen, J.M., Menard, P., Sand, J.C., Stucki, M. *et al.* (2010) The chromatin-remodeling factor CHD4 coordinates signaling and repair after DNA damage. *J Cell Biol*, **190**, 731-740.
6. Spruijt, C.G., Luijsterburg, M.S., Menafra, R., Lindeboom, R.G., Jansen, P.W., Edupuganti, R.R., Baltissen, M.P., Wiegant, W.W., Voelker-Albert, M.C., Matarese, F. *et al.* (2016) ZMYND8 Co-localizes with NuRD on Target Genes and Regulates Poly(ADP-Ribose)-Dependent Recruitment of GATAD2A/NuRD to Sites of DNA Damage. *Cell Rep*, **17**, 783-798.
7. Bou-Nader, C., Bothra, A., Garboczi, D.N., Leppla, S.H. and Zhang, J. (2022) Structural basis of R-loop recognition by the S9.6 monoclonal antibody. *Nat Commun*, **13**, 1641.

8. Aymard, F., Bugler, B., Schmidt, C.K., Guillou, E., Caron, P., Briois, S., Iacovoni, J.S., Daburon, V., Miller, K.M., Jackson, S.P. *et al.* (2014) Transcriptionally active chromatin recruits homologous recombination at DNA double-strand breaks. *Nat Struct Mol Biol*, **21**, 366-374.
9. Dumelie, J.G. and Jaffrey, S.R. (2017) Defining the location of promoter-associated R-loops at near-nucleotide resolution using bisDRIP-seq. *Elife*, **6**.
10. Ray Chaudhuri, A. and Nussenzweig, A. (2017) The multifaceted roles of PARP1 in DNA repair and chromatin remodelling. *Nat Rev Mol Cell Biol*, **18**, 610-621.
11. Laspata, N., Kaur, P., Mersaoui, S.Y., Muoio, D., Liu, Z.S., Bannister, M.H., Nguyen, H.D., Curry, C., Pascal, J.M., Poirier, G.G. *et al.* (2023) PARP1 associates with R-loops to promote their resolution and genome stability. *Nucleic Acids Res*, **51**, 2215-2237.
12. Laspata, N., Muoio, D. and Fouquerel, E. (2024) Multifaceted Role of PARP1 in Maintaining Genome Stability Through Its Binding to Alternative DNA Structures. *J Mol Biol*, **436**, 168207.
13. Sukhanova, M.V., Abrakhi, S., Joshi, V., Pastre, D., Kutuzov, M.M., Anarbaev, R.O., Curmi, P.A., Hamon, L. and Lavrik, O.I. (2016) Single molecule detection of PARP1 and PARP2 interaction with DNA strand breaks and their poly(ADP-ribosylation) using high-resolution AFM imaging. *Nucleic Acids Res*, **44**, e60.
14. Clouaire, T., Rocher, V., Lashgari, A., Arnould, C., Aguirrebengoa, M., Biernacka, A., Skrzypczak, M., Aymard, F., Fongang, B., Dojer, N. *et al.* (2018) Comprehensive Mapping of Histone Modifications at DNA Double-Strand Breaks Deciphers Repair Pathway Chromatin Signatures. *Mol Cell*, **72**, 250-262 e256.

Prof. Monika Gullerova
University of Oxford
Sir William Dunn School of Pathology
South Parks Road
Oxford OX1 3RE
United Kingdom

16th Apr 2024

Re: EMBOJ-2023-116131R
GATAD2B-NuRD complex drives DNA:RNA hybrid-dependent chromatin boundary formation upon DNA damage

Dear Monika,

Thank you for submitting your revised manuscript to The EMBO Journal. All three original referees have now looked at it again, and except for a few presentational concerns, were all satisfied with the revisions. Following incorporation of these remaining points, we shall therefore be happy to accept the study for EMBO Journal publication!

In addition, there are a number of editorial issues that need to be addressed at this stage:

- The complexity and small labeling of the main figures is likely to compromise their readability in the final version. Therefore, please carefully review the attached Figure preparation guide (also available from our Guide to Authors website) and rearrange the figures and especially their labeling accordingly. If necessary, you might use more than 7 main figures.
- Please make sure to fully complete the Author Checklist, which currently lacks manuscript/journal/author information in the header.
- Please rename the conflict of interest statement into "Disclosure and competing interests statement" as specified in our Guide to Authors.
- As we are switching from a free-text author contribution statement towards a more formal statement based on Contributor Role Taxonomy (CRediT) terms, please remove the present Author Contribution section and instead specify each author's contribution(s) directly in the Author Information page of our submission system during upload of the final manuscript. See <https://casrai.org/credit/> for more information.
- In the Data Availability section, please include hyperlinks to the specific databases (PRIDE, GEO) to which data have been submitted. Also, please remove the referee access tokens now, and ensure that the data will be promptly released latest at the time of online publication.
- Please double-check to make sure to all relevant funding information in the manuscript is congruent with the info entered into our submission system; several listed sources are currently not present in the submission system (EPA Trust Fund [BVR01670]; Lee Placito Fund; grant number 32090033 for the National Natural Science Foundation of China; The Science and Technology Program of the Guangdong Province in China (2017B030301016), and the Shenzhen Municipal Commission of Science and Technology Innovation (JCYJ20200109114214463))
- Please upload the source data files as one ZIP archive per figure, instead of one archive for all figures. Also, please double-check the contents, as it seems that source data for Figure 6C may still be missing.
- Please note that Supplementary Materials have been superseded by more specific file types at EMBO Press (see: www.embopress.org/page/journal/14602075/authorguide#expandedview). Therefore, I would suggest converting the "Supplementary table" into a "Reagent and Tools Table" and refer to it as such. Ideally, please adhere to one of the attached templates for this table.
- The file with "Supplementary Figures and Legends" should be renamed "Appendix". It needs a front page with a brief Table of Contents, and each Figure Legend should be directly below the respective figure. Importantly, please rename all these figures into "Appendix Figure S1/2/3...", in the Appendix file as well as when referring to them in the text.
- Finally, our data editors have raised the following queries regarding figures, data, and legends - please make the required modifications with activated TRACK CHANGES option in the manuscript file to facilitate our checking, and also briefly describe your answers/actions in the resubmission cover letter:
 1. Please define the annotated p values ***/**/* in the legend of figure 5c; 7a-b; as appropriate.

2. Please indicate the statistical test used for data analysis in the legends of figures 1c; 3c, f, h; 4c; 5c; EV 5a-d.
3. Please note that in figures 1f; 2f; 6b; EV 5d; there is a mismatch between the annotated p values in the figure legend and the annotated p values in the figure file that should be corrected.
4. Please note that the box plots need to be defined in terms of minima, maxima, centre, bounds of box and whiskers, and percentile in the legends of figures 3c, f, h; 4c; 5c.
5. Please note that information related to 'n' is missing in the legends of figures 3c, g, h; 4c; 5c, f; 6e; 7a-c; EV 1g; EV 4b-c.
6. Although 'n' is provided, please describe the nature of entity for 'n' in the legends of figures 1e-g; 6a-b; EV 1b, f; EV 2b-d; EV 3a-e; EV 4a, EV 5e-g.
7. Please note that the error bars are not defined in the legends of figures 5b, f; 7a-b; EV 1g; EV 5a-d.
8. Please note that scale bar and its definition are missing for figures 5a, 7c.

I am therefore returning the manuscript to you for a final round of minor revision, to allow you to make these adjustments and clarifications, and to upload all modified files. Once we will have received them, we should hopefully be able to swiftly proceed with formal acceptance and production of the manuscript.

With kind regards,

Hartmut

- 1) Every manuscript requires a Data Availability section (even if only stating that no deposited datasets are included). Primary datasets or computer code produced in the current study have to be deposited in appropriate public repositories prior to resubmission, and reviewer access details provided in case that public access is not yet allowed. Further information: embopress.org/page/journal/14602075/authorguide#dataavailability
- 2) Each figure legend must specify
 - size of the scale bars that are mandatory for all micrograph panels
 - the statistical test used to generate error bars and P-values
 - the type error bars (e.g., S.E.M., S.D.)
 - the number (n) and nature (biological or technical replicate) of independent experiments underlying each data point
 - Figures may not include error bars for experiments with $n < 3$; scatter plots showing individual data points should be used instead.
- 3) Revised manuscript text (including main tables, and figure legends for main and EV figures) has to be submitted as editable text file (e.g., .docx format). We encourage highlighting of changes (e.g., via text color) for the referees' reference.
- 4) Each main and each Expanded View (EV) figure should be uploaded as individual production-quality files (preferably in .eps, .tif, .jpg formats). For suggestions on figure preparation/layout, please refer to our Figure Preparation Guidelines: <http://bit.ly/EMBOPressFigurePreparationGuideline>
- 5) Point-by-point response letters should include the original referee comments in full together with your detailed responses to them (and to specific editor requests if applicable), and also be uploaded as editable (e.g., .docx) text files.
- 6) Please complete our Author Checklist, and make sure that information entered into the checklist is also reflected in the manuscript; the checklist will be available to readers as part of the Review Process File. A download link is found at the top of our Guide to Authors: embopress.org/page/journal/14602075/authorguide
- 7) All authors listed as (co-)corresponding need to deposit, in their respective author profiles in our submission system, a unique ORCID identifier linked to their name. Please see our Guide to Authors for detailed instructions.
- 8) Please note that supplementary information at EMBO Press has been superseded by the 'Expanded View' for inclusion of additional figures, tables, movies or datasets; with up to five EV Figures being typeset and directly accessible in the HTML version of the article. For details and guidance, please refer to: embopress.org/page/journal/14602075/authorguide#expandedview

9) Digital image enhancement is acceptable practice, as long as it accurately represents the original data and conforms to community standards. If a figure has been subjected to significant electronic manipulation, this must be clearly noted in the figure legend and/or the 'Materials and Methods' section. The editors reserve the right to request original versions of figures and the original images that were used to assemble the figure. Finally, we generally encourage uploading of numerical as well as gel/blot image source data; for details see: embopress.org/page/journal/14602075/authorguide#sourcedata

At EMBO Press, we ask authors to provide source data for the main manuscript figures. Our source data coordinator will contact you to discuss which figure panels we would need source data for and will also provide you with helpful tips on how to upload and organize the files.

In the interest of ensuring the conceptual advance provided by the work, we recommend submitting a revision within 3 months (15th Jul 2024). Please discuss the revision progress ahead of this time with the editor if you require more time to complete the revisions. Use the link below to submit your revision:

Link Not Available

Referee #1:

With the inclusion of additional data and discussion, the manuscript is suitable for publication.

Referee #2:

The revised manuscript is very much improved and includes new experiments that strengthen the conclusions. The authors have addressed my concerns satisfactorily.

I would suggest the following minor changes concerning the PCR-based DNA end assay:

1. The presentation of the PCR-based DNA end resection results (new Fig.6E) is very succinct and non-expert readers may have difficulties understanding this experiment. I would recommend adding a short explanation of the rationale of the method (in the main text or figure legend).
2. The reference to Zhou et al. should be included in the methods description.
3. Please indicate what "si NC" is in the figure legend.
4. The description of the method should also be corrected: "Control or Drosha siRNA were transfected..." cannot be correct, I guess it should be GATAD2B siRNA.

Referee #3:

The authors have done an excellent job addressing the concerns and suggestions raised by the reviewers. The revisions have significantly improved the overall quality of the paper. I believe the manuscript is now ready for publication in the EMBO Journal.

Prof. Monika Gullerova
University of Oxford
Sir William Dunn School of Pathology
South Parks Road
Oxford OX1 3RE
United Kingdom

18th Apr 2024

Re: EMBOJ-2023-116131R1
The GATAD2B-NuRD complex drives DNA:RNA hybrid-dependent chromatin boundary formation upon DNA damage

Dear Monika,

Thank you for submitting your final revised manuscript for our consideration. I am pleased to inform you that we have now accepted it for publication in The EMBO Journal.

With kind regards,

Hartmut
